# Efficient Source-Free Time-Series Adaptation via Parameter Subspace Disentanglement

**Gaurav Patel**[†*]**, Chris Sandino**[‡]**, Behrooz Mahasseni**[‡]**, Ellen Zippi**[‡]
**Erdrin Azemi**[‡]**, Ali Moin**[‡]**, Juri Minxha**[‡]
Purdue University[†], Apple[‡]

## Abstract

In this paper, we propose a framework for efficient Source-Free Domain Adaptation (SFDA) in the context of time-series, focusing on enhancing both parameter efficiency and data-sample utilization. Our approach introduces an improved paradigm for source-model preparation and target-side adaptation, aiming to enhance training efficiency during target adaptation. Specifically, we reparameterize the source model's weights in a Tucker-style decomposed manner, factorizing the model into a compact form during the source model preparation phase. During target-side adaptation, only a subset of these decomposed factors is fine-tuned, leading to significant improvements in training efficiency. We demonstrate using PAC Bayesian analysis that this selective fine-tuning strategy implicitly regularizes the adaptation process by constraining the model's learning capacity. Furthermore, this re-parameterization reduces the overall model size and enhances inference efficiency, making the approach particularly well suited for resource-constrained devices. Additionally, we demonstrate that our framework is compatible with various SFDA methods and achieves significant computational efficiency, reducing the number of fine-tuned parameters and inference overhead in terms of MACs by over 90% while maintaining model performance.

## 1 Introduction

In a typical Source-Free Domain Adaptation (SFDA) setup, the source-pretrained model must adapt to the target distribution using unlabeled samples from the target domain. SFDA strategies have become prevalent due to the restrictive nature of conventional domain adaptation methods (Li et al., 2024) which require access to both source and target domain data simultaneously and therefore may not be feasible in real-world scenarios due to privacy and confidentiality concerns. Although SFDA techniques have been extensively investigated for visual tasks (Liang et al., 2020; 2021; Li et al., 2020; Yang et al., 2021b;a; 2022; 2023; Kim et al., 2021; Xia et al., 2021; Kundu et al., 2021; 2022b), their application to time series analysis remains relatively nascent (Ragab et al., 2023b; Gong et al., 2024). Nevertheless, time-series models adaptation is crucial due to the nonstationary and heterogeneous nature of time-series data (Park et al., 2023), where each user's data exhibit distinct patterns, necessitating adaptive models that can learn idiosyncratic features.

Despite growing interest in SFDA, sample and parameter efficiency during adaptation is still largely unexplored (Karim et al., 2023; Lee et al., 2023). These aspects of SFDA are of particular importance in situations where there is a large resource disparity between the source and the target. For instance, a source-pretrained model may be deployed to a resource-constrained target device, rendering full model adaptation impractical. Additionally, the target-side often has access to substantially fewer reliable samples compared to the volume of data used during source pretraining, increasing the risk of overfitting.

In response to these challenges, we revisit both the *source-model preparation* and *target-side adaptation* processes. We demonstrate that disentangling the backbone parameter subspace during *source-model preparation* and then fine-tuning a selected subset of those parameters during *target-side adaptation* leads to a computationally efficient adaptation process, while maintaining superior predictive performance. Figure 1B illustrates our proposed framework. During source-model prepa-

---

[*]Work done during an internship at Apple.
Contact: gpatel10@purdue.edu, j_minxha@apple.com

ration, we re-parameterize the backbone weights in a low-rank Tucker-style factorized (Tucker, 1966; Lathauwer et al., 2000) way, decomposing the model into *compact parameter subspaces*. Tucker-style factorization offers great flexibility and interpretability by breaking down tensors into a *core tensor* and *factor matrices* along each mode, capturing multi-dimensional interactions while independently reducing dimensionality. This reparameterization results in a model that is both parameter- and computation-efficient, significantly reducing the model size and inference overhead in terms of Multiply-Accumulate Operations (MACs).

The re-parameterized source-model is then deployed to the target side. During target-side adaptation, we perform selective fine-tuning (SFT) within a selected subspace (*i.e.*, the *core tensor*) that constitutes a very small fraction of the total number of parameters in the backbone. Our findings show that this strategy not only enhances parameter efficiency but also serves as a form of regularization, mitigating overfitting to unreliable target samples by restricting the model's learning capacity. We provide theoretical insights into this regularization effect using PAC-Bayesian generalization bounds (McAllester, 1998; 1999; Li & Zhang, 2021; Wang et al., 2023) for the fine-tuned target model.

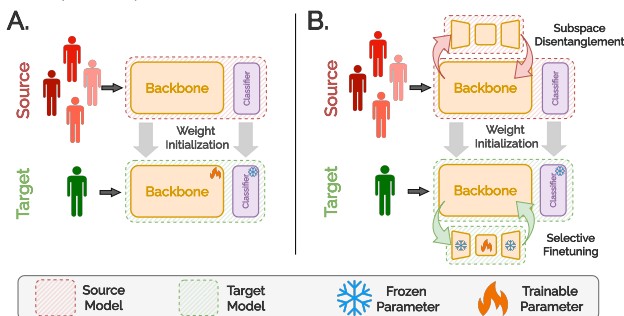

Figure 1: **Prior Paradigm vs. Ours. A.** Existing approaches (Liang et al., 2020; Yang et al., 2021a; 2022; Ragab et al., 2023b) adapt the entire model to the target distribution. **B.** Our method disentangles the backbone parameters using Tucker-style factorization. Fine-tuning a small subset of these parameters proves both parameter- and sample-efficient (*cf*. Section 3), while also being robust against overfitting (*cf*. Figure 2B and Section 3), leading to superior adaptation to the target distribution.

Empirical results demonstrate that low-rank weight disentanglement during source-model preparation enables parameter-efficient adaptation on the target side, consistently improving performance across various SFDA methods (Liang et al., 2020; Yang et al., 2021a; 2022; Ragab et al., 2023b) and time-series benchmarks (Ragab et al., 2023a;b). It is important to emphasize that our contribution does not introduce a novel SFDA method for time-series data. Instead, we focus on making the target adaptation process more parameter- and sample-efficient, and demonstrating that our framework can be seamlessly integrated with existing, and potentially future, SFDA techniques. In summary, our key contributions are:

1. We propose a novel strategy for source-model preparation by reparameterizing the backbone network's weights using low-rank Tucker-style factorization. This decomposition into a *core tensor* and *factor matrices* creates a compact parameter subspace representation, leading to a parameter- and computation-efficient model with reduced size and inference overhead.

2. During target-side adaptation, we introduce a selective fine-tuning (SFT) approach that adjusts only a small fraction of the backbone's parameters (*i.e.*, the *core tensor*) within the decomposed subspace. This strategy enhances parameter efficiency and acts as an implicit regularization mechanism, mitigating the risk of overfitting to unreliable target samples by limiting the model's learning capacity.

3. We ground the regularization effect of SFT using the PAC Bayesian generalization bound. Empirical analyses demonstrate that our proposed framework improves the parameter and sample efficiency of adaptation process of various existing SFDA methods across the time-series benchmarks, showcasing its generalizability.

## 2 RELATED WORK AND OBSERVATIONS

**Unsupervised Domain Adaptation in Time Series.** Unsupervised Domain Adaptation (UDA) for time-series data addresses the critical challenge of distribution shifts between the source and target domains. Discrepancy-based methods, such as those of Cai et al. (2021), Liu & Xue (2021) and He et al. (2023), align feature representations through statistical measures, while adversarial approaches (Wilson et al., 2020; 2023; Ragab et al., 2022; Jin et al., 2022; Ozyurt et al., 2023) focus on

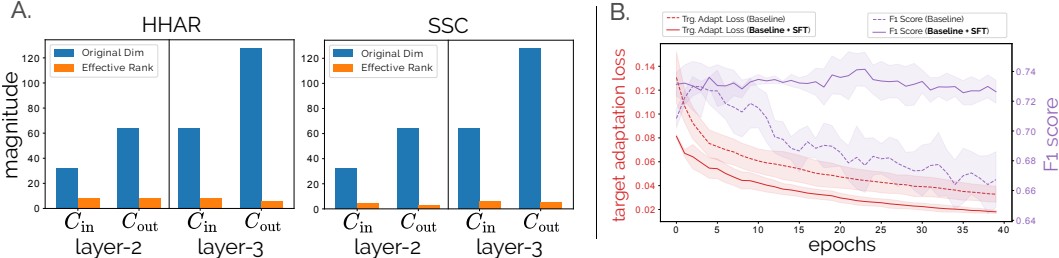

Figure 2: **A. Original Dimensions vs. Effective Rank**: Comparison of input/output channel dimensions ($C_{\text{in}}, C_{\text{out}}$) in the last two layers of the source model trained on the HHAR (Stisen et al., 2015) and SSC (Goldberger et al., 2000) datasets, alongside their effective rank computed via VBMF (Nakajima et al., 2013). **B. Regularization Effect**: Training dynamics illustrate the regularization effect of our source decomposition and selective fine-tuning (SFT) compared to the SHOT (Baseline) (Liang et al., 2020) on the SSC dataset.

minimizing distributional discrepancies through adversarial training. The comprehensive study by Ragab et al. (2023a) provides a broad overview of domain adaptation in time series. However, SFDA assumes access to only the source-pretrained model (Liang et al., 2020; 2021; Li et al., 2020; Yang et al., 2021a; 2022; 2023; Kim et al., 2021; Xia et al., 2021). In SFDA, one of the most commonly used strategies is based on unsupervised clustering techniques (Yang et al., 2022; Li et al., 2024), promoting the *discriminability* and *diversity* of feature spaces through information maximization (Liang et al., 2020), neighborhood clustering (Yang et al., 2021a) or contrastive learning objectives (Yang et al., 2022). Recent advances incorporate auxiliary tasks, such as masking and imputation, to enhance SFDA performance, as demonstrated by Ragab et al. (2023b) and Gong et al. (2024), taking inspiration from Liang et al. (2021) and Kundu et al. (2022a) to include masked reconstruction as an auxiliary task. However, exploration of SFDA in time series contexts remains limited, especially with regard to parameter and sample efficiency during target-side adaptation (Ragab et al., 2023a; Liu et al., 2024; Gong et al., 2024; 2025; Furqon et al., 2025).

**Parameter Redundancy and Low-Rank Subspaces.** Neural network pruning has demonstrated that substantial reductions in parameters, often exceeding 90%, can be achieved with minimal accuracy loss, revealing significant redundancy in trained deep networks (LeCun et al., 1989; Hassibi & Stork, 1992; Li et al., 2017; Frankle & Carbin, 2018; Sharma et al., 2024). Structured pruning methods, such as those of Molchanov et al. (2017) and Hoefler et al. (2021), have optimized these reductions to maintain or even improve inference speeds. Low-rank models have played a crucial role in pruning, with techniques such as Singular Value Decomposition (SVD) (Eckart & Young, 1936) applied to fully connected layers (Denil et al., 2013) and tensor-train decomposition used to compress neural networks (Novikov et al., 2015). Methods developed by Jaderberg et al. (2014), Denton et al. (2014), Tai et al. (2016), and Lebedev et al. (2015) accelerate Convolutional Neural Networks (CNNs) through low-rank regularization. Decomposition models such as CANDECOMP/PARAFAC (CP) (Carroll & Chang, 1970) and Tucker decomposition (Tucker, 1966) have effectively reduced the computational complexity of CNNs (Lebedev et al., 2015; Kim et al., 2016). Recent works have extended these techniques to recurrent and transformer layers (Ye et al., 2018; Ma et al., 2019), broadening their applicability. In Figure 2A, we identify significant parameter redundancies in source-pretrained models in terms of effective parameter ranks, revealing low effective ranks, prompting us to employ Tucker decomposition for reparameterizing the weights. Tucker decomposition offers several advantages; it naturally accommodates the multi-modal structure of convolutional weight tensors by disentangling different modes (*e.g.*, temporal and channel dimensions), allows for mode-specific rank selection to capture varying complexities across dimensions, and enhances interpretability by modeling interactions within the weight tensors effectively. Compared to other methods like CP decomposition, which impose uniform ranks and lack flexibility, Tucker decomposition better aligns with our goals of parameter and sample efficiency during target adaptation. We utilize low-rank tensor factorization (Tucker-style decomposition) to disentangle weight tensors into compact *disentangled* subspaces, enabling selective fine-tuning during target adaptation. This strategy enhances the parameter efficiency of the adaptation process and implicitly mitigates overfitting. As shown in Figure 2B, the validation F1 score for baseline methods declines over epochs, whereas our proposed SFT method maintains consistent performance, demonstrating its robustness against overfitting.

Figure 3: Illustration of Tucker-style factorization (Tucker, 1966; Lathauwer et al., 2000).

## 3 PROPOSED METHODOLOGY

In this section, we discuss our proposed methodology through several key steps: **(1)** We begin by discussing the SFDA setup. **(2)** We discuss the Tucker-style tensor factorization on the weight tensors of the pre-trained source-model, decomposing them into a core tensor and mode-specific factor matrices; where we introduce the *rank factor* ($RF$) hyperparameter to control the mode ranks in the decomposition, allowing for flexible trade-offs between model compactness and capacity. **(3)** During target-side adaptation, we selectively fine-tune the *core tensor* while keeping the mode factor matrices fixed, effectively adapting to distributional shifts in the target domain while reduced computational overhead and mitigating overfitting. **(4)** Finally, we offer theoretical insights grounded in PAC-Bayesian generalization analysis, demonstrating how source weight decomposition and target-side selective fine-tuning implicitly regularize the adaptation process, leading to enhanced generalization in predictive performance while rendering the process parameter- and sample-efficient.

**SFDA Setup.** SFDA aims to adapt a pre-trained source model to a target domain without access to the original source data. Specifically, in a classification problem, we start with a source dataset $\mathcal{D}_s = \{(x_s, y_s); x_s \in \mathcal{X}_s, y_s \in \mathcal{C}_g\}$, where $\mathcal{X}_s$ denotes the input space, and $\mathcal{C}_g$ represents the set of class labels for the *goal task* in a closed-set setting. On the target side, we have access to an unlabeled target dataset $\mathcal{D}_t = \{x_t; x_t \in \mathcal{X}_t\}$, where $\mathcal{X}_t$ is the input space for the target domain. The aim of SFDA is to learn a target function $f_t : \mathcal{X}_t \to \mathcal{C}_g$ that can accurately infer the labels $y_t \in \mathcal{C}_g$ for the samples in $\mathcal{D}_t$. $f_t$ is obtained using only the unlabeled target samples in $\mathcal{D}_t$, and the source-model $f_s : \mathcal{X}_s \to \mathcal{C}_g$, which is pre-trained on the source domain. Each $x$ (either from the source or target domain) is a sample of multivariate time series, *i.e.*, $x \in \mathbb{R}^{M \times L}$, where $L$ is the number of time steps (sequence length) and $M$ denotes number of observations (channels) for the corresponding time step.

**Tucker-style Factorization.** After the supervised source pre-training (*cf*. Appendix A.9 for details on source pre-training), we adopt Tucker-style factorization (Tucker, 1966; Lathauwer et al., 2000) of the weight tensors for its effectiveness in capturing multi-way interactions among different mode-independent factors. The compact core tensor encapsulates these interactions, and the factor matrices in the decomposition offer low-dimensional representations for each tensor mode. As illustrated in Figure 3A, the rank-$(R_1, R_2, R_3)$ Tucker-style factorization of the 3-way tensor $\mathcal{A} \in \mathbb{R}^{I_1 \times I_2 \times I_3}$ is represented as:

$$\mathcal{A} = \mathcal{G} \times_1 \mathbf{U}^{(1)} \times_2 \mathbf{U}^{(2)} \times_3 \mathbf{U}^{(3)} \quad \text{or,} \quad \mathcal{A}_{i,j,k} = \sum_{r_1=1}^{R_1} \sum_{r_2=1}^{R_2} \sum_{r_3=1}^{R_3} \mathcal{G}_{r_1,r_2,r_3} \mathbf{U}^{(1)}_{i,r_1} \mathbf{U}^{(2)}_{j,r_2} \mathbf{U}^{(3)}_{k,r_3}, \quad (1)$$

where $\mathcal{G} \in \mathbb{R}^{R_1 \times R_2 \times R_3}$ is referred to as the *core tensor*, and $\mathbf{U}^{(1)} \in \mathbb{R}^{I_1 \times R_1}$, $\mathbf{U}^{(2)} \in \mathbb{R}^{I_2 \times R_2}$ and $\mathbf{U}^{(3)} \in \mathbb{R}^{I_3 \times R_3}$ as *factor matrices*.

Furthermore, the linear operations involved in Tucker-style representation (Equation (1)), which are primarily mode-independent matrix multiplications between the core tensor and factor matrices, simplify both mathematical treatment and computational implementation. This linearity ensures that linear operations (*e.g.*, convolution or linear projection) in deep networks using the complete tensor can be efficiently represented as sequences of linear suboperations on the core tensor and factor matrices.

For instance, in one-dimensional convolutional neural networks (1D-CNNs), the convolution operation transforms an input representation $I \in \mathbb{R}^{C_{\text{in}} \times L}$ into an output representation $O \in \mathbb{R}^{C_{\text{out}} \times L'}$ using a kernel $\mathcal{W} \in \mathbb{R}^{C_{\text{out}} \times C_{\text{in}} \times K}$. Here, $C_{\text{in}}$, $C_{\text{out}}$, $K$, $L$, and $L'$ denote the number of input channels, output channels, kernel size, input sequence length, and output sequence length, respectively.

The operation is mathematically defined as:

$$O_{i,l} = \sum_{j=1}^{C_{\text{in}}} \sum_{k=-\frac{K}{2}}^{\frac{K}{2}} \boldsymbol{\mathcal{W}}_{i,j,k} I_{j,l-k}. \tag{2}$$

Given that the dimensions of the channels ($C_{\text{in}}$ and $C_{\text{out}}$) are typically much larger than the size of the kernel ($K$), we restrict the decomposition to modes 1 and 2 only, focusing on the input and output channels. The 2-mode decomposition is represented by:

$$\boldsymbol{\mathcal{W}} = \boldsymbol{\mathcal{T}} \times_1 \mathbf{V}^{(1)} \times_2 \mathbf{V}^{(2)} \quad \text{or,} \quad \boldsymbol{\mathcal{W}}_{i,j,k} = \sum_{r_1=1}^{R_{\text{out}}} \sum_{r_2=1}^{R_{\text{in}}} \boldsymbol{\mathcal{T}}_{r_1,r_2,k} \mathbf{V}_{i,r_1}^{(1)} \mathbf{V}_{j,r_2}^{(2)}, \tag{3}$$

where $\boldsymbol{\mathcal{T}} \in \mathbb{R}^{R_{\text{out}} \times R_{\text{in}} \times K}$ is the 2-mode decomposed core tensor, and $\mathbf{V}^{(1)} \in \mathbb{R}^{C_{\text{out}} \times R_{\text{out}}}$ and $\mathbf{V}^{(2)} \in \mathbb{R}^{C_{\text{in}} \times R_{\text{in}}}$ represent the respective factor matrices, also illustrated in Figure 3B. With this decomposed form, the convolution operation can be represented as a sequence of the following linear operations:

$$Z_{r_2,l} = \sum_{j=1}^{C_{\text{in}}} \mathbf{V}_{j,r_2}^{(2)} I_{j,l}, \qquad \text{(Channel Down-Projection)} \tag{4}$$

$$Z'_{r_1,l} = \sum_{r_2=1}^{R_{\text{in}}} \sum_{k=-\frac{K}{2}}^{\frac{K}{2}} \boldsymbol{\mathcal{T}}_{r_1,r_2,k} Z_{r_2,l-k}, \qquad \text{(Core Convolution)} \tag{5}$$

$$O_{i,l'} = \sum_{r_1=1}^{R_{\text{out}}} \mathbf{V}_{i,r_1}^{(1)} Z'_{r_1,l'}, \qquad \text{(Channel Up-Projection)} \tag{6}$$

where both the *Channel Down-Projection* and the *Channel Up-Projection* operations are implemented as unit window-sized convolution operations.

Building on these insights, we utilize Tucker-style decomposition to reparameterize the weights, as it allows us to express convolution operations as sequences of linear operations that can each be implemented using standard convolutional layers. This reparameterization simplifies integration into existing architectures and leverages efficient convolution computations, facilitating both ease of implementation, computational efficiency, and overall reduction in the total number of parameters. We decompose the pre-trained source model weights using Tucker decomposition, optimized via the Higher-Order Orthogonal Iteration (HOOI) algorithm (De Lathauwer et al., 2000; Kolda & Bader, 2009), an alternating least squares method for low-rank tensor approximation (detailed algorithm in Appendix A.1). The optimization problem is defined as:

$$\boldsymbol{\mathcal{T}}^*, \mathbf{V}^{(1)*}, \mathbf{V}^{(2)*} = \operatorname*{arg\,min}_{\boldsymbol{\mathcal{T}}, \mathbf{V}^{(1)}, \mathbf{V}^{(2)}} \|\boldsymbol{\mathcal{W}} - \boldsymbol{\mathcal{T}} \times_1 \mathbf{V}^{(1)} \times_2 \mathbf{V}^{(2)}\|_F. \tag{7}$$

The weight tensor is then re-parameterized with the core tensor $\boldsymbol{\mathcal{T}}^* \in \mathbb{R}^{R_{\text{out}} \times R_{\text{in}} \times K}$ and mode factor matrices $\mathbf{V}^{(1)*} \in \mathbb{R}^{C_{\text{out}} \times R_{\text{out}}}$ and $\mathbf{V}^{(2)*} \in \mathbb{R}^{C_{\text{in}} \times R_{\text{in}}}$, as formulated in Equations (4), (5), and (6).

However, the re-parameterization is performed after minimizing the linear reconstruction error of the weights (Equation 7), which may degrade the predictive performance on the source domain (Kim et al., 2016). To mitigate this effect, we fine-tune the core tensor and factor matrices using the source data for an additional 2-3 epochs, prior to deploying the model on the target side. This fine-tuning effectively restores the original predictive performance, preserving the model's performance while leveraging the benefits of fewer parameters (*cf*. Figure 11). This behavior strongly aligns with the *Lottery Ticket Hypothesis* (Frankle & Carbin, 2018), which suggests that compressing and retraining pre-trained, over-parameterized deep networks can result in superior performance and storage efficiency compared to training smaller networks from scratch. Similar observations have been made by Zimmer et al. (2023) in the context of large language model training.

Moreover, we introduce the *rank factor* ($RF$), a hyperparameter to standardize the mode rank analysis. The mode ranks are set as $(R_{\text{in}}, R_{\text{out}}) = \left(\left\lfloor \frac{C_{\text{in}}}{RF} \right\rfloor, \left\lfloor \frac{C_{\text{out}}}{RF} \right\rfloor\right)$, where a higher $RF$ results in lower

mode ranks, and vice versa. Although $RF$ can be set independently for the input and output channels, for simplicity in our analysis, we control both $R_{\text{in}}$ and $R_{\text{out}}$ with a single hyperparameter, $RF$. This tunable parameter allows for flexible control over the trade-off between parameter reduction and model capacity.

**Target Side Adaptation.** The *core* tensor in our setup plays a pivotal role by capturing multi-way interactions between different modes, encoding essential inter-modal relationships with the factor matrices. Additionally, its sensitivity to temporal dynamics (*cf*. Equation (5)) makes it particularly well-suited for addressing distributional shifts in time-series data, where discrepancies often arise due to temporal variations (Fan et al., 2023). Therefore, by selectively fine-tuning the core tensor during target adaptation, we can effectively adapt to these shifts while maintaining a compact parameterization. Figure 4 presents a toy experiment that demonstrates how fine-tuning the core tensor suffices in addressing domain shift, aligning the model more effectively with the target domain. This approach not only ensures that the model aligns better with the target domain but also enhances parameter efficiency by reducing the number of parameters that need to be updated, minimizing computational overhead. Moreover, selective fine-tuning mitigates the risk of overfitting, providing a more robust adaptation process, as shown in Figure 2B. This strategy leverages the expressiveness of the core tensor to address temporal domain shifts while maintaining the computational benefits of a structured, low-rank parameterization.

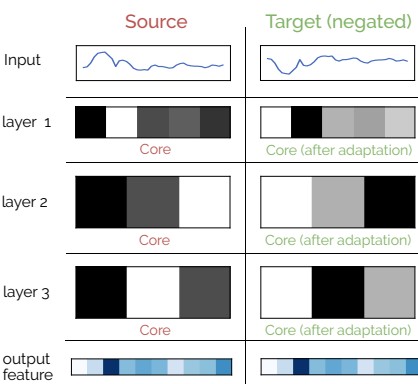

Figure 4: The two columns show the inputs from MNIST-1D (Greydanus & Kobak, 2024) and the core tensors for the source and target domains. The target domain is generated manually by vertically flipping (*i.e.*, negating) the source. Both $R_{\text{in}}$ and $R_{\text{out}}$ are set to 1 (*cf*. Figure 3B) to facilitate visualization of the core tensors. Domain-invariant output features are achieved as the core tensor adapts to the target samples to mitigate the domain shift (negation).

**Computational and Parameter Efficiency.** The decomposition significantly reduces the parameter count from $C_{\text{out}} \times C_{\text{in}} \times K$ to:

$$\underbrace{R_{\text{out}} \times R_{\text{in}} \times K}_{\text{Core Tensor } (\boldsymbol{\mathcal{T}})} + \underbrace{C_{\text{out}} \times R_{\text{out}}}_{\text{Factor Matrix } (\mathbf{V}^{(1)})} + \underbrace{C_{\text{in}} \times R_{\text{in}}}_{\text{Factor Matrix } (\mathbf{V}^{(2)})} \tag{8}$$

where $R_{\text{out}} = \left\lfloor \frac{C_{\text{out}}}{RF} \right\rfloor \ll C_{\text{out}}$ and $R_{\text{in}} = \left\lfloor \frac{C_{\text{in}}}{RF} \right\rfloor \ll C_{\text{in}}$, ensuring substantial parameter savings. Furthermore, since we selectively finetune the *core* tensor, the parameter efficiency is further enhanced.

In terms of computational efficiency, the factorized convolution reduces the operation count from $\mathcal{O}(C_{\text{out}} \times C_{\text{in}} \times K \times L')$ to a series of smaller convolutions with complexity:

$$\underbrace{\mathcal{O}(C_{\text{in}} \times R_{\text{in}} \times L)}_{\text{Equation (4)}} + \underbrace{\mathcal{O}(R_{\text{out}} \times R_{\text{in}} \times K \times L')}_{\text{Equation (5)}} + \underbrace{\mathcal{O}(C_{\text{out}} \times R_{\text{out}} \times L')}_{\text{Equation (6)}} \tag{9}$$

resulting in a notable reduction in computational cost, dependent on the values of $R_{\text{out}}$ and $R_{\text{in}}$ (see Appendix A.2 for a detailed explanation).

**Robust Target Adaptation.** In this subsection, we aim to uncover the underlying factors contributing to the robustness of the proposed SFT strategy, particularly in terms of sample efficiency and its implicit regularization effects during adaptation. To achieve this, we leverage the PAC-Bayesian generalization theory (McAllester, 1998; 1999), which provides a principled framework for bounding the generalization error in deep neural networks during fine-tuning (Dziugaite & Roy, 2017; Li & Zhang, 2021; Wang et al., 2023). We analyze the generalization error on the network parameters $\boldsymbol{W} = \{\boldsymbol{\mathcal{W}}^{(i)}\}_{i=1}^{D}$, *i.e.*, $\mathcal{L}(\boldsymbol{W}) - \hat{\mathcal{L}}(\boldsymbol{W})$, where $\mathcal{L}(\boldsymbol{W})$ is the test loss, $\hat{\mathcal{L}}(\boldsymbol{W})$ is the empirical training loss, and $D$ denotes the number of layers.

**Theorem 1.** *(PAC-Bayes generalization bound for fine-tuning) Let $\boldsymbol{W}$ be some hypothesis class (network parameters). Let $P$ be a prior (source) distribution on $\boldsymbol{W}$ that is independent of the target training set. Let $Q(S)$ be a posterior (target) distribution on $\boldsymbol{W}$ that depends on the target training set $S$ consisting of $n$ number of samples. Suppose the loss function $\mathcal{L}(.)$ is bounded by $C$. If we set*

*the prior distribution $P = \mathcal{N}(\boldsymbol{W}_{src}, \sigma^2 I)$, where $\boldsymbol{W}_{src}$ are the weights of the pre-trained network. The posterior distribution $Q(S)$ is centered at the fine-tuned model as $\mathcal{N}(\boldsymbol{W}_{trg}, \sigma^2 I)$. Then with probability $1 - \delta$ over the randomness of the training set, the following holds:*

$$\mathbb{E}_{\boldsymbol{W} \sim Q(S)} \left[ \mathcal{L}(\boldsymbol{W}) \right] \leq \mathbb{E}_{\boldsymbol{W} \sim Q(S)} \left[ \hat{\mathcal{L}}(\boldsymbol{W}, S) \right] + C \sqrt{\frac{\sum_{i=1}^{D} \|\mathcal{W}_{trg}^{(i)} - \mathcal{W}_{src}^{(i)}\|_F^2}{2\sigma^2 n} + \frac{k \ln \frac{n}{\delta} + l}{n}}. \quad (10)$$

*for some $\delta, k, l > 0$, where $\mathcal{W}_{trg}^{(i)} \in \boldsymbol{W}_{trg}$, and $\mathcal{W}_{src}^{(i)} \in \boldsymbol{W}_{src}$, $\forall 1 \leq i \leq D$, $D$ denoting the total number of layers.*

*Proof.* See Appendix A.5 (Theorem 1) □

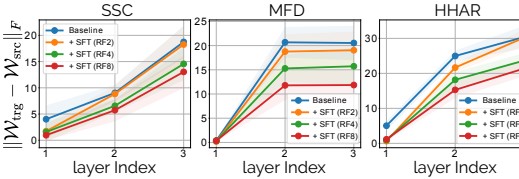

Figure 5: Layer-wise parameter distance between the source-pretrained model and the target-adapted model using the SFT strategy for different rank-factor values ($RF \in \{2, 4, 8\}$) on the SSC (Goldberger et al., 2000), MFD (Lessmeier et al., 2016), and HHAR (Stisen et al., 2015) datasets. The values represent the average parameter distances across all source-target pairs provided for the respective datasets. Lower values indicate smaller parameter distances.

In Equation (10), the test error (LHS) is bounded by the empirical training loss and the divergence between the source pre-trained weights ($\boldsymbol{W}_{src}$) and the target fine-tuned weights ($\boldsymbol{W}_{trg}$), measured using the layer-wise Frobenius norm. Motivated by this theoretical insight, we empirically analyze the layer-wise parameter distances between the source and target models in terms of the Frobenius norm. Figure 5 illustrates these distances for both vanilla fine-tuning (Baseline: Fine-tuning all parameters) and our SFT approach which selectively fine-tunes the core subspace, across various time series datasets, including SSC (Goldberger et al., 2000), MFD (Lessmeier et al., 2016), and HHAR (Stisen et al., 2015), for different $RF$ values. The results show that SFT naturally constrains the parametric distance between the source and target weights, with a higher $RF$ exhibiting a greater constraining ability, thus regulating the adaptation process in line with the distance-based regularization hypothesis of Gouk et al. (2021). Furthermore, in Appendix A.4 (Lemma 2), we provide a formal bound on the parameter distance in terms of the decomposed ranks of the weight matrices. This observation reveals a trade-off in the source-target parameter distance; while a smaller distance tightens the generalization bound, it also limits the model's capacity, potentially regularizing or, in extreme cases, hindering adaptability. In Section 4, we use this bound to further discuss SFT's sample efficiency.

## 4 EXPERIMENTS AND ANALYSIS

**Datasets and Methods.** We utilize the AdaTime benchmarks proposed by Ragab et al. (2023a;b) to evaluate the SFDA methods: SSC (Goldberger et al., 2000), and MFD (Lessmeier et al., 2016), HHAR (Stisen et al., 2015), UCIHAR (Anguita et al., 2013), WISDM (Kwapisz et al., 2011). Here each dataset involves distinct domains based on individual subjects (SSC), devices (HHAR, UCI-HAR, WISDM) or entities (MFD). For comprehensive dataset descriptions and domain details, refer to the Appendix A.6. To assess the effectiveness and generalizability of our decomposition framework, we integrate it with prominent SFDA methods: SHOT (Liang et al., 2020), NRC (Yang et al., 2021a), AAD (Yang et al., 2022), and MAPU (Ragab et al., 2023b) within time-series contexts. For more details on the adaptation methods we direct readers to Appendix A.7 and A.8.

**Experimental Setup.** Our experimental setup systematically evaluates the proposed strategy (SFT) alongside contemporary SFDA methods evaluated by Ragab et al. (2023b); Gong et al. (2024), including SHOT (Liang et al., 2020), NRC (Yang et al., 2021a), AAD (Yang et al., 2022), and MAPU (Ragab et al., 2023b). In SFT, the backbone fine-tuning is restricted to the core subspace of the decomposed parameters, and the hyperparameters for source training and target adaptation are kept consistent between each vanilla SFDA method and its SFT variant. In addition, we assess the impact of different ranking factors ($RF$). For values of $RF$ greater than 8, we observed a significant underfitting, leading us to limit our evaluation to $RF \in \{2, 4, 8\}$. The experiments are carried out using the predefined source-target pairs from the AdaTime benchmark (Ragab et al., 2023a).

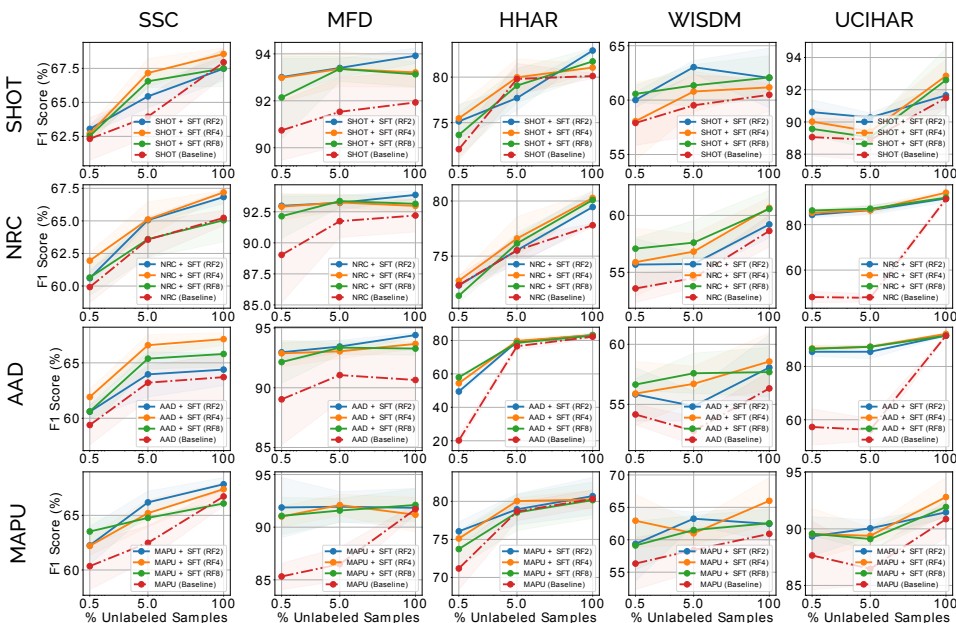

Figure 6: Comparison of the predictive performance of our selective fine-tuning (SFT) strategy w.r.t. F1 score (%) at different target sample ratios against baseline methods (SHOT, NRC, AAD, MAPU). Adaptations are conducted using 0.5%, 5%, and 100% of the total unlabeled target samples, randomly sampled in a stratified manner. The figure demonstrates performance differences across methods as the amount of target data varies, highlighting the sample efficiency of the proposed SFT strategy. Results are averaged over three runs.

The predictive performance of the methods is evaluated by comparing the average F1 score of the target-adapted model across all source-target pairs for the respective dataset in the benchmark. Each experiment is repeated three times with different random seeds to ensure robustness. More details are provided in Appendix A.9.

Table 1: Performance and efficiency comparison on SSC dataset across SFDA methods, reported as average F1 score (%) at target sample ratios (0.5%, 5%, 100%), inference MACs (M), and fine-tuned parameters (K). Highlighted rows show results for SFT, where only the core tensor is fine-tuned at different $RF$ values. Green numbers represent average percentage improvement, while Red numbers indicate reduction in MACs and fine-tuned parameters. In Appendix A.9 (Table 4), we extend this analysis to MFD, HHAR, WISDM, and UCIHAR datasets.

| Methods | $RF$ | F1 Score (%) ↑ | | | | MACs (M) ↓ | # Params. (K) ↓ |
| | | 0.5% | 5% | 100% | Average | | |
|---|---|---|---|---|---|---|---|
| SHOT (Liang et al., 2020) | - | 62.32 ± 1.57 | 63.95 ± 1.51 | 67.95 ± 1.04 | 64.74 | 12.92 | 83.17 |
| SHOT + SFT | 8 | 62.53 ± 0.46 | 66.55 ± 0.46 | 67.50 ± 1.33 | 65.53 (1.22%) | 0.80 (93.81%) | 1.38 (98.34%) |
| | 4 | 62.71 ± 0.57 | 67.16 ± 1.06 | 68.56 ± 0.44 | 66.14 (2.16%) | 1.99 (84.60%) | 5.32 (93.60%) |
| | 2 | 63.05 ± 0.32 | 65.44 ± 0.83 | 67.48 ± 0.89 | 65.32 (0.90%) | 5.54 (57.12%) | 20.88 (74.89%) |
| NRC (Yang et al., 2021a) | - | 59.92 ± 1.19 | 63.56 ± 1.35 | 65.23 ± 0.59 | 62.90 | 12.92 | 83.17 |
| NRC + SFT | 8 | 60.65 ± 1.37 | 63.60 ± 1.43 | 65.05 ± 1.66 | 63.10 (0.32%) | 0.80 (93.81%) | 1.38 (98.34%) |
| | 4 | 61.95 ± 0.62 | 65.11 ± 1.34 | 67.19 ± 0.14 | 64.75 (2.94%) | 1.99 (84.60%) | 5.32 (93.60%) |
| | 2 | 60.60 ± 0.58 | 65.06 ± 0.24 | 66.83 ± 0.51 | 64.16 (2.00%) | 5.54 (57.12%) | 20.88 (74.89%) |
| AAD (Yang et al., 2022) | - | 59.39 ± 1.80 | 63.21 ± 1.53 | 63.71 ± 2.06 | 62.10 | 12.92 | 83.17 |
| AAD + SFT | 8 | 60.62 ± 1.40 | 65.38 ± 0.93 | 65.80 ± 1.17 | 63.93 (2.95%) | 0.80 (93.81%) | 1.38 (98.34%) |
| | 4 | 61.92 ± 0.68 | 66.59 ± 0.95 | 67.14 ± 0.57 | 65.22 (5.02%) | 1.99 (84.60%) | 5.32 (93.60%) |
| | 2 | 60.59 ± 0.59 | 63.96 ± 2.04 | 64.39 ± 1.45 | 62.98 (1.42%) | 5.54 (57.12%) | 20.88 (74.89%) |
| MAPU (Ragab et al., 2023b) | - | 60.35 ± 2.15 | 62.48 ± 1.57 | 66.73 ± 0.85 | 63.19 | 12.92 | 83.17 |
| MAPU + SFT | 8 | 63.52 ± 1.24 | 64.77 ± 0.22 | 66.09 ± 0.19 | 64.79 (2.53%) | 0.80 (93.81%) | 1.38 (98.34%) |
| | 4 | 62.21 ± 0.88 | 65.20 ± 1.25 | 67.40 ± 0.59 | 64.94 (2.77%) | 1.99 (84.60%) | 5.32 (93.60%) |
| | 2 | 62.25 ± 1.74 | 66.19 ± 1.02 | 67.85 ± 0.62 | 65.43 (3.54%) | 5.54 (57.12%) | 20.88 (74.89%) |

**Sample-efficiency across SFDA Methods and Datasets.** We assess the sample efficiency of the proposed SFT method by evaluating its predictive performance in the low-data regime. To conduct this analysis, we randomly select 0.5% and 5% of the total available unlabeled target samples in a stratified manner, utilizing fixed seeds for consistency. These sampled subsets serve exclusively for the adaptation process. In Figure 6, we present a comparative analysis of the average F1 score post-adaptation, contrasting SFT with baseline methods (SHOT, NRC, AAD, and MAPU) across

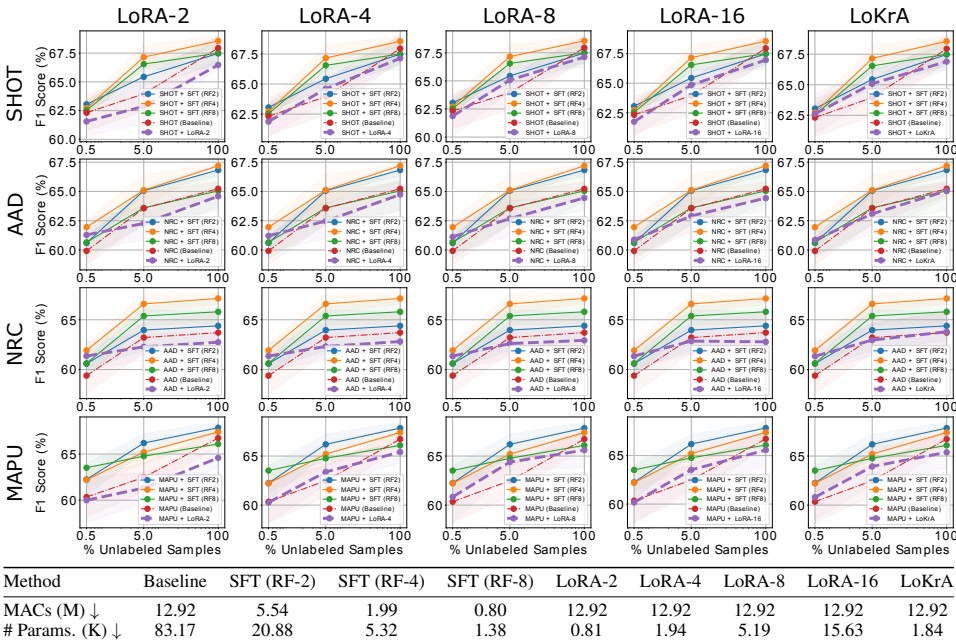

| Method | Baseline | SFT (RF-2) | SFT (RF-4) | SFT (RF-8) | LoRA-2 | LoRA-4 | LoRA-8 | LoRA-16 | LoKrA |
|---|---|---|---|---|---|---|---|---|---|
| MACs (M) ↓ | 12.92 | 5.54 | 1.99 | 0.80 | 12.92 | 12.92 | 12.92 | 12.92 | 12.92 |
| # Params. (K) ↓ | 83.17 | 20.88 | 5.32 | 1.38 | 0.81 | 1.94 | 5.19 | 15.63 | 1.84 |

Figure 7: Comparison of LoRA (Hu et al., 2021) and LoKrA (Edalati et al., 2022; Yeh et al., 2024) (in Purple) against baseline methods (SHOT, NRC, AAD, MAPU) and our proposed approach (SFT) on the SSC dataset, evaluated across varying target sample ratios used during adaptation. The table at the bottom shows the target model's inference overhead after adaptation in terms of MACs and the number of parameters finetuned at the time of adaptation.

multiple datasets: SSC, MFD, HHAR, WISDM, and UCIHAR. Notably, our empirical observations are grounded in the theoretical framework established by Theorem 1, which is encapsulated in Equation (10). While all SFDA methods aim to minimize the empirical loss on the target domain (the first term on the right-hand side of Equation (10)) through their unsupervised objectives (cf. Appendix A.8), the second term—dependent on both the number of samples and the distance between parameters—plays a critical role in generalization. In low-data regimes, as the number of target samples reduces, this second term increases, thereby weakening the generalization bound. As a result, baseline methods exhibit a noticeable drop in predictive performance, distinctly observed for the MFD and UCIHAR datasets (Figure 6), when adapting under the low sample ratio setting. In contrast, SFT effectively mitigates the adverse effects of a small number of samples by implicitly constraining the parameter distance term (the numerator of the second term on the RHS), as empirically observed in Figure 5, therefore, balancing the overall fraction to not loosen the bound on the RHS. This renders SFT significantly more sample-efficient, preserving robust generalization even in low-data regimes. We extend this analysis in Appendix A.9.

**Parameter-efficiency across SFDA Methods and Datasets.** In addition, Table 1 demonstrates consistent improvements in the F1 score alongside enhanced parameter efficiency during the adaptation process, as evidenced by the reduced parameter count (# Params.) of the parameters fine-tuned. As the rank factor ($RF$) increases, the size of the *core subspace* decreases (cf. Equation (8)), leading to fewer parameters that require updates during adaptation. This reduction not only enhances efficiency but also lowers computational costs (MACs) (cf. Equation (9)). The observed performance gains and improved efficiency are consistent across different SFDA methods and datasets, underscoring the robustness, generalizability, and efficiency of our approach across different SFDA methods and benchmark datasets. In Appendix A.9, we extend this analysis to MFD, HHAR, WISDM and UCIHAR datasets and observe similar trends.

**Comparison with Parameter-efficient Tuning Methods.** To further substantiate the robustness of SFT, we benchmark its performance against Parameter-Efficient Fine-Tuning (PEFT) approaches (Han et al., 2024; Xin et al., 2024). PEFT methods primarily aim to match the performance of full model fine-tuning while substantially reducing the number of trainable parameters. Among the most prominent approaches, Low-Rank Adaptation (LoRA) (Hu et al., 2021) has gained significant traction in this domain. In Figure 7, we analyze the performance of LoRA at varying intermediate ranks

Table 2: Ablation studies on finetuning different parameter subspaces during target-side adaptation on MAPU.

| Method | RF | Parameters Finetuned | SSC | | | HHAR | | |
|---|---|---|---|---|---|---|---|---|
| | | | F1 Score (%)↑ | MACs (M)↓ | # Params. (K)↓ | F1 Score (%)↑ | MACs (M)↓ | # Params. (K)↓ |
| No Adaptation | - | None | 54.96 ± 0.87 | 12.92 | 0 | 66.67 ± 1.03 | 9.04 | 0 |
| MAPU | - | Entire Backbone | 66.73 ± 0.85 | 12.92 | 83.17 | 80.32 ± 1.16 | 9.04 | 198.21 |
| MAPU | - | BN | 61.07 ± 1.58 | 12.92 | 0.45 | 71.40 ± 1.63 | 9.04 | 0.64 |
| MAPU + SFT | 8 | Factor Matrices ($\mathbf{V}^{(1)}, \mathbf{V}^{(2)}$) | 66.05 ± 0.75 | 0.8 | 3.33 | 80.42 ± 1.51 | 0.53 | 7.18 |
| | | Core Tensor ($\mathcal{T}$) | 66.09 ± 0.19 | 0.8 | 1.38 | 80.16 ± 2.38 | 0.53 | 3.19 |
| | | Factor Matrices ($\mathbf{V}^{(1)}, \mathbf{V}^{(2)}$) + Core Tensor ($\mathcal{T}$) | 66.17 ± 0.53 | 0.8 | 4.71 | 80.29 ± 1.04 | 0.53 | 10.37 |
| | 4 | Factor Matrices ($\mathbf{V}^{(1)}, \mathbf{V}^{(2)}$) | 67.60 ± 0.07 | 1.99 | 6.66 | 79.88 ± 1.20 | 1.34 | 14.35 |
| | | Core Tensor ($\mathcal{T}$) | 67.40 ± 0.59 | 1.99 | 5.32 | 80.24 ± 1.22 | 1.34 | 12.53 |
| | | Factor Matrices ($\mathbf{V}^{(1)}, \mathbf{V}^{(2)}$) + Core Tensor ($\mathcal{T}$) | 67.82 ± 0.21 | 1.99 | 11.98 | 79.63 ± 1.08 | 1.34 | 26.88 |
| | 2 | Factor Matrices ($\mathbf{V}^{(1)}, \mathbf{V}^{(2)}$) | 67.13 ± 0.69 | 5.54 | 13.31 | 80.51 ± 2.41 | 3.79 | 28.68 |
| | | Core Tensor ($\mathcal{T}$) | 67.85 ± 0.62 | 5.54 | 20.88 | 80.69 ± 2.45 | 3.79 | 49.63 |
| | | Factor Matrices ($\mathbf{V}^{(1)}, \mathbf{V}^{(2)}$) + Core Tensor ($\mathcal{T}$) | 67.37 ± 0.36 | 5.54 | 34.19 | 80.69 ± 1.26 | 3.79 | 78.31 |

($\{2, 4, 8, 16\}$), and Low-Rank Kronecker Adaptation (LoKrA) (Edalati et al., 2022; Yeh et al., 2024) under its most parameter-efficient configuration, alongside SFT, on the SSC dataset (additional results are provided in Appendix Figure 12). Remarkably, SFT consistently outperforms across SFDA tasks, demonstrating superior adaptability and performance. This advantage arises from the structured decomposition applied during source model preparation, which explicitly disentangles the parameter subspace. In contrast, LoRA-style methods introduce a residual branch over the pretrained weights that relies on the fine-tuning objective to uncover the underlying low-rank subspaces (Hu et al., 2021), which results in weaker control over the adaptation process. It is important to highlight that PEFT methods can also be seamlessly integrated with SFT during fine-tuning. For instance, LoRA or LoKrA-style adaptation frameworks can be applied to the core tensor, further enhancing the efficiency of the adaptation process, which we discuss in detail in Appendix A.11. However, while LoRA-style methods can effectively adapt model weights to the target domain, they do not reduce inference overhead, achieving this requires explicit decomposition and reparameterization of the weights.

**Ablation Studies on Fine-tuning different Parameter Subspaces.** In Table 2, we present an ablation study evaluating the impact of fine-tuning different components of the decomposed backbone for MAPU. We compare against baseline methods: (1) fine-tuning the entire backbone and (2) tuning only Batch-Norm (BN) parameters. For our decomposed framework, we assess: 1. Fine-tuning only the factor matrices: Here, we update the factor matrices while keeping the core tensor fixed, adjusting directional transformations in the weight space. 2. Fine-tuning only the core tensor: We freeze the factor matrices and tune only the core tensor, the smallest low-rank subspace, capturing critical multi-dimensional interactions. This is efficient, especially with a high $RF$ values, as the core tensor remains compact, improving performance with fewer parameters and enhancing sample efficiency. 3. Fine-tuning both the core tensor and factor matrices: Both are updated, offering more flexibility but introducing more parameters, risking overfitting. Primarily tuning the core tensor is advantageous due to its compact representation, yielding better generalization, and freezing the factor matrices preserves mode-specific transformations while optimizing key interactions. We observe that core tensor fine-tuning strikes an optimal balance between adaptation and parameter efficiency, delivering strong performance with a minimal computational footprint. Appendix Tables 11, 12, and 13 show the ablation analysis for SHOT, NRC and AAD, respectively.

## 5 CONCLUSION

In this work, we presented a framework for improving the parameter and sample efficiency of SFDA methods in time-series data through a low-rank decomposition of source-pretrained models. By leveraging Tucker-style tensor factorization during the source-model preparation phase, we were able to reparameterize the backbone of the model into a compact subspace. This enabled selective fine-tuning (SFT) of the core tensor on the target side, achieving robust adaptation with significantly fewer parameters. Our empirical results demonstrated that the proposed SFT strategy consistently outperformed baseline SFDA methods across various datasets, especially in resource-constrained and low-data scenarios. Theoretical analysis grounded in PAC-Bayesian generalization bounds provided insights into the regularization effect of SFT, highlighting its ability to mitigate overfitting by constraining the distance between source and target model parameters. Our ablation studies further reinforced the effectiveness of selective finetuning, showing that this approach strikes a balance between adaptation flexibility and parameter efficiency. In Appendix A.13, we discuss the limitations of the presented work. Overall, our contributions positively complement the SFDA methods, offering a framework that can be seamlessly integrated with existing SFDA techniques to improve adaptation efficiency. This lays the groundwork for future research into more resource-efficient and personalized domain adaptation techniques.

## ACKNOWLEDGMENT

We would like to express our sincere gratitude to Sheikh Shams Azam[1] for his invaluable feedback, thoughtful suggestions, and insightful advice throughout the drafting process of this work.

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

# A  Appendix

## A.1  Higher Order Orthogonal Iteration

---

**Algorithm 1:** The higher-order orthogonal iteration (HOOI) algorithm. (De Lathauwer et al., 2000; Kolda & Bader, 2009)

---

**Input:** Tensor $\mathcal{A} \in \mathbb{R}^{I_1 \times I_2 \times \cdots \times I_N}$, Truncation $(R_1, R_2, \ldots, R_N)$, Initial guess
  $\{\mathbf{U}_0^{(n)} : n = 1, 2, \ldots, N\}$
**Output:** Core tensor $\mathcal{G}$, Factor matrices $\{\mathbf{U}_k^{(n)} : n = 1, 2, \ldots, N\}$

$k \leftarrow 0$;
**while** *not converged* **do**
  **for** *all* $n \in \{1, 2, \ldots, N\}$ **do**
    $\mathcal{B} \leftarrow \mathcal{A} \times_1 (\mathbf{U}_{k+1}^{(1)})^\top \times_2 \cdots \times_{n-1} (\mathbf{U}_{k+1}^{(n-1)})^\top \times_{n+1} (\mathbf{U}_k^{(n+1)})^\top \cdots \times_N (\mathbf{U}_k^{(N)})^\top$;
    $\mathbf{B}_{(n)} \leftarrow \mathcal{B}$ in matrix format;
    $\mathbf{U}, \mathbf{\Sigma}, \mathbf{V}^\top \leftarrow$ truncated rank-$R_n$ SVD of $\mathbf{B}_{(n)}$;
    $\mathbf{U}_{k+1}^{(n)} \leftarrow \mathbf{U}$;
    $k \leftarrow k + 1$;

$\mathcal{G} \leftarrow \mathbf{\Sigma}\mathbf{V}^\top$ in tensor format;

---

## A.2  Parameter and Computational Efficiency with Tucker Decomposition

### A.2.1  Tucker Factorization for CNN Weights

Consider a 3D weight tensor $\mathcal{W} \in \mathbb{R}^{C_\text{out} \times C_\text{in} \times K}$ for a 1D convolutional layer, where $C_\text{out}$ and $C_\text{in}$ represent the output and input channels, and $K$ is the kernel size. Using Tucker factorization, $\mathcal{W}$ can be decomposed into a core tensor $\mathcal{T} \in \mathbb{R}^{R_1 \times R_2 \times K}$ and factor matrices $\mathbf{V}^{(1)} \in \mathbb{R}^{C_{out} \times R_1}$, $\mathbf{V}^{(2)} \in \mathbb{R}^{C_{in} \times R_2}$, such that:

$$\mathcal{W} = \mathcal{T} \times_1 \mathbf{V}^{(1)} \times_2 \mathbf{V}^{(2)} \tag{11}$$

**Parameter Efficiency.** The number of parameters before factorization is:

$$\text{Params}_\text{original} = C_\text{out} \times C_\text{in} \times K \tag{12}$$

After 2-Mode Tucker factorization, the number of parameters become:

$$\text{Params}_\text{Tucker} = R_\text{out} \times R_\text{in} \times K + C_\text{out} \times R_\text{out} + C_\text{in} \times R_\text{in} \tag{13}$$

Given that $R_\text{in} \ll C_\text{in}$, and $R_\text{out} \ll C_\text{out}$, the reduction in the number of parameters is significant, leading to a parameter-efficient model.

**Computational Efficiency.** The convolution operation requires $\mathcal{O}(C_\text{out} \times C_\text{in} \times K \times L')$ operations, where $L'$ is the length of the output feature map. After Tucker factorization, the convolutional operations become a sequence of smaller convolutions involving the factor matrices and the core tensor, reducing the computational complexity to:

$$\mathcal{O}(C_\text{in} \times R_\text{in} \times L) + \mathcal{O}(R_\text{out} \times R_\text{in} \times K \times L') + \mathcal{O}(C_\text{out} \times R_\text{out} \times L') \tag{14}$$

where $L$ denotes the length of the input sequence. This reduction depends on the rank $R_\text{in}$ and $R_\text{out}$, leading to lower computational costs compared to the original convolution.

### A.2.2  Tucker Factorization for Fully Connected Weights

Consider a 2D weight matrix $\mathbf{W} \in \mathbb{R}^{M \times N}$ in a fully connected (FC) layer, where $M$ is the input dimension and $N$ is the output dimension.

Using Tucker factorization, $\mathbf{W}$ can be decomposed into a core matrix $\mathbf{G} \in \mathbb{R}^{R_1 \times R_2}$ and factor matrices $\mathbf{U}^{(1)} \in \mathbb{R}^{M \times R_1}$ and $\mathbf{U}^{(2)} \in \mathbb{R}^{N \times R_2}$, such that:

$$\mathbf{W} = \mathbf{U}^{(1)} \mathbf{G} (\mathbf{U}^{(2)})^\top \tag{15}$$

**Parameter Efficiency.** The number of parameters before factorization is:

$$\text{Params}_{\text{original}} = M \times N \tag{16}$$

After Tucker factorization, the number of parameters becomes:

$$\text{Params}_{\text{Tucker}} = R_1 \times R_2 + M \times R_1 + N \times R_2 \tag{17}$$

Again, with $R_1 \ll M$ and $R_2 \ll N$, this leads to a substantial reduction in the number of parameters.

**Computational Efficiency.** The original matrix multiplication requires $\mathcal{O}(M \times N)$ operations.

After Tucker factorization, the computation is broken down into smaller matrix multiplications:

$$\mathcal{O}(M \times R_1) + \mathcal{O}(R_1 \times R_2) + \mathcal{O}(R_2 \times N) \tag{18}$$

This decomposition reduces the computational cost, particularly when $R_1$ and $R_2$ are much smaller than $M$ and $N$.

### A.2.3 COMPUTATION ANALYSIS

**Original Convolutional Layer**: Consider a convolutional layer with the following dimensions:

- Input Channels ($C_{\text{in}}$): 128
- Output Channels ($C_{\text{out}}$): 256
- Kernel Size ($K$): 8
- Input Length ($L$): Length of the input sequence
- Output Length ($L'$): Length of the output feature map

$$\text{Computations}_{\text{original}} = \mathcal{O}(C_{\text{out}} \times C_{\text{in}} \times K \times L') = 256 \times 128 \times 8 \times L' = 262{,}144 \times L' \tag{19}$$

**After Tucker Decomposition:** We apply Tucker decomposition along the channel dimensions (input and output channels) of the weight tensor. The decomposition factorizes the original weight tensor into smaller tensors, introducing two rank parameters:

- Input Rank ($R_{\text{in}}$): Reduced dimension for input channels
- Output Rank ($R_{\text{out}}$): Reduced dimension for output channels

The computational complexity after Tucker decomposition becomes:

$$\text{Computations}_{\text{Tucker}} = \mathcal{O}(C_{\text{in}} \times R_{\text{in}} \times L) + \mathcal{O}(R_{\text{out}} \times R_{\text{in}} \times K \times L') + \mathcal{O}(C_{\text{out}} \times R_{\text{out}} \times L')$$
$$= \text{Input Projection} + \text{Core Convolution} + \text{Output Projection} \tag{20}$$

**Components Computation Explanation:**

- Input Projection: Project the input feature maps from $C_{\text{in}}$ channels to $R_{\text{in}}$ channels:

$$\text{Input Projection} = C_{\text{in}} \times R_{\text{in}} \times L = 128 \times R_{\text{in}} \times L \tag{21}$$

- Core Convolution: Perform convolution using the core tensor of size $R_{\text{out}} \times R_{\text{in}} \times K$:

$$\text{Core Convolution} = R_{\text{out}} \times R_{\text{in}} \times K \times L' = R_{\text{out}} \times R_{\text{in}} \times 8 \times L' \tag{22}$$

- Output Projection: Project the result from $R_{\text{out}}$ channels to $C_{\text{out}}$ output channels:
$$\text{Output Projection} = C_{\text{out}} \times R_{\text{out}} \times L' = 256 \times R_{\text{out}} \times L' \tag{23}$$

**Computational Efficiency Demonstration:** For demonstration purposes, let us assume equal reduced ranks for input and output channels: $R_{\text{in}} = R_{\text{out}} = R$, where $R = 32$

**Computations After Decomposition:**

- Input Projection:
$$\text{Input Projection} = 128 \times 32 \times L = 4{,}096 \times L \tag{24}$$

- Core Convolution:
$$\text{Core Convolution} = 32 \times 32 \times 8 \times L' = 8{,}192 \times L' \tag{25}$$

- Output Projection:
$$\text{Output Projection} = 256 \times 32 \times L' = 8{,}192 \times L' \tag{26}$$

**Total Computations After Decomposition:**
$$\text{Computations}_{\text{decomposed}} = 4{,}096 \times L + 8{,}192 \times L' + 8{,}192 \times L' = 4{,}096 \times L + 16{,}384 \times L' \tag{27}$$

**Comparing Computational Costs:** Assuming the input and output lengths are approximately equal ($L \approx L'$), we can directly compare the total computations.

- Total Computations Before Decomposition:
$$\text{Computations}_{\text{original}} = 262{,}144 \times L' \tag{28}$$

- Total Computations After Decomposition:
$$\text{Computations}_{\text{Tucker}} = 4{,}096 \times L' + 16{,}384 \times L' = 20{,}480 \times L' \tag{29}$$

**Computational Reduction:**
$$\text{Reduction Ratio} = \frac{\text{Computations}_{\text{Tucker}}}{\text{Computations}_{\text{original}}} = \frac{20{,}480 \times L'}{262{,}144 \times L'} \approx 0.078 \tag{30}$$

This indicates a reduction of approximately 92% in computational cost after Tucker decomposition. The reduced ranks $R_{\text{in}}$ and $R_{\text{out}}$ lead to a lower computational complexity compared to the original convolutional layer. The substantial decrease in computations directly results in faster inference times, as the model performs significantly fewer operations.

In sum, Tucker factorization significantly reduces both the number of parameters and the computational cost for 1D CNN and FC weights, making it an effective technique for achieving parameter efficiency and computational efficiency in deep neural networks.

### A.3 FORWARD- AND BACKWARD- PASS EQUIVALENCE WITH TUCKER DECOMPOSITION

In the interest of simplicity and ease of explanation, we demonstrate equivalence by conducting the Tucker decomposition for fully connected weights, *i.e.*, a 2-D matrix, such that, $\mathbf{W} = \mathbf{U}^{(1)}\mathbf{G}(\mathbf{U}^{(2)})^{\top}$.

**Forward Propagation.**
Case 1: Using $\mathbf{W} = \mathbf{U}^{(1)}\mathbf{G}(\mathbf{U}^{(2)})^{\top}$ directly Let the input to the layer be $x$. Forward propagation through $\mathbf{W}$ is represented as:
$$z = \mathbf{W}x = \mathbf{U}^{(1)}\mathbf{G}(\mathbf{U}^{(2)})^{\top}x$$
Case 2: Using $\mathbf{U}^{(1)}, \mathbf{G}, (\mathbf{U}^{(2)})^{\top}$ separately. We compute the forward pass in steps:

1. Compute $y_1 = (\mathbf{U}^{(2)})^{\top}x$,
2. Compute $y_2 = \mathbf{G}y_1 = \mathbf{G}\big((\mathbf{U}^{(2)})^{\top}x\big)$,
3. Compute $z = \mathbf{U}^{(1)}y_2 = \mathbf{U}^{(1)}\big(\mathbf{G}\big((\mathbf{U}^{(2)})^{\top}x\big)\big)$.

By associativity of matrix multiplication, this is equivalent to:
$$z = \mathbf{U}^{(1)}\mathbf{G}(\mathbf{U}^{(2)})^{\top}x = \mathbf{W}x$$
Hence, forward propagation through $\mathbf{U}^{(1)}, \mathbf{G}, (\mathbf{U}^{(2)})^{\top}$ in sequence is equivalent to propagating through $\mathbf{W}$ directly.

**Backward Propagation.** Let the gradient of the loss $\mathcal{L}$ with respect to the output $z$ be $\frac{\partial \mathcal{L}}{\partial z} = g_z$. The task is to propagate this gradient backward.

Case 1: Using $\mathbf{W} = \mathbf{U}^{(1)}\mathbf{G}(\mathbf{U}^{(2)})^\top$ directly. The gradient of the loss with respect to $x$ is:

$$g_x = \mathbf{W}^\top g_z = \left(\mathbf{U}^{(1)}\mathbf{G}(\mathbf{U}^{(2)})^\top\right)^\top g_z = \mathbf{U}^{(2)}\mathbf{G}^\top(\mathbf{U}^{(1)})^\top g_z.$$

Case 2: Using $\mathbf{U}^{(1)}, \mathbf{G}, (\mathbf{U}^{(2)})^\top$ separately.
Backward propagation proceeds as:

1. Compute $g_{y_2} = (\mathbf{U}^{(1)})^\top g_z$,
2. Compute $g_{y_1} = \mathbf{G}^\top g_{y_2} = \mathbf{G}^\top\left((\mathbf{U}^{(1)})^\top g_z\right)$,
3. Compute $g_x = \mathbf{U}^{(2)} g_{y_1} = \mathbf{U}^{(2)}\left(\mathbf{G}^\top\left((\mathbf{U}^{(1)})^\top g_z\right)\right)$.

By the associativity of matrix multiplication, this simplifies to:

$$g_x = \mathbf{U}^{(2)}\mathbf{G}^\top(\mathbf{U}^{(1)})^\top g_z = \mathbf{W}^\top g_z.$$

**Gradients with respect to $\mathbf{U}^{(1)}, \mathbf{G}, (\mathbf{U}^{(2)})^\top$.** The gradients of $\mathcal{L}$ with respect to the parameters are:

1. Gradient with respect to $\mathbf{U}^{(1)}$:

$$\frac{\partial \mathcal{L}}{\partial \mathbf{U}^{(1)}} = g_z y_2^\top = g_z \left(\mathbf{G}\left((\mathbf{U}^{(2)})^\top x\right)\right)^\top$$

2. Gradient with respect to $\mathbf{G}$:

$$\frac{\partial \mathcal{L}}{\partial \mathbf{G}} = g_{y_2} y_1^\top = \left((\mathbf{U}^{(1)})^\top g_z\right)\left((\mathbf{U}^{(2)})^\top x\right)^\top$$

3. Gradient with respect to $(\mathbf{U}^{(2)})^\top$:

$$\frac{\partial \mathcal{L}}{\partial (\mathbf{U}^{(2)})^\top} = g_{y_1} x^\top = \left(\mathbf{G}^\top\left((\mathbf{U}^{(1)})^\top g_z\right)\right) x^\top$$

In sum, the output $z$ is identical whether computed or via $\mathbf{U}^{(1)}, \mathbf{G}, (\mathbf{U}^{(2)})^\top$ or via its composed reconstructed weight $\mathbf{W}$ separately. The gradient $g_x$ also renders identical whether computed via $\mathbf{U}^{(1)}, \mathbf{G}, (\mathbf{U}^{(2)})^\top$ via $\mathbf{W}$ separately. Thus, forward and backward propagation through the decomposed factors $\mathbf{U}^{(1)}, \mathbf{G}, (\mathbf{U}^{(2)})^\top$ is mathematically equivalent to propagating through $\mathbf{W}$ directly.

## A.4 Lemmas

**Lemma 1.** *Let $\mathbf{W}_0 \in \mathbb{R}^{M \times N}$ be the initial weight matrix, and $\mathbf{W}_t \in \mathbb{R}^{M \times N}$ denote the weight matrix after $t$ iterations of gradient descent with a fixed learning rate $\eta > 0$. Suppose the gradient of the loss function $\mathcal{L}(\mathbf{W})$ is bounded, such that for all iterations $k = 0, 1, \ldots, t-1$, we have $\|\nabla \mathcal{L}(\mathbf{W}_k)\|_\infty \leq G$ where $G > 0$ is a constant.*

*Then, the Frobenius norm of the difference between the initial weight matrix $w_0$ and the weight matrix $w_t$ after $t$ iterations is bounded by:*

$$\|\mathbf{W}_t - \mathbf{W}_0\|_F \leq \eta t \sqrt{MN} G.$$

*Proof.* The gradient descent algorithm updates the weight matrix according to the rule:

$$\mathbf{W}_{k+1} = \mathbf{W}_k - \eta \nabla \mathcal{L}(\mathbf{W}_k).$$

Starting from the initial weights $\mathbf{W}_0$, the weights after $t$ iterations are:

$$\mathbf{W}_t = \mathbf{W}_0 - \eta \sum_{k=0}^{t-1} \nabla \mathcal{L}(\mathbf{W}_k).$$

Taking the Frobenius norm of the difference between $w_t$ and $w_0$, we have:

$$\|\mathbf{W}_t - \mathbf{W}_0\|_F = \eta \left\| \sum_{k=0}^{t-1} \nabla \mathcal{L}(\mathbf{W}_k) \right\|_F.$$

Using the triangle inequality for norms:

$$\|\mathbf{W}_t - \mathbf{W}_0\|_F \leq \eta \sum_{k=0}^{t-1} \|\nabla \mathcal{L}(\mathbf{W}_k)\|_F.$$

Since the Frobenius norm $\|\cdot\|_F$ is sub-multiplicative and satisfies $\|\nabla \mathcal{L}(\mathbf{W}_k)\|_F \leq \sqrt{MN}\|\nabla \mathcal{L}(\mathbf{W}_k)\|_\infty$, and by the assumption $\|\nabla \mathcal{L}(\mathbf{W}_k)\|_\infty \leq G$, it follows that:

$$\sum_{k=0}^{t-1} \|\nabla \mathcal{L}(\mathbf{W}_k)\|_F \leq t\sqrt{MN}G.$$

Substituting this into the earlier expression, we obtain the bound:

$$\|\mathbf{W}_t - \mathbf{W}_0\|_F \leq \eta t \sqrt{MN}G.$$

$\square$

**Lemma 2.** *Consider the weight matrices $\mathbf{W}_0$ and $\mathbf{W}_t$ expressed as:*

$$\mathbf{W}_0 = \mathbf{U}_0^{(1)}\mathbf{G}_0(\mathbf{U}_0^{(2)})^\top \quad and \quad \mathbf{W}_t = \mathbf{U}_0\mathbf{G}_t(\mathbf{U}_0^{(2)})^\top,$$

*where $\mathbf{U}_0^{(1)} \in \mathbb{R}^{M \times R_1}$ and $\mathbf{U}_0^{(2)} \in \mathbb{R}^{N \times R_2}$ are fixed matrices, and $\mathbf{G}_0 \in \mathbb{R}^{r_1 \times r_2}$ and $\mathbf{G}_t \in \mathbb{R}^{r_1 \times r_2}$ represent the evolving core matrices. Assume that the entries of $\mathbf{U}_0^{(1)}$ and $\mathbf{U}_0^{(2)}$ are bounded by constants $C_u > 0$ and $C_v > 0$, respectively.*

*Let the Frobenius norm of the difference between $\mathbf{G}_t$ and $\mathbf{G}_0$ after $t$ iterations of gradient descent be bounded as:*

$$\|\mathbf{G}_t - \mathbf{G}_0\|_F \leq \eta t \sqrt{R_1 R_2}G_k,$$

*where $\eta > 0$ is the learning rate, and $G_k > 0$ is a constant.*

*Then, the Frobenius norm of the difference between $\mathbf{W}_t$ and $\mathbf{W}_0$ is bounded by:*

$$\|\mathbf{W}_t - \mathbf{W}_0\|_F \leq R_1 R_2 t \eta \sqrt{MN} \cdot C_u G_k C_v.$$

*Proof.* Given the case where $\mathbf{W}_0 = \mathbf{U}_0^{(1)}\mathbf{G}_0(\mathbf{U}_0^{(2)})^\top$ and $\mathbf{W}_t = \mathbf{U}_0^{(1)}\mathbf{G}_t(\mathbf{U}_0^{(2)})^\top$, with $\mathbf{U}_0^{(1)}$ and $\mathbf{U}_0^{(1)}$ being fixed in both $\mathbf{W}_0$ and $\mathbf{W}_t$. Where $\mathbf{U}_0^{(1)}$ and $\mathbf{U}_0^{(2)}$ are matrices of dimensions $M \times R_1$ and $N \times R_2$, respectively, and $\mathbf{G}_0$ and $\mathbf{G}_t$ are matrices of dimensions $R_1 \times R_2$.

The Frobenius norm of the difference between $\mathbf{W}_t$ and $\mathbf{W}_0$ is:

$$\|\mathbf{W}_t - \mathbf{W}_0\|_F = \|\mathbf{U}_0^{(1)}\mathbf{G}_0(\mathbf{U}_0^{(2)})^\top - \mathbf{U}_0\mathbf{G}_t(\mathbf{U}_0^{(2)})^\top\|_F$$

Since $\mathbf{U}_0^{(1)}$ and $\mathbf{U}_0^{(2)}$ are common in both $\mathbf{W}_0$ and $\mathbf{W}_t$, we can factor them out:

$$\|\mathbf{W}_t - \mathbf{W}_0\|_F = \|\mathbf{U}_0^{(1)}(\mathbf{G}_t - \mathbf{G}_0)(\mathbf{U}_0^{(2)})^\top\|_F$$

Using the sub-multiplicative property of the Frobenius norm:

$$\|\mathbf{U}_0^{(1)}(\mathbf{G}_t - \mathbf{G}_0)(\mathbf{U}_0^{(2)})^\top\|_F \leq \|\mathbf{U}_0^{(1)}\|_F\|\mathbf{G}_t - \mathbf{G}_0\|_F\|\mathbf{U}_0^{(2)}\|_F$$

Since $\mathbf{U}_0^{(1)}$ and $\mathbf{U}_0^{(2)}$ are of dimensions $M \times R_1$ and $N \times R_2$, and $(\mathbf{G}_t - \mathbf{G}_0)$ is $R_1 \times R_2$.

Assuming the maximum entries of $u_0$ and $v_0$ are bounded by constants $C_u$ and $C_v$:

$$\|\mathbf{U}_0^{(1)}\|_F \leq \sqrt{M \cdot R_1} \cdot C_u$$
$$\|\mathbf{G}_t - \mathbf{G}_0\|_F \leq \eta t \cdot \sqrt{R_1 \cdot R_2} \cdot G_k \text{ (from Lemma 1)}$$
$$\|\mathbf{U}_0^{(2)}\|_F \leq \sqrt{N \cdot R_2} \cdot C_v$$

Thus:
$$\|\mathbf{W}_t - \mathbf{W}_0\|_F \leq \sqrt{M \cdot R_1} \cdot C_u \cdot \eta t \cdot \sqrt{R_1 \cdot R_2} \cdot G_k \cdot \sqrt{N \cdot R_2} \cdot C_v$$

This simplifies to:

$$\|\mathbf{W}_t - \mathbf{W}_0\|_F \leq R_1 R_2 t \eta \sqrt{MN} \cdot C_u G_k C_v$$

$\square$

## A.5 PROOF OF THEOREM 1

**Background.** PAC-Bayesian generalization theory offers an appealing method for incorporating data-dependent aspects, like noise robustness and sharpness, into generalization bounds. Recent studies, such as (Bartlett et al., 2017; Neyshabur et al., 2018), have expanded these bounds for deep neural networks to address the mystery of why such models generalize effectively despite possessing more trainable parameters than training samples. Traditionally, the VC dimension of neural networks has been approximated by their number of parameters (Bartlett et al., 2019). While these refined bounds mark a step forward over classical learning theory, questions remain as to whether they are sufficiently tight or non-vacuous. To address this, Dziugaite & Roy (2017) proposed a computational framework that optimizes the PAC-Bayes bound, resulting in a tighter bound and lower test error. Zhou et al. (2019) validated this framework in a large-scale study. More recently, Jiang* et al. (2020) compared different complexity measures and found that PAC-Bayes-based tools align better with empirical results. Furthermore, Li & Zhang (2021) and Wang et al. (2023) utilized this bound to motivate their proposed improved regularization and genralization, respectively. Consequently, the classical PAC-Bayesian framework (McAllester, 1998; 1999) provides generalization guarantees for randomized predictors (McAllester, 2003; Li & Zhang, 2021). In particular, let $f_{\mathbf{W}}$ be any predictor (not necessarily a neural network) learned from the training data and parametrized by $\mathbf{W}$. We consider the distribution $Q$ over parameters of predictors of the form $f_{\mathbf{W}}$, where $\mathbf{W}$ is a random variable whose distribution may also depend on the training data. Given a *prior* distribution $P$ over the set of predictors that is independent of the training data, $S$. The PAC-Bayes theorem states that with probability at least $1 - \delta$ over the draw of the training data, the expected error of $f_{\mathbf{W}}$ can be bounded as follows

$$\mathbb{E}_{\mathbf{W} \sim Q(S)}\left[\mathcal{L}(\mathbf{W})\right] \leq \mathbb{E}_{\mathbf{W} \sim Q(S)}\left[\hat{\mathcal{L}}(\mathbf{W}, S)\right] + C\sqrt{\frac{\text{KL}(Q(S) \parallel P) + k \ln \frac{n}{\delta} + l}{n}}, \quad (31)$$

for some $C, k, l > 0$. Then based on the above described bound we reduce it for our case, where the *prior* distribution on the parameters is centered on source pre-trained weights and the posterior distribution on the target-adapted model and define Theorem 1.

**Theorem 1.** *(PAC-Bayes generalization bound for fine-tuning) Let $\mathbf{W}$ be some hypothesis class (network parameter). Let $P$ be a prior (source) distribution on $\mathbf{W}$ that is independent of the target training set. Let $Q(S)$ be a posterior (target) distribution on $\mathbf{W}$ that depends on the target training set $S$ consisting of $n$ number of samples. Suppose the loss function $\mathcal{L}(.)$ is bounded by $C$. If we set*

*the prior distribution $P = \mathcal{N}(\boldsymbol{W}_{src}, \sigma^2 I)$, where $\boldsymbol{W}_{src}$ are the weights of the pre-trained network. The posterior distribution $Q(S)$ is centered at the fine-tuned model as $\mathcal{N}(\boldsymbol{W}_{trg}, \sigma^2 I)$. Then with probability $1 - \delta$ over the randomness of the training set, the following holds:*

$$\mathbb{E}_{\boldsymbol{W} \sim Q(S)} [\mathcal{L}(\boldsymbol{W})] \leq \mathbb{E}_{\boldsymbol{W} \sim Q(S)} \left[ \hat{\mathcal{L}}(\boldsymbol{W}, S) \right] + C \sqrt{\frac{\sum_{i=1}^{D} \|\boldsymbol{\mathcal{W}}_{trg}^{(i)} - \boldsymbol{\mathcal{W}}_{src}^{(i)}\|_F^2}{2\sigma^2 n} + \frac{k \ln \frac{n}{\delta} + l}{n}}. \quad (32)$$

*for some $\delta, k, l > 0$, where $\boldsymbol{\mathcal{W}}_{trg}^{(i)} \in \boldsymbol{W}_{trg}$, and $\boldsymbol{\mathcal{W}}_{src}^{(i)} \in \boldsymbol{W}_{src}$, $\forall 1 \leq i \leq D$, $D$ denoting the total number of layers.*

It is important to note that the original formulation in (McAllester, 1999) assumes the loss function is restricted to values between 0 and 1. In contrast, the modified version discussed here extends the applicability to loss functions that are bounded between 0 and some positive constant $C$. This adjustment is made by rescaling the loss function by a factor of $\frac{1}{C}$, which introduces the constant $C$ in the right-hand side of Equation (31).

*Proof.* We expand the definition using the density of multivariate normal distributions.

$$\text{KL}(Q(S) \parallel P) = \mathbb{E}_{\boldsymbol{W} \sim Q(S)} \left[ \log \left( \frac{\Pr(\boldsymbol{W} \sim Q(S))}{\Pr(\boldsymbol{W} \sim P)} \right) \right]$$

Substituting the densities of the Gaussian distributions, this can be written as:

$$\text{KL}(Q(S) \parallel P) = \mathbb{E}_{\boldsymbol{W} \sim Q(S)} \left[ \log \frac{\exp \left( -\frac{1}{2\sigma^2} \|\boldsymbol{W} - \boldsymbol{W}_{trg}\|^2 \right)}{\exp \left( -\frac{1}{2\sigma^2} \|\boldsymbol{W} - \boldsymbol{W}_{src}\|^2 \right)} \right]. \quad (33)$$

This simplifies to:

$$\text{KL}(Q(S) \parallel P) = \frac{1}{2\sigma^2} \mathbb{E}_{\boldsymbol{W} \sim Q(S)} \left[ \|\boldsymbol{W} - \boldsymbol{W}_{src}\|^2 - \|\boldsymbol{W} - \boldsymbol{W}_{trg}\|^2 \right]. \quad (34)$$

Expanding the squared terms:

$$\text{KL}(Q(S) \parallel P) = \frac{1}{2\sigma^2} \mathbb{E}_{\boldsymbol{W} \sim Q(S)} \left[ \|\boldsymbol{W}_{trg} - \boldsymbol{W}_{src}\|^2 + 2 \langle \boldsymbol{W} - \boldsymbol{W}_{trg}, \boldsymbol{W}_{trg} - \boldsymbol{W}_{src} \rangle \right]. \quad (35)$$

Since the expectation $\mathbb{E}_{\boldsymbol{W} \sim Q(S)} [\boldsymbol{W} - \boldsymbol{W}_{trg}] = 0$ (because $\boldsymbol{W}$ is distributed around $\boldsymbol{W}_{trg}$), the cross-term vanishes:

$$\implies \text{KL}(Q(S) \parallel P) = \frac{1}{2\sigma^2} \|\boldsymbol{W}_{trg} - \boldsymbol{W}_{src}\|_F^2. \quad (36)$$

$$\implies \text{KL}(Q(S) \parallel P) = \frac{1}{2\sigma^2} \|\boldsymbol{W}_{trg} - \boldsymbol{W}_{src}\|_F^2 \leq \frac{\sum_{i=1}^{D} \|\boldsymbol{\mathcal{W}}_{trg}^{(i)} - \boldsymbol{\mathcal{W}}_{src}^{(i)}\|_F^2}{2\sigma^2}. \quad (37)$$

where , $\boldsymbol{\mathcal{W}}_{trg}^{(i)} \in \boldsymbol{W}_{trg}$, and $\boldsymbol{\mathcal{W}}_{src}^{(i)} \in \boldsymbol{W}_{src}$, $\forall 1 \leq i \leq D$, $D$ denoting the total number of layers. Then, we can obtain Equation (32) by substituting Equation (37) in Equation (31). $\square$

We would also like to emphasize that our primary goal is not to introduce a novel theorem but to utilize established PAC-Bayesian bounds as a theoretical framework to explain our empirical observations of implicit regularization and sample efficiency (*cf*. Figure 2B)). The PAC-Bayesian analysis serves as a tool to provide theoretical insights into why our strategy exhibits improved performance under sample scarce scenario (discussed in Section 4) in the SFDA setting. By grounding our findings in existing theoretical work, we aim to bridge the gap between empirical results and theoretical understanding in this specific context.

Table 3: Dataset Summary (Ragab et al., 2023a).

| Dataset | # Users/Domains | # Channels | # Classes | Sequence Length | Training Set | Testing Set |
|---|---|---|---|---|---|---|
| UCIHAR | 30 | 9 | 6 | 128 | 2300 | 990 |
| WISDM | 36 | 3 | 6 | 128 | 1350 | 720 |
| HHAR | 9 | 3 | 6 | 128 | 12716 | 5218 |
| SSC | 20 | 1 | 5 | 3000 | 14280 | 6130 |
| MFD | 4 | 1 | 3 | 5120 | 7312 | 3604 |

## A.6 DATASET DESCRIPTION

We utilize the benchmark datasets provided by AdaTime (Ragab et al., 2023a). These datasets exhibit diverse attributes such as varying complexity, sensor types, sample sizes, class distributions, and degrees of domain shift, allowing for a comprehensive evaluation across multiple factors.

Table 3 outlines the specific details of each dataset, including the number of domains, sensor channels, class categories, sample lengths, and the total sample count for both training and testing sets. A detailed description of the selected datasets is provided below:

- **UCIHAR** (Anguita et al., 2013): The UCIHAR dataset consists of data collected from three types of sensors—accelerometer, gyroscope, and body sensors—used on 30 different subjects. Each subject participated in six distinct activities: walking, walking upstairs, walking downstairs, standing, sitting, and lying down. Given the variability across subjects, each individual is considered a separate domain. From the numerous possible cross-domain combinations, we selected the ten scenarios set by Ragab et al. (2023a).

- **WISDM** (Kwapisz et al., 2011): The WISDM dataset uses accelerometer sensors to gather data from 36 subjects engaged in the same activities as those in the UCIHAR dataset. However, this dataset presents additional challenges due to class imbalance among different subjects. Specifically, some subjectsonly contribute samples from a limited set of the overall activity classes. As with the UCIHAR dataset, each subject is treated as an individual domain, and ten cross-domain scenarios set by Ragab et al. (2023a) are used for evaluation.

- **HHAR** (Stisen et al., 2015): The Heterogeneity Human Activity Recognition (HHAR) dataset was collected from 9 subjects using sensor data from both smartphones and smartwatches. Ragab et al. (2023a) standardized the use of a single device, specifically a Samsung smartphone, across all subjects to minimize variability. Each subject is treated as an independent domain, and a total of 10 cross-domain scenarios are created by randomly selecting subjects.

- **SSC** (Goldberger et al., 2000): The sleep stage classification (SSC) task focuses on categorizing electroencephalography (EEG) signals into five distinct stages: Wake (W), Non-Rapid Eye Movement stages (N1, N2, N3), and Rapid Eye Movement (REM). This dataset is derived the Sleep-EDF dataset, which provides EEG recordings from 20 healthy individuals. Consistent with prior research (Ragab et al., 2023a), we select a single EEG channel (Fpz-Cz) and ten cross-domain scenarios for evaluation.

- **MFD** (Lessmeier et al., 2016): The Machine Fault Diagnosis (MFD) dataset has been collected by Paderborn University to identify various types of incipient faults using vibration signals. The data was collected under four different operating conditions, and in our experiments, each of these conditions was treated as a separate domain. We used twelve cross-condition scenarios to evaluate the domain adaptation performance. Each sample in the dataset consists of a single univariate channel.

## A.7 SFDA METHODS

- **SHOT** (Liang et al., 2020; 2021): SHOT optimizes mutual information by minimizing conditional entropy $H(Y|X)$ to enforce unambiguous cluster assignments, while maximizing marginal entropy $H(Y)$ to ensure uniform cluster sizes, thereby preventing degeneracy.

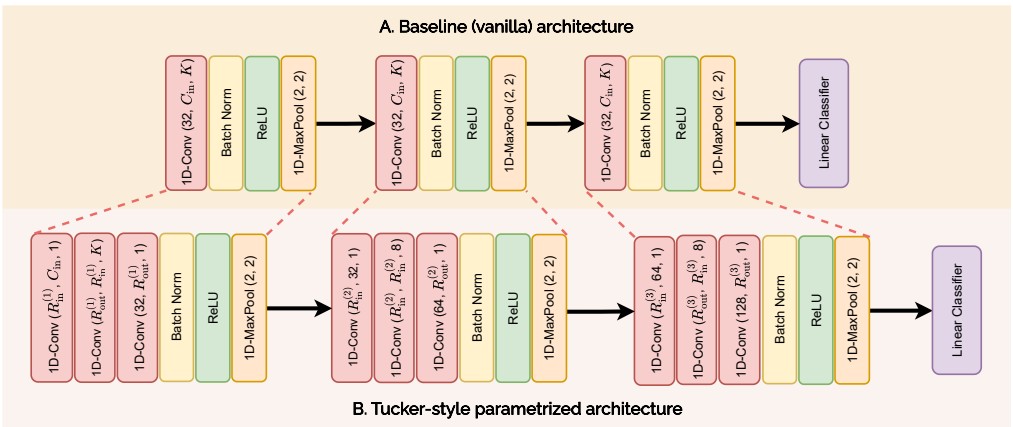

Figure 8: **A.** Baseline architecture (Ragab et al., 2023a;b) **B.** Architecture reprametrization after Tucker-style factorization of convolution weights. $C_{\text{in}}$ and $K$ denote the number of input channels and the filter size, respectively. $R_{\text{out}}^{(1)}$ and $R_{\text{in}}^{(i)}$ and $R_{\text{out}}^{(i)}$, denote the mode ranks for the $i^{th}$ layer.

- **NRC** (Yang et al., 2021a; 2023): NRC leverages neighborhood clustering, with an objective function comprising two main components: a neighborhood clustering term for prediction consistency and a marginal entropy term $H(Y)$ to promote prediction diversity.
- **AAD** (Yang et al., 2022): AAD also utilizes neighborhood clustering but incorporates a contrastive objective similar to InfoNCE (Oord et al., 2018), which attracts predictions of nearby features in the feature space while dispersing (repelling) those that are farther apart.
- **MAPU** (Ragab et al., 2023b; Gong et al., 2024): Building on the concepts from Liang et al. (2021) and Kundu et al. (2022a), and focusing on time-series context, MAPU introduces masked reconstruction as an auxiliary task to enhance SFDA performance.

## A.8 SFDA TARGET ADAPTATION OBJECTIVES

**SHOT.** The overall objective is given as follows:

$$\mathcal{L}_{\text{SHOT}}(g_t) = \mathcal{L}_{\text{ent}}(f_t; \mathcal{X}_t) + \mathcal{L}_{\text{div}}(f_t; \mathcal{X}_t) - \beta \mathbb{E}_{(x_t,\hat{y}_t) \in \mathcal{X}_t \times \hat{\mathcal{Y}}_t} \sum_{k=1}^{K} \mathbf{1}_{[k=\hat{y}_t]} \log \delta_k(f_t(x_t))), \quad (38)$$

where,

$$\mathcal{L}_{\text{ent}}(f_t; \mathcal{X}_t) = -\mathbb{E}_{x_t \in \mathcal{X}_t} \sum_{k=1}^{K} \delta_k(f_t(x_t)) \log \delta_k(f_t(x_t)), \quad (39)$$

$$\mathcal{L}_{\text{div}}(f_t; \mathcal{X}_t) = \sum_{k=1}^{K} \hat{p}_k \log \hat{p}_k = D_{\text{KL}}\left(\hat{p}, \frac{1}{K}\mathbf{1}_K\right) - \log K, \quad (40)$$

in the above equation $D_{\text{KL}}$ stands for the KL divergence. $f_t(x) = h_t(g_t(x))$ is the $K$-dimensional output of each target sample, $\mathbf{1}_K$ is a $K$-dimensional vector with all ones, and $\hat{p} = \mathbb{E}_{x_t \in \mathcal{X}_t}[\delta(\hat{f}_t(x_t))]$ is the mean output embedding of the whole target domain. Recall that $\delta_k(a) = \frac{\exp(a_k)}{\sum_i \exp(a_i)}$ represents the $k$-th element of the softmax output.

Moreover, the $\hat{y}_t \in \hat{\mathcal{Y}}_t$ denotes the pseudo label on for the target samples, based on DeepCluster (Caron et al., 2018), derived as follows. The centroid for each class in the target domain is obtained, similar to weighted k-means clustering:

$$c_k^{(0)} = \frac{\sum_{x_t \in \mathcal{X}_t} \delta_k(\hat{f}_t(x_t)) \, \hat{g}_t(x_t)}{\sum_{x_t \in \mathcal{X}_t} \delta_k(\hat{f}_t(x_t))}, \quad (41)$$

where $\hat{f}_t = \hat{g}_t \circ h_t$ denotes the previously learned target hypothesis. These centroids robustly and reliably characterize the distribution of different categories within the target domain. Then, pseudo labels are obtained via the nearest centroid classifier:

$$\hat{y}_t = \arg\min_k D_f(\hat{g}_t(x_t), c_k^{(0)}), \tag{42}$$

where $D_f(a, b)$ measures the cosine distance between $a$ and $b$. Then, the target centroids based on the new pseudo labels are computed:

$$c_k^{(1)} = \frac{\sum_{x_t \in \mathcal{X}_t} \mathbf{1}(\hat{y}_t = k) \, \hat{g}_t(x_t)}{\sum_{x_t \in \mathcal{X}_t} \mathbf{1}(\hat{y}_t = k)}, \tag{43}$$

$$\hat{y}_t = \arg\min_k D_f(\hat{g}_t(x_t), c_k^{(1)}). \tag{44}$$

$\hat{y}_t$ denotes the self-supervised pseudo-labels, since they are generated by the centroids obtained in an unsupervised manner. In practice, the centroids and labels are updated for multiple rounds.

**NRC.** The NRC adaptation objective leverages reciprocal neighborhood clustering to guide learning through a combination of losses:

$$\mathcal{L}_{\text{NRC}} = \mathcal{L}_{\text{div}} + \mathcal{L}_N + \mathcal{L}_E + \mathcal{L}_{\text{self}}. \tag{45}$$

The approach maintains two memory banks: one for feature representations ($\mathcal{F} = \{z_1, z_2, \ldots, z_n\}$) and another for prediction scores ($\mathcal{S} = \{p_1, p_2, \ldots, p_n\}$), where $z_i$ and $p_i$ denote the feature and the probability score of the $i$-th sample, respectively. These banks are updated per mini-batch by replacing outdated entries with newly computed values, ensuring efficient and consistent updates. Nearest neighbors are identified using cosine similarity, and their predictions, weighted by affinity values, provide a supervision signal:

$$\mathcal{L}_N = -\frac{1}{n_t} \sum_i \sum_{k \in \mathcal{N}_i^K} A_{i,k} S_k^\top p_i, \tag{46}$$

where $A_{i,k}$ reflects the affinity between a sample $z_i$ and its $k$-th neighbor, $S_k$ is the stored prediction, and $\mathcal{N}_i^K$ denotes the $K$-nearest neighbors of $z_i$. Neighbors are further classified into reciprocal (RNN) and non-reciprocal (nRNN) based on mutual membership in each other's nearest-neighbor sets. Reciprocal neighbors, being more reliable, are assigned higher affinity values:

$$A_{i,j} = \begin{cases} 1, & \text{if } j \in \mathcal{N}_i^K \wedge i \in \mathcal{N}_j^M, \\ \frac{1}{r}, & \text{otherwise.} \end{cases} \tag{47}$$

This selective weighting ensures that supervision focuses on semantically similar neighbors. To enhance learning, a self-regularization loss aligns a sample's current prediction with its stored memory:

$$\mathcal{L}_{\text{self}} = -\frac{1}{n_t} \sum_i S_i^\top p_i. \tag{48}$$

Additionally, expanded neighbors, defined as the $M$-nearest neighbors of $K$-nearest neighbors, provide supplementary supervision with reduced affinities to account for potential noise:

$$\mathcal{L}_E = -\frac{1}{n_t} \sum_i \sum_{k \in \mathcal{N}_i^K} \sum_{m \in E_M^k} r S^\top p_i. \tag{49}$$

Finally, the diversity loss $\mathcal{L}_{\text{div}}$ prevents degenerate solutions by encouraging predictions to span diverse clusters. Together, these components enable robust adaptation by combining reliable neighbor supervision, self-regularization, and expanded relationships.

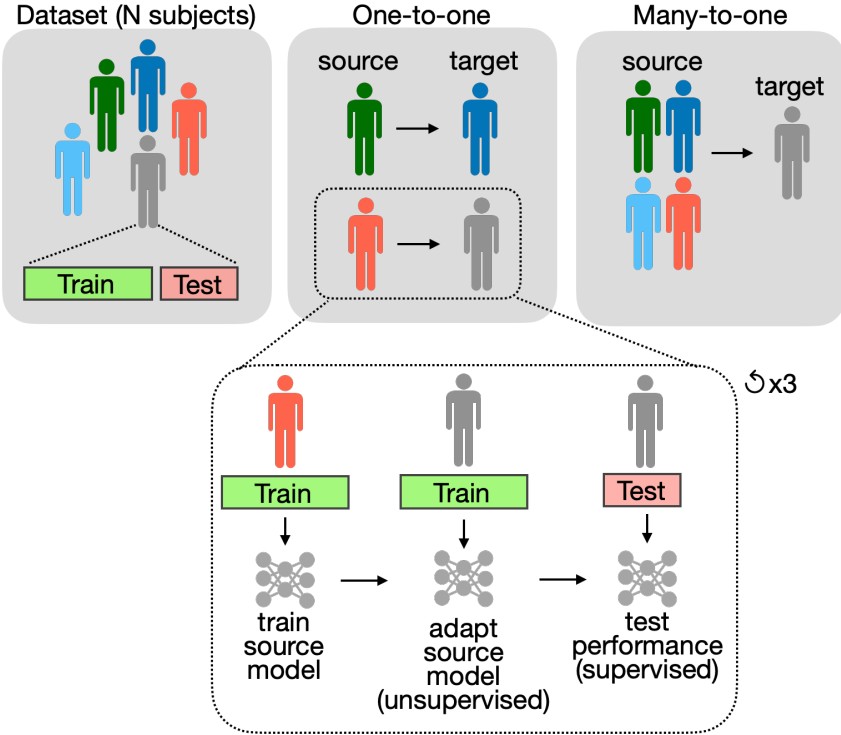

Figure 9: A visual representation of the experimental setup used for evaluating the Source-Free Domain Adaptation (SFDA) frameworks.

**AAD.** The AAD objective is pretty straingforward as follows

$$\mathcal{L}_{\text{AAD}} = \mathbb{E}[L_i(C_i, B_i)], \text{ with } L_i(C_i, B_i) = -\sum_{j \in C_i} p_i^T p_j + \lambda \sum_{m \in B_i} p_i^T p_m \tag{5}$$

Note the gradient will come from both $p_i$ and $p_m$. For this objective, there existstwo sets for each feature $z_i$: close neighbor set $C_i$ containing $K$-nearest neighbors of $z_i$ (with distances as cosine similarity), and background set $B_i$, which contains the features that are not in $C_i$ (features potentially from different classes). To retrieve nearest neighbors for training, two memory banks are maintained to store all *target* features along with their predictions similar to NRC (Yang et al., 2021a; 2023), which is efficient in both memory and computation, since only the features along with their predictions computed in each mini-batch are used to update the memory bank.

**MAPU.** In the MAPU framework, the SHOT objective serves as the primary adaptation objective. To further enhance the adaptation process, an auxiliary objective based on masked reconstruction is used, defined as follows:

$$\mathcal{L}_{\text{MAPU}}(g_t) = \mathcal{L}_{\text{SHOT}}(g_t) + \mathcal{L}_{\text{recon}}(g_t), \tag{50}$$

where the reconstruction loss, $\mathcal{L}_{\text{recon}}$, is given by:

$$\mathcal{L}_{\text{recon}}(g_t) = \mathbb{E}_{x_t \in \mathcal{X}_s} \|h_t(x_t) - j_s(h_t(\hat{x}_t))\|_2^2, \quad \text{with } \hat{x}_t = \texttt{MASK}(x_t). \tag{51}$$

Here, $\texttt{MASK}(\cdot)$ represents a masking function applied to the input $x_t$, and $j_s$ is the imputer module. Importantly, during the adaptation process, the imputer module $j_s$ remains frozen. This auxiliary reconstruction objective guides the target encoder $h_t$ to preserve meaningful feature representations while adapting to the target domain, ensuring alignment with the source representations reconstructed by $j_s$.

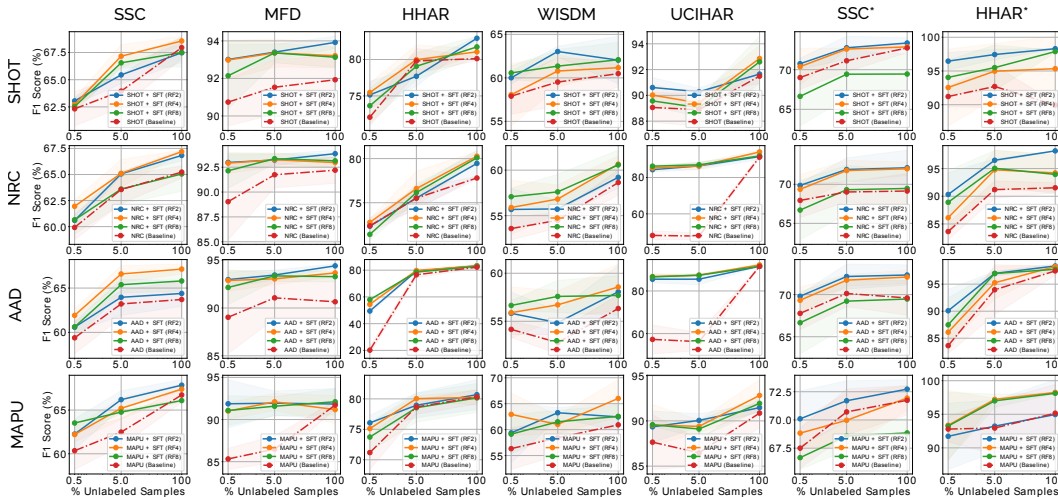

Figure 10: Same experimental analysis as done in Figure 6 with an additional many-to-one SFDA adaptation experiment on the SSC (SSC*) and the HHAR (HHAR*), marked with an asterisk (*).

## A.9 Training Details, Experimental Setup, and Extended Analysis

For all datasets, we utilize a simple 3-layer 1D-CNN backbone following (Ragab et al., 2023b), which has shown superior performance compared to more complex architectures (Donghao & Xue, 2024; Cheng et al., 2024). Figure 8 illustrates both the baseline (vanilla) architecture and the proposed Tucker factorized architecture obtained after reparametrized the weights. The filter size $(K)$ for the input layers varies across datasets to account for differences in sequence lengths. Specifically, we set the filter sizes to 25 for SSC, 32 for MFD, 5 for HHAR, 5 for WISDM, and 5 for UCIHAR, following (Ragab et al., 2023a).

**Source Pretraining.**   Following Liang et al. (2020; 2021), we pre-train a deep neural network source model $f_s : \mathcal{X}_s \to \mathcal{C}_s$, where $f_s = g_s \circ h_s$, with $h_s$ as the backbone and $g_s$ as the classifier. The model is trained by minimizing the standard cross-entropy loss:

$$\mathcal{L}_{\text{src}}(f_s) = -\mathbb{E}_{(x_s, y_s) \in \mathcal{X}_s \times \mathcal{C}_g} \sum_{k=1}^{K} q_k \log \delta_k(f_s(x_s)), \tag{52}$$

where $\delta_k(a) = \frac{\exp(a_k)}{\sum_i \exp(a_i)}$ represents the $k$-th element of the softmax output, and $q$ is the one-hot encoding of the label $y_s$. To enhance the model's discriminability, we incorporate label smoothing as described in Müller et al. (2019). The loss function with label smoothing is:

$$\mathcal{L}_{\text{src}}^{\text{ls}}(f_s) = -\mathbb{E}_{(x_s, y_s) \in \mathcal{X}_s \times \mathcal{C}_g} \sum_{k=1}^{K} q_k^{ls} \log \delta_k(f_s(x_s)), \tag{53}$$

where $q_k^{ls} = (1 - \alpha)q_k + \alpha/K$ represents the smoothed label, with the smoothing parameter $\alpha$ set to 0.1.

Additionally, MAPU (Ragab et al., 2023b) introduces an auxiliary objective optimized alongside the cross-entropy loss, namely the imputation task loss. This auxiliary task is performed by an imputer network $j_s$, which takes the output features of the masked input from the backbone and maps them to the output features of the non-masked input. The imputer network minimizes the following loss:

$$\mathcal{L}_{\text{recon}}(j_s) = \mathbb{E}_{(x_s, y_s) \in \mathcal{X}_s \times \mathcal{C}_g} \|h_s(x_s) - j_s(h_s(\hat{x}_s))\|_2^2, \quad \text{where } \hat{x}_s = \texttt{MASK}(x_s). \tag{54}$$

The backbone and classifier weights are optimized using the Adam optimizer (Kingma, 2014) with a learning rate of 1e-3. We follow Ragab et al. (2023b) for setting all other hyperparameters.

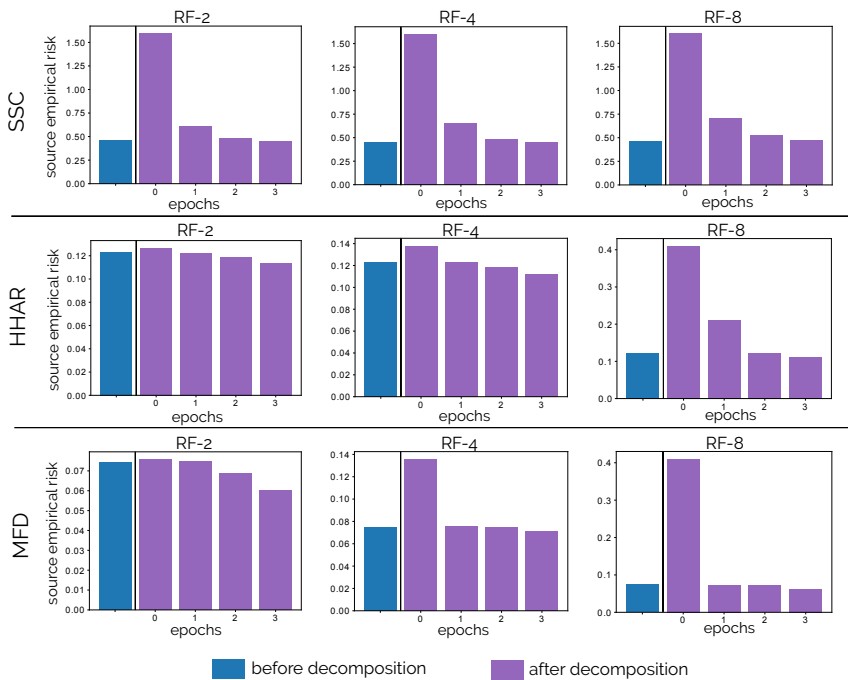

Figure 11: Evolution of source empirical risk after backbone decomposition, showing the network trained on the source domain as it transitions from the freshly reparameterized state (epoch 0) to progressively recover its original source empirical risk.

**Target Adaptation.** For target adaptation, we apply the objective functions and training strategies specific to each SFDA method (detailed in Appendix A.7 and A.8). The backbone weights are optimized to adapt to the target distribution, with Adam (Kingma, 2014) used as the optimizer to learn the target-adapted weights. We experiment with a range of learning rates: {5e-4, 1e-4, 5e-5, 1e-5, 5e-6, 1e-6, 5e-7, 1e-7} for each method (including the baseline) and report the best performance achieved.

**One-to-One Analysis.** In a typical SFDA evaluation setting, a source model is pre-trained using labeled data from one domain (the source domain) and subsequently adapted to another domain (the target domain), where it is evaluated using unlabeled target data. This evaluation strategy, known as **One-to-one** evaluation, involves a single source domain for pretraining and a single target domain for adaptation. In this paper, we use multiple source-target pairs derived from the datasets introduced in (Ragab et al., 2023a;b). Each source-target pair is evaluated separately, and the average performance across all pairs is reported to assess the effectiveness of the employed SFDA methods.

**Many-to-one Analysis.** To provide a more comprehensive and realistic evaluation, we also conduct an experiment under the many-to-one setting. In this setting, data from all source domains in the source-target pairs are merged to form a single, larger source domain for pretraining. The pretrained model is then adapted to target domains that were not part of the combined source domain. This many-to-one approach allows for much robust evaluation. Figure 9 visually describes the overall evaluation setup. In Figure 10, we extend our analysis from Figure 6 to include results for the Many-to-one setup of the SSC (Goldberger et al., 2000) and HHAR (Stisen et al., 2015) datasets, results for the Many-to-one setting are marked with an asterisk (*).

**Comparison with Parameter-Efficient Tuning Methods (Extended Analysis).** In Table 4, we extend the results from Table 1 by including the outcomes for the WISDM and UCIHAR datasets. Consistent improvements are observed across various sample ratios, along with a significant reduction in both inference overhead and the number of parameters fine-tuned during adaptation. Additionally, we provide the F1 scores for each source-target pair across all the discussed datasets in Tables 5-9, for a direct comparison with the number reported by Ragab et al. (2023b).

A.10 COMBINING WITH PARAMETER-EFFICIENT METHODS (EXTENDED ANALYSIS)

Our extended analysis investigates a modified LoRA-style PEFT method, where we freeze the randomly initialized factor and fine-tune only the zero initialized factor; freezing the zero-initialized factor is infeasible as it results in learning nothing due to its multiplicative zero effect. To evaluate its performance against the standard LoRA adaptation and our Source-Free Target (SFT) strategy at different $RF$ values. However, we observe notable underperformance compared to standard LoRA adaptation and our proposed method, as shown in Figures 13, 14, and 15. The frozen factor, which is randomly initialized, imposes arbitrary constraints on the optimization of the fine-tuned factor, severely restricting its ability to explore the target domain effectively. This leads to a suboptimal discovery of the ideal subspace for adaptation and results in significant underfitting.

A.11 COMBINING SFT WITH LoRA-STYLE PEFT METHODS.

To evaluate the hybrid integration of SFT with LoRA-style PEFT methods, we conducted an in-depth analysis, applying a range of SFDA approaches—namely SHOT, NRC, AAD, and MAPU—across the SSC, HHAR, and MFD datasets in the One-to-one and Many-to-one settings (*cf*. Appendix A.9), as depicted in Figures 16-20. This evaluation explores the efficacy of combining SFT with LoRA-like frameworks under different domain adaptation scenarios, offering insights into the trade-offs between performance and efficiency across a diverse set of tasks. Additionally, Table 10 presents a detailed comparison of parameter efficiency and inference overhead for all hybrid combinations tested.

Our results indicate that the parameter efficiency of SFT can be further improved by incorporating LoRA-style PEFTs during the fine-tuning stage. However, we observed a slight degradation in predictive performance compared to the vanilla SFT. This decline can likely be attributed to the compounded effect of low-rank approximations: the initial rank reduction from source model decomposition, followed by an additional low-rank fine-tuning for target-adaptation, excessively constrains the model's learning capacity. Nevertheless, combining SFT with LoRA-style adaptation still outperforms using LoRA alone in terms of predictive performance, while also achieving superior parameter efficiency, demonstrating the advantage of source model decomposition.

A.12 EXTENDED RELATED WORKS, OBSERVATIONS AND MOTIVATION

**Unsupervised Domain Adaptation.** Unsupervised Domain Adaptation (UDA) for time-series data addresses the fundamental challenge posed by distributional discrepancies between source and target domains, which can lead to significant performance degradation when models are deployed in new environments. Discrepancy-based methods (Cai et al., 2021; Liu & Xue, 2021; He et al., 2023) aim to align feature representations by minimizing statistical divergences—such as Maximum Mean Discrepancy (MMD) or Kullback-Leibler divergence—between the source and target feature distributions. Adversarial approaches (Wilson et al., 2020; 2023; Ragab et al., 2022; Jin et al., 2022; Ozyurt et al., 2023) employ adversarial training frameworks where a discriminator is trained to distinguish between source and target features, and the feature extractor is trained to produce representations that the discriminator cannot differentiate, thereby promoting domain-invariant features. The comprehensive survey by Ragab et al. (2023a) provides an extensive overview of these domain adaptation techniques specific to time-series data. However, conventional domain adaptation methods generally assume access to both source and target domain data during adaptation—an assumption that is often impractical in real-world scenarios. Source data may be inaccessible due to privacy concerns, confidentiality agreements, or intellectual property restrictions (Kundu et al., 2020; 2021). Moreover, transmitting and storing large-scale source datasets on resource-constrained devices, such as IoT devices or mobile platforms, may be infeasible due to limited computational resources and storage capacity (Liu et al., 2023).

**Source-Free Domain Adaptation.** To address the limitations associated with source data accessibility, Source-Free Domain Adaptation (SFDA) has emerged as a compelling alternative. SFDA operates under the assumption that only a pre-trained source model is available for adaptation to the target domain, thereby eliminating the need for access to source data (Liang et al., 2020; 2021; Li

et al., 2020; Yang et al., 2021a; 2022; 2023; Kim et al., 2021; Xia et al., 2021; Ding et al., 2022; Litrico et al., 2023; Tang et al., 2024b;a). This paradigm has garnered significant attention in computer vision tasks, including image classification (Kundu et al., 2020; Liang et al., 2020), semantic segmentation (Liu et al., 2021; Bateson et al., 2022), and object detection (Saltori et al., 2020; Xiong et al., 2022). In SFDA, prominent strategies involve leveraging unsupervised clustering techniques to refine feature representations. These methods promote the *discriminability* and *diversity* of the feature space through various objectives, such as information maximization (Liang et al., 2020), which encourages the model to produce confident and diverse predictions; neighborhood clustering (Yang et al., 2021a), which clusters target samples based on feature similarity; and contrastive learning objectives (Yang et al., 2022), which enhance representation learning by contrasting positive and negative sample pairs. Recent advancements incorporate auxiliary self-supervised tasks, such as masking and imputation, to improve model adaptation, as demonstrated by Ragab et al. (2023b) and Gong et al. (2024). These approaches build on earlier works (Liang et al., 2021; Kundu et al., 2022a) that integrate auxiliary objectives to capture intrinsic data structures and enhance representation learning. Furthermore, the utilization of pre-trained foundation models (Radford et al., 2021; Jia et al., 2021) has been explored to guide adaptation by leveraging the extensive knowledge embedded in models trained on large-scale datasets (Tang et al., 2024b). However, the applicability of such models to specialized domains like healthcare and agriculture remains limited due to domain mismatch. Foundation models may fail to capture the specific characteristics inherent in niche datasets, making the adaptation process inefficient, particularly when a substantial amount of inference is required through the large foundation model to represent target samples. While foundational models have recently been developed for time-series data (Gao et al., 2024; Ye et al., 2024), their applicability to SFDA is yet to be verified. Notably, the exploration of SFDA in time-series contexts, especially concerning parameter efficiency and sample efficiency, remains largely unexplored (Ragab et al., 2023a; Liu et al., 2024; Gong et al., 2024; 2025; Furqon et al., 2025). In this work, we aim to address these limitations by proposing a strategy to conduct SFDA efficiently in the context of time-series data utilizing the established framework by Ragab et al. (2023b). We demonstrate that our approach achieves both parameter efficiency and sample efficiency, providing a unified solution for these challenges.

**Parameter Redundancy and Low-Rank Subspaces.** Neural network pruning has demonstrated that substantial reductions in parameters, often exceeding 90%, can be achieved with minimal accuracy loss, revealing significant redundancy in trained deep networks (LeCun et al., 1989; Hassibi & Stork, 1992; Li et al., 2017; Frankle & Carbin, 2018; Sharma et al., 2024). Structured pruning methods, such as those proposed by Molchanov et al. (2017) and Hoefler et al. (2021), have optimized these reductions to maintain or even improve inference speeds. Low-rank models have played a crucial role in pruning, with techniques such as Singular Value Decomposition (SVD) (Eckart & Young, 1936) applied to fully connected layers (Denil et al., 2013) and tensor-train decomposition used to compress neural networks (Novikov et al., 2015). Methods developed by Jaderberg et al. (2014), Denton et al. (2014), Tai et al. (2016), and Lebedev et al. (2015) accelerate Convolutional Neural Networks (CNNs) through low-rank regularization. Decomposition models such as CAN-DECOMP/PARAFAC (CP) (Carroll & Chang, 1970) and Tucker decomposition (Tucker, 1966) have effectively reduced the computational complexity of CNNs (Lebedev et al., 2015; Kim et al., 2016). Recent works have extended these techniques to recurrent and transformer layers (Ye et al., 2018; Ma et al., 2019), broadening their applicability. Meanwhile, Chen et al. (2024b) propose filter subspace decomposition for CNN weights along the channel and spatial dimensions. This filter subspace decomposition has shown effectiveness in continual learning (Miao et al., 2022; Chen et al., 2024a), video representation learning (Miao et al., 2021), graph learning (Cheng et al., 2021), and generative tasks (Wang et al., 2021). Moreover, recent advances in structured parameterization, such as Monarch (Dao et al., 2022) and block tensor train (BTT) (Qiu et al., 2024), have further explored parameter-efficient representations for dense layers. Monarch uses structured matrices to achieve hardware efficiency and expressiveness in NLP transformer layers, while BTT leverages block tensor train decomposition for computational efficiency in similar settings. However, these methods primarily target dense layers and remain matrix-based, lacking native support for higher-order tensors, such as convolution filters. In contrast, our work focuses on source-free domain adaptation (SFDA). Tucker decomposition, central to our approach, natively accommodates higher-order tensors, such as those in convolutional layers, by decoupling the core tensor and factor matrices. This decoupling enables fine-grained control over adaptation by selectively tuning the core tensor while freezing the mode factors. For example, as demonstrated in Figure 4, adapting only the core tensor

allows the model to effectively adjust to the target domain (negating source influences) while maintaining parameter efficiency. This fine-tuned control over adaptation contrasts with the uniform rank structures imposed by CP decomposition and the block-wise parameterization of BTT, where it is not entirely clear what should be fine-tuned at the time of target adaptation.

Compared to Monarch and BTT, our approach offers several unique advantages:

- **Parameter Efficiency:** By adapting only the tensor core while freezing the mode factors, our method further minimizes the tunable parameters, achieving computational efficiency without sacrificing the quality of adaptation. This design is particularly beneficial in low-resource settings (Ma et al., 2024) often encountered in time-series.

- **Sample Efficiency:** Beyond computational efficiency, our method excels in sample efficiency, a critical metric in data-scarce scenarios that is central to real-world SFDA applications. This property of Monarch and BTT explicitly needs further investigation.

- **Flexibility Across Modes:** Tucker decomposition allows independent rank adjustments for each mode (*e.g.*, time steps vs. channels), enabling adaptive modeling across diverse dimensions.

- **Interpretability:** The factors resulting from Tucker decomposition can provide insights into domain-specific characteristics, such as temporal correlations or channel-specific importance. This interpretability could be advantageous for analyzing the adaptation dynamics in SFDA (Calvi et al., 2019; Halatsis et al., 2024).

**Our Motivation and Link to Time-Series.** In this work, we aim to address the simultaneous challenges of parameter efficiency and sample efficiency in Source-Free Domain Adaptation (SFDA) for time-series data, leveraging inherent properties of this data type to design a generalizable yet powerful approach.

A key insight that guided our approach is the observation of significant parameter redundancy along the channel dimension of source models trained on time-series data. Figure 2A demonstrates this redundancy, highlighting the inter-channel dependency inherent in multivariate time-series features as they propagate through deep networks. Prior work (Donghao & Xue, 2024; Lai et al., 2017) has shown that modeling dependencies among variables through convolutions along the variable dimension effectively captures these cross-variable relationships. Inspired by this, our method decomposes the source model along the channel dimension, as described in Equation 3, enabling us to focus on the most significant channels or variables while maintaining model performance.

By leveraging the inter-variable dependencies inherent in time-series data, our decomposition approach achieves significant parameter efficiency. Instead of operating on the full channel space, we focus on the principal channels, which correspond to the most critical variables in the parameter space. This focus allows for a compact representation of the model, reducing the number of parameters without sacrificing performance as we observe empirically. The empirical results in Section 4 highlight the effectiveness of this approach across a variety of settings and datasets, demonstrating its versatility.

To ensure the generalizability of our method, we refrained from introducing specific inductive biases, allowing our approach to be applicable across a wide range of SFDA methods. We validated our method with both prominent generalized SFDA techniques (*e.g.*, SHOT, NRC, AAD) and state-of-the-art time-series-specific SFDA method (MAPU) introduced by (Ragab et al., 2023b). Our experiments encompass the five datasets in the AdaTime benchmark (Ragab et al., 2023a)—where prior works (Ragab et al., 2023b) typically focus on only three—along with ten source-target pairs-where prior works experiment with five source-target pairs—including many-to-one scenarios where a source model trained on multiple domains is adapted to new targets.

## A.13 LIMITATIONS

The proposed method has a limitation in that it is not rank-adaptive, meaning it does not adjust the rank based on the intrinsic low-rank structure of the data or model layers. Each dataset and mode (*i.e.*, training vs. inference) has its own optimal low rank, and this ideal rank can also vary across different layers of the model (Sharma et al., 2024). When the rank is not appropriately chosen, the model risks either under-fitting—if the rank is too low and unable to capture the necessary complexity—or over-fitting—if the rank is too high and introduces unnecessary complexity, leading

to poor generalization. Since the current method uses a fixed rank for the entire source model, it may result in suboptimal performance, with inefficiencies in both parameter usage and generalization. Addressing this limitation by incorporating rank-adaptiveness could allow the model to dynamically adjust its rank based on the properties of the data and model layers, thereby reducing the risk of under-fitting and over-fitting and improving overall performance.

Moreover, another limitation of this work is the lack of analysis on the interpretability of the *factor matrices* obtained during decomposition. These factors are presumed to capture representations that generalize across different domains (*i.e.*, source vs. target). However, no effort has been made to explicitly analyze or interpret what these factors represent, leaving open the question of whether or how they effectively capture cross-dataset generalization. This important aspect of understanding the model's behavior is left for future work and could provide valuable insights into how the method works across various domains.

### A.14 FUTURE WORK

While our method leverages significant parameter redundancy along the channel (or variable) dimension in models trained on time-series data (*cf*. Figure 2), this redundancy highlights substantial inter-channel dependencies as time-series data propagate through deep networks. To this end, we propose our method that involves decomposing the backbone. Importantly, we acknowledge that the core principles of our approach are not inherently limited to this domain, and we plan to explore whether such properties are also exhibited in other modalities.

In future work, we aim to extend our method to other areas, such as computer vision, adapting it to handle the distinct characteristics and dependencies of visual data. This exploration will assess the universality of our method, enhance its broader applicability, and contribute to general adaptation strategies across various domains beyond time-series data.

Furthermore, a promising direction involves integrating our method with time-series foundation models (Gao et al., 2024; Ye et al., 2024), which have shown significant potential in capturing generalized representations across diverse domains. By combining their ability to leverage broad pretraining with our fine-tuning efficiency approach, we aim to explore how such synergies can address domain-specific adaptation challenges while maintaining computational feasibility. Investigating the application of time-series foundation models in source-free domain adaptation would also provide valuable insights into the role of generalized features in facilitating efficient adaptation across diverse time-series scenarios.

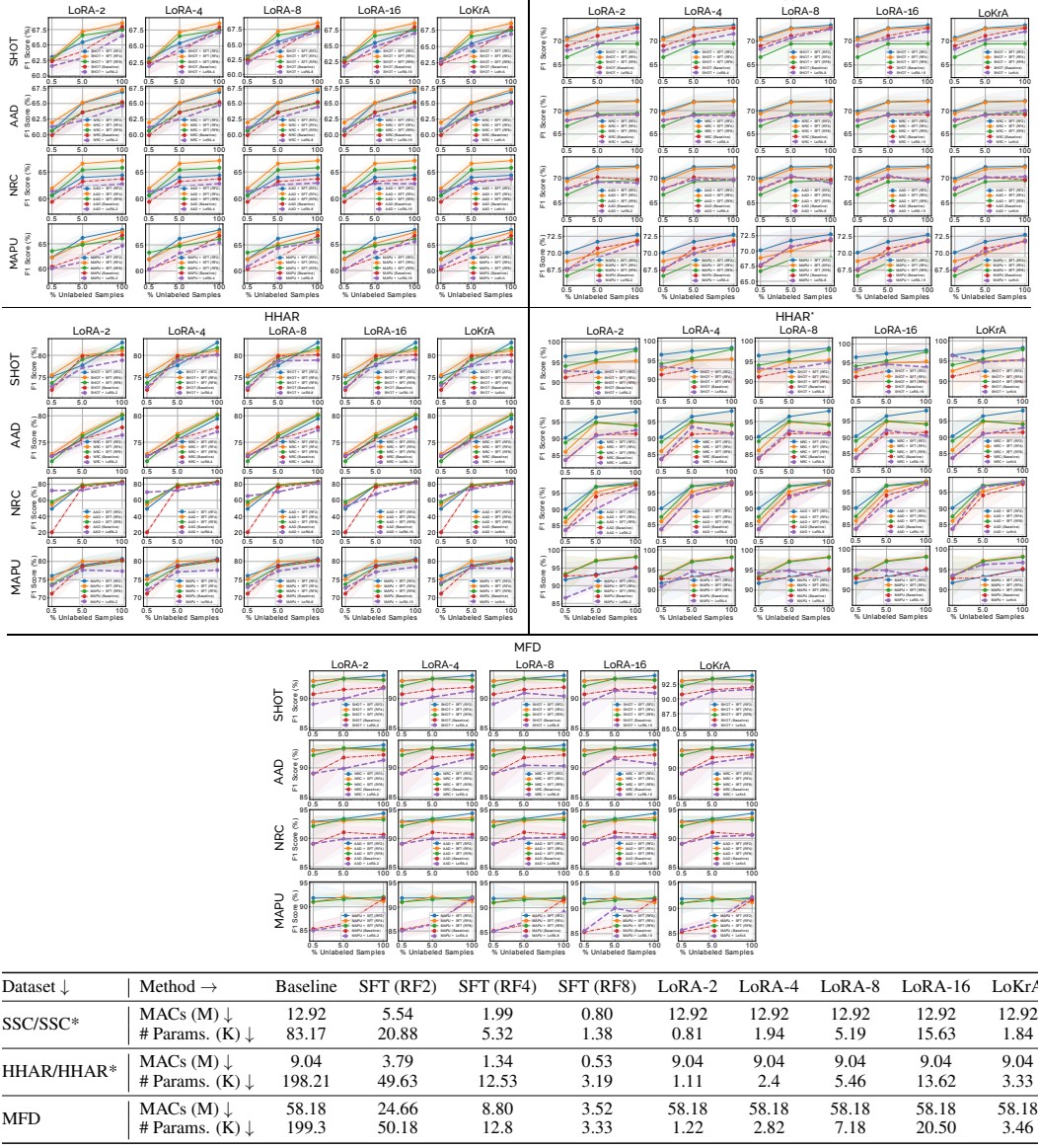

| Dataset ↓ | Method → | Baseline | SFT (RF2) | SFT (RF4) | SFT (RF8) | LoRA-2 | LoRA-4 | LoRA-8 | LoRA-16 | LoKrA |
|---|---|---|---|---|---|---|---|---|---|---|
| SSC/SSC* | MACs (M) ↓ | 12.92 | 5.54 | 1.99 | 0.80 | 12.92 | 12.92 | 12.92 | 12.92 | 12.92 |
|  | # Params. (K) ↓ | 83.17 | 20.88 | 5.32 | 1.38 | 0.81 | 1.94 | 5.19 | 15.63 | 1.84 |
| HHAR/HHAR* | MACs (M) ↓ | 9.04 | 3.79 | 1.34 | 0.53 | 9.04 | 9.04 | 9.04 | 9.04 | 9.04 |
|  | # Params. (K) ↓ | 198.21 | 49.63 | 12.53 | 3.19 | 1.11 | 2.4 | 5.46 | 13.62 | 3.33 |
| MFD | MACs (M) ↓ | 58.18 | 24.66 | 8.80 | 3.52 | 58.18 | 58.18 | 58.18 | 58.18 | 58.18 |
|  | # Params. (K) ↓ | 199.3 | 50.18 | 12.8 | 3.33 | 1.22 | 2.82 | 7.18 | 20.50 | 3.46 |

Figure 12: Comparison of LoRA (Hu et al., 2021) and LoKrA (Edalati et al., 2022; Yeh et al., 2024) (in Purple) against baseline methods (SHOT, NRC, AAD, MAPU) and our proposed approach (SFT) on the SSC, HHAR and MFD datasets in the One-to-one setting and in the Manu-to-one setting for SSC (SSC*) and HHAR (HHAR*), evaluated across varying target sample ratios used during adaptation. The table at the bottom shows the target model's inference overhead after adaptation in terms of MACs and the number of parameters finetuned at the time of adaptation. Extension of the analysis presented in Figure 7.

Table 4: Performance and efficiency comparison on SSC, MFD, HHAR, WISDM, and UCIHAR datasets across SFDA methods, reported as average F1 score (%) at target sample ratios (0.5%, 5%, 100%), inference MACs (M), and fine-tuned parameters (K). Highlighted rows show results for SFT, where only the core tensor is fine-tuned at different $RF$ values. Green numbers represent average percentage improvement, while Red numbers indicate reduction in MACs and fine-tuned parameters.

| Methods | RF | F1 Score (%) ↑ 0.5% | 5% | 100% | Average | MACs (M) ↓ | # Params. (K) ↓ |
|---|---|---|---|---|---|---|---|
| **SSC** | | | | | | | |
| SHOT (Liang et al., 2020) | - | 62.32 ± 1.57 | 63.95 ± 1.51 | 67.95 ± 1.04 | 64.74 | 12.92 | 83.17 |
| SHOT + SFT | 8 | 62.53 ± 0.46 | 66.55 ± 0.46 | 67.50 ± 1.33 | 65.53 (1.22%) | 0.80 (93.81%) | 1.38 (98.34%) |
| | 4 | 62.71 ± 0.57 | 67.16 ± 1.06 | 68.56 ± 0.44 | 66.14 (2.16%) | 1.99 (84.60%) | 5.32 (93.60%) |
| | 2 | 63.05 ± 0.32 | 65.44 ± 0.83 | 67.48 ±0.89 | 65.32 (0.90%) | 5.54 (57.12%) | 20.88 (74.89%) |
| NRC (Yang et al., 2021a) | - | 59.92 ± 1.19 | 63.56 ± 1.35 | 65.23 ± 0.59 | 62.90 | 12.92 | 83.17 |
| NRC + SFT | 8 | 60.65 ± 1.37 | 63.60 ± 1.43 | 65.05 ± 1.66 | 63.10 (0.32%) | 0.80 (93.81%) | 1.38 (98.34%) |
| | 4 | 61.95 ± 0.62 | 65.11 ± 1.34 | 67.19 ± 0.14 | 64.75 (2.94%) | 1.99 (84.60%) | 5.32 (93.60%) |
| | 2 | 60.60 ± 0.58 | 65.06 ± 0.24 | 66.83 ± 0.51 | 64.16 (2.00%) | 5.54 (57.12%) | 20.88 (74.89%) |
| AAD (Yang et al., 2022) | - | 59.39 ± 1.80 | 63.21 ± 1.53 | 63.71 ± 2.06 | 62.10 | 12.92 | 83.17 |
| AAD + SFT | 8 | 60.62 ± 1.40 | 65.38 ± 0.93 | 65.80 ± 1.17 | 63.93 (2.95%) | 0.80 (93.81%) | 1.38 (98.34%) |
| | 4 | 61.92 ± 0.68 | 66.59 ± 0.95 | 67.14 ± 0.57 | 65.22 (5.02%) | 1.99 (84.60%) | 5.32 (93.60%) |
| | 2 | 60.59 ± 0.59 | 63.96 ± 2.04 | 64.39 ± 1.45 | 62.98 (1.42%) | 5.54 (57.12%) | 20.88 (74.89%) |
| MAPU (Ragab et al., 2023b) | - | 60.35 ± 2.15 | 62.48 ± 1.57 | 66.73 ± 0.85 | 63.19 | 12.92 | 83.17 |
| MAPU + SFT | 8 | 63.52 ± 1.24 | 64.77 ± 0.22 | 66.09 ± 0.19 | 64.79 (2.53%) | 0.80 (93.81%) | 1.38 (98.34%) |
| | 4 | 62.21 ± 0.88 | 65.20 ± 1.25 | 67.40 ± 0.59 | 64.94 (2.77%) | 1.99 (84.60%) | 5.32 (93.60%) |
| | 2 | 62.25 ± 1.74 | 66.19 ± 1.02 | 67.85 ± 0.62 | 65.43 (3.54%) | 5.54 (57.12%) | 20.88 (74.89%) |
| **MFD** | | | | | | | |
| SHOT (Liang et al., 2020) | - | 90.74 ± 1.27 | 91.53 ± 1.44 | 91.93 ± 1.37 | 91.40 | 58.18 | 199.3 |
| SHOT + SFT | 8 | 92.14 ± 1.77 | 93.36 ± 0.53 | 93.13 ± 0.54 | 92.88 (1.62%) | 3.52 (93.95%) | 3.33 (98.33%) |
| | 4 | 92.98 ± 1.04 | 93.36 ± 0.75 | 93.21 ± 0.64 | 93.18 (1.95%) | 8.80 (84.87%) | 12.80 (93.58%) |
| | 2 | 93.01 ± 0.79 | 93.40 ± 0.41 | 93.92 ± 0.33 | 93.44 (2.23%) | 24.66 (57.61%) | 50.18 (74.82%) |
| NRC (Yang et al., 2021a) | - | 89.03 ± 3.84 | 91.74 ± 1.31 | 92.20 ± 1.33 | 90.99 | 58.18 | 199.3 |
| NRC + SFT | 8 | 92.14 ± 1.77 | 93.36 ± 0.51 | 93.14 ± 0.7 | 92.88 (2.08%) | 3.52 (93.95%) | 3.33 (98.33%) |
| | 4 | 92.91 ± 1.05 | 93.25 ± 0.69 | 92.97 ± 0.45 | 93.04 (2.25%) | 8.80 (84.87%) | 12.80 (93.58%) |
| | 2 | 92.98 ± 0.78 | 93.22 ± 0.58 | 93.86 ± 0.36 | 93.35 (2.59%) | 24.66 (57.61%) | 50.18 (74.82%) |
| AAD (Yang et al., 2022) | - | 89.03 ± 3.82 | 91.06 ± 2.21 | 90.65 ± 2.48 | 90.25 | 58.18 | 199.3 |
| AAD + SFT | 8 | 92.15 ± 1.78 | 93.36 ± 0.58 | 93.28 ± 0.55 | 92.93 (2.97%) | 3.52 (93.95%) | 3.33 (98.33%) |
| | 4 | 92.88 ± 1.12 | 93.05 ± 0.61 | 93.67 ± 0.41 | 93.20 (3.27%) | 8.80 (84.87%) | 12.80 (93.58%) |
| | 2 | 92.97 ± 0.79 | 93.45 ± 0.53 | 94.41 ± 0.19 | 93.61 (3.72%) | 24.66 (57.61%) | 50.18 (74.82%) |
| MAPU (Ragab et al., 2023b) | - | 85.32 ± 1.25 | 86.49 ± 1.59 | 91.71 ± 0.47 | 87.84 | 58.18 | 199.3 |
| MAPU + SFT | 8 | 91.07 ± 1.76 | 91.58 ± 1.14 | 92.11 ± 1.56 | 91.59 (4.27%) | 3.52 (93.95%) | 3.33 (98.33%) |
| | 4 | 91.01 ± 0.97 | 92.10 ± 1.02 | 91.18 ± 1.64 | 91.43 (4.09%) | 8.80 (84.87%) | 12.80 (93.58%) |
| | 2 | 91.87 ± 2.85 | 91.94 ± 1.44 | 91.84 ± 1.85 | 91.88 (4.60%) | 24.66 (57.61%) | 50.18 (74.82%) |
| **HHAR** | | | | | | | |
| SHOT (Liang et al., 2020) | - | 72.08 ± 1.20 | 79.80 ± 1.70 | 80.12 ± 0.29 | 77.33 | 9.04 | 198.21 |
| SHOT + SFT | 8 | 73.65 ± 2.39 | 79.07 ± 1.88 | 81.73 ± 1.47 | 78.15 (1.06%) | 0.53 (94.14%) | 3.19 (98.39%) |
| | 4 | 75.47 ± 1.34 | 79.98 ± 2.01 | 81.05 ± 0.45 | 78.83 (1.94%) | 1.34 (85.18%) | 12.53 (93.68%) |
| | 2 | 75.14 ± 1.85 | 77.70 ± 1.37 | 82.92 ± 0.27 | 78.59 (1.63%) | 3.79 (58.08%) | 49.63 (74.96%) |
| NRC (Yang et al., 2021a) | - | 72.38 ± 0.87 | 75.52 ± 1.15 | 77.80 ± 0.16 | 75.23 | 9.04 | 198.21 |
| NRC + SFT | 8 | 71.41 ± 0.62 | 76.15 ± 0.99 | 80.09 ± 0.56 | 75.88 (0.86%) | 0.53 (94.14%) | 3.19 (98.39%) |
| | 4 | 72.77 ± 0.43 | 76.60 ± 1.97 | 80.27 ± 0.55 | 76.55 (1.75%) | 1.34 (85.18%) | 12.53 (93.68%) |
| | 2 | 72.34 ± 0.14 | 75.53 ± 1.31 | 79.45 ± 1.51 | 75.77 (0.72%) | 3.79 (58.08%) | 49.63 (74.96%) |
| AAD (Yang et al., 2022) | - | 20.18 ± 2.70 | 76.55 ± 2.17 | 82.25 ± 1.14 | 59.66 | 9.04 | 198.21 |
| AAD + SFT | 8 | 57.93 ± 0.90 | 78.57 ± 0.74 | 82.92 ± 1.66 | 73.14 (22.59%) | 0.53 (94.14%) | 3.19 (98.39%) |
| | 4 | 54.43 ± 1.04 | 78.57 ± 0.74 | 83.15 ± 0.88 | 72.05 (20.77%) | 1.34 (85.18%) | 12.53 (93.68%) |
| | 2 | 49.46 ± 2.22 | 79.31 ± 0.72 | 83.25 ± 0.34 | 70.67 (18.45%) | 3.79 (58.08%) | 49.63 (74.96%) |
| MAPU (Ragab et al., 2023b) | - | 71.18 ± 2.78 | 78.67 ± 1.20 | 80.32 ± 1.16 | 76.72 | 9.04 | 198.21 |
| MAPU + SFT | 8 | 73.73 ± 2.35 | 78.54 ± 2.37 | 80.16 ± 2.38 | 77.48 (0.99%) | 0.53 (94.14%) | 3.19 (98.39%) |
| | 4 | 75.12 ± 0.88 | 80.04 ± 0.42 | 80.24 ± 1.22 | 78.47 (2.28%) | 1.34 (85.18%) | 12.53 (93.68%) |
| | 2 | 76.06 ± 1.24 | 78.95 ± 2.04 | 80.69 ± 2.45 | 78.57 (2.41%) | 3.79 (58.08%) | 49.63 (74.96%) |
| **WISDM** | | | | | | | |
| SHOT (Liang et al., 2020) | - | 57.90 ± 2.07 | 59.52 ± 2.75 | 60.49 ± 1.53 | 59.3 | 9.04 | 198.21 |
| SHOT + SFT | 8 | 60.57 ± 0.79 | 61.35 ± 1.91 | 62.08 ± 2.00 | 61.33 (3.42%) | 0.53 (94.14%) | 3.19 (98.39%) |
| | 4 | 58.06 ± 3.58 | 60.79 ± 1.86 | 61.17 ± 0.32 | 60.01 (1.20%) | 1.34 (85.18%) | 12.53 (93.68%) |
| | 2 | 60.01 ± 0.72 | 63.03 ± 0.61 | 62.04 ± 2.75 | 61.69 (4.03%) | 3.79 (58.08%) | 49.63 (74.96%) |
| NRC (Yang et al., 2021a) | - | 53.58 ± 1.24 | 54.49 ± 0.85 | 58.64 ± 1.39 | 55.57 | 9.04 | 198.21 |
| NRC + SFT | 8 | 57.08 ± 0.94 | 57.61 ± 1.94 | 60.57 ± 1.63 | 58.42 (5.13%) | 0.53 (94.14%) | 3.19 (98.39%) |
| | 4 | 55.89 ± 2.98 | 56.82 ± 1.57 | 60.57 ± 1.63 | 57.76 (3.94%) | 1.34 (85.18%) | 12.53 (93.68%) |
| | 2 | 55.68 ± 2.28 | 55.74 ± 0.95 | 59.21 ± 1.18 | 56.88 (2.36%) | 3.79 (58.08%) | 49.63 (74.96%) |
| AAD (Yang et al., 2022) | - | 54.16 ± 0.85 | 52.79 ± 1.17 | 56.33 ± 1.40 | 54.43 | 9.04 | 198.21 |
| AAD + SFT | 8 | 56.65 ± 0.50 | 57.59 ± 1.67 | 57.69 ± 2.06 | 57.31 (5.29%) | 0.53 (94.14%) | 3.19 (98.39%) |
| | 4 | 55.90 ± 2.67 | 56.71 ± 1.47 | 58.57 ± 2.39 | 57.06 (4.83%) | 1.34 (85.18%) | 12.53 (93.68%) |
| | 2 | 55.83 ± 2.22 | 54.86 ± 1.25 | 58.07 ± 2.75 | 56.25 (3.34%) | 3.79 (58.08%) | 49.63 (74.96%) |
| MAPU (Ragab et al., 2023b) | - | 56.33 ± 4.02 | 58.42 ± 3.39 | 60.92 ± 1.70 | 58.56 | 9.04 | 198.21 |
| MAPU + SFT | 8 | 59.12 ± 3.73 | 61.50 ± 2.23 | 62.55 ± 2.17 | 61.06 (4.27%) | 0.53 (94.14%) | 3.19 (98.39%) |
| | 4 | 62.94 ± 4.08 | 61.00 ± 1.77 | 66.01 ± 3.60 | 63.32 (8.13%) | 1.34 (85.18%) | 12.53 (93.68%) |
| | 2 | 59.36 ± 5.74 | 63.25 ± 1.01 | 62.45 ± 2.66 | 61.69 (5.34%) | 3.79 (58.08%) | 49.63 (74.96%) |
| **UCIHAR** | | | | | | | |
| SHOT (Liang et al., 2020) | - | 89.06 ± 1.17 | 88.89 ± 1.26 | 91.49 ± 0.60 | 89.81 | 9.28 | 200.13 |
| SHOT + SFT | 8 | 89.57 ± 0.44 | 89.05 ± 0.69 | 92.60 ± 1.99 | 90.41 (0.67%) | 0.57 (93.86%) | 3.43 (98.29%) |
| | 4 | 90.02 ± 0.86 | 89.41 ± 0.56 | 92.86 ± 1.02 | 90.76 (1.06%) | 1.41 (84.81%) | 13.01 (93.50%) |
| | 2 | 90.61 ± 0.76 | 90.28 ± 0.28 | 91.65 ± 0.73 | 90.85 (1.16%) | 3.92 (57.76%) | 50.59 (74.72%) |
| NRC (Yang et al., 2021a) | - | 48.13 ± 2.43 | 47.78 ± 1.83 | 91.24 ± 1.08 | 62.38 | 9.28 | 200.13 |
| NRC + SFT | 8 | 86.23 ± 0.60 | 86.99 ± 1.48 | 91.86 ± 0.85 | 88.36 (41.65%) | 0.57 (93.86%) | 3.43 (98.29%) |
| | 4 | 85.21 ± 1.35 | 86.34 ± 1.84 | 94.05 ± 1.01 | 88.53 (41.92%) | 1.41 (84.81%) | 13.01 (93.50%) |
| | 2 | 84.29 ± 1.21 | 86.38 ± 2.02 | 91.51 ± 1.65 | 87.39 (40.09%) | 3.92 (57.76%) | 50.59 (74.72%) |
| AAD (Yang et al., 2022) | - | 57.27 ± 6.75 | 56.21 ± 4.25 | 91.55 ± 0.42 | 68.34 | 9.28 | 200.13 |
| AAD + SFT | 8 | 86.62 ± 1.20 | 87.38 ± 0.17 | 91.56 ± 0.66 | 88.52 (29.53%) | 0.57 (93.86%) | 3.43 (98.29%) |
| | 4 | 86.73 ± 1.36 | 87.42 ± 0.64 | 92.24 ± 0.40 | 88.80 (29.94%) | 1.41 (84.81%) | 13.01 (93.50%) |
| | 2 | 85.51 ± 1.69 | 85.53 ± 2.62 | 91.60 ± 0.18 | 87.55 (28.11%) | 3.92 (57.76%) | 50.59 (74.72%) |
| MAPU (Ragab et al., 2023b) | - | 87.65 ± 3.02 | 86.47 ± 1.46 | 90.86 ± 1.62 | 88.33 | 9.28 | 200.13 |
| MAPU + SFT | 8 | 89.57 ± 0.17 | 89.10 ± 0.98 | 91.93 ± 0.77 | 90.20 (2.12%) | 0.57 (93.86%) | 3.43 (98.29%) |
| | 4 | 89.54 ± 2.26 | 89.39 ± 0.72 | 92.81 ± 1.73 | 90.58 (2.55%) | 1.41 (84.81%) | 13.01 (93.50%) |
| | 2 | 89.34 ± 1.74 | 90.05 ± 0.74 | 91.46 ± 0.49 | 90.28 (2.21%) | 3.92 (57.76%) | 50.59 (74.72%) |

Table 5: F1 scores for each source-target pair, with domain IDs as established by Ragab et al. (2023a), on the SSC dataset across different SFDA methods.

| Method | RF | 0 → 11 | 12 → 5 | 13 → 17 | 16 → 1 | 18 → 12 | 3 → 19 | 5 → 15 | 6 → 2 | 7 → 18 | 9 → 14 | Average (%) ↑ | MACs (M) ↓ | # Params. (K) ↓ |
|---|---|---|---|---|---|---|---|---|---|---|---|---|---|---|
| SHOT | - | 46.83 ± 1.58 | 69.19 ± 5.28 | 64.19 ± 0.18 | 67.99 ± 3.36 | 59.25 ± 2.53 | 75.31 ± 0.77 | 75.08 ± 1.00 | 71.52 ± 1.47 | 74.7 ± 0.54 | 75.39 ± 4.37 | 67.95 ± 1.04 | 12.92 | 83.17 |
| SHOT + SFT | 8 | 46.68 ± 3.81 | 68.98 ± 2.56 | 66.83 ± 3.96 | 66.88 ± 5.08 | 62.74 ± 0.67 | 72.93 ± 1.71 | 71.58 ± 5.43 | 67.75 ± 0.78 | 74.59 ± 1.17 | 76.03 ± 2.35 | 67.50 ± 1.33 | 0.80 | 1.38 |
|  | 4 | 51.32 ± 3.33 | 70.48 ± 1.16 | 64.61 ± 0.67 | 64.81 ± 6.70 | 60.05 ± 3.08 | 74.61 ± 1.14 | 74.29 ± 0.98 | 73.41 ± 0.88 | 75.21 ± 0.47 | 76.83 ± 0.99 | 68.56 ± 0.44 | 1.99 | 5.32 |
|  | 2 | 49.06 ± 5.45 | 68.21 ± 2.77 | 65.67 ± 1.86 | 66.53 ± 4.80 | 58.56 ± 1.41 | 73.44 ± 0.83 | 74.81 ± 0.76 | 72.40 ± 0.42 | 74.50 ± 0.43 | 71.62 ± 9.77 | 67.48 ± 0.89 | 5.54 | 20.88 |
| NRC | - | 42.51 ± 4.43 | 65.93 ± 0.30 | 61.98 ± 0.40 | 62.21 ± 0.71 | 60.82 ± 2.61 | 68.09 ± 0.58 | 71.49 ± 4.75 | 72.36 ± 1.51 | 72.81 ± 0.85 | 74.16 ± 1.66 | 65.23 ± 0.59 | 12.92 | 83.17 |
| NRC + SFT | 8 | 47.14 ± 1.76 | 56.97 ± 9.93 | 61.63 ± 1.25 | 64.28 ± 0.59 | 64.29 ± 2.05 | 69.59 ± 0.95 | 72.81 ± 3.67 | 65.83 ± 2.03 | 74.73 ± 1.74 | 73.25 ± 0.94 | 65.05 ± 1.66 | 0.80 | 1.38 |
|  | 4 | 51.01 ± 4.43 | 65.71 ± 2.33 | 62.65 ± 0.86 | 62.22 ± 0.24 | 63.43 ± 2.19 | 71.41 ± 1.69 | 75.13 ± 3.54 | 71.25 ± 1.55 | 74.53 ± 0.57 | 74.59 ± 1.13 | 67.19 ± 0.14 | 1.99 | 5.32 |
|  | 2 | 50.53 ± 4.21 | 64.57 ± 4.55 | 63.93 ± 0.17 | 60.87 ± 0.34 | 60.55 ± 1.62 | 68.77 ± 6.08 | 76.94 ± 1.51 | 72.88 ± 0.70 | 73.93 ± 0.16 | 75.30 ± 1.53 | 66.83 ± 0.51 | 5.54 | 20.88 |
| AAD | - | 47.83 ± 4.05 | 60.68 ± 2.05 | 57.71 ± 2.08 | 63.64 ± 1.29 | 55.70 ± 2.50 | 71.51 ± 0.81 | 61.46 ± 16.01 | 73.53 ± 1.08 | 73.73 ± 3.37 | 71.35 ± 5.25 | 63.71 ± 2.06 | 12.92 | 83.17 |
| AAD + SFT | 8 | 47.94 ± 3.36 | 64.35 ± 2.71 | 58.85 ± 1.59 | 64.48 ± 0.65 | 61.13 ± 1.71 | 71.31 ± 0.95 | 73.07 ± 4.57 | 69.09 ± 1.66 | 75.37 ± 0.68 | 72.40 ± 0.81 | 65.80 ± 1.17 | 0.80 | 1.38 |
|  | 4 | 48.58 ± 5.64 | 69.56 ± 2.74 | 60.04 ± 3.42 | 65.13 ± 5.29 | 59.44 ± 2.78 | 72.91 ± 0.85 | 74.97 ± 3.48 | 72.17 ± 1.11 | 75.53 ± 0.28 | 73.07 ± 1.17 | 67.14 ± 0.57 | 1.99 | 5.32 |
|  | 2 | 52.66 ± 1.20 | 64.00 ± 9.31 | 62.04 ± 2.91 | 62.99 ± 0.77 | 61.47 ± 1.98 | 72.09 ± 2.10 | 48.76 ± 15.24 | 73.48 ± 1.76 | 74.53 ± 0.40 | 71.91 ± 2.60 | 64.39 ± 1.45 | 5.54 | 20.88 |
| MAPU | - | 47.43 ± 2.31 | 68.24 ± 2.98 | 60.04 ± 0.56 | 61.04 ± 0.86 | 57.35 ± 0.98 | 73.18 ± 0.26 | 76.35 ± 0.49 | 69.89 ± 1.93 | 76.44 ± 0.56 | 77.33 ± 1.97 | 66.73 ± 0.85 | 12.92 | 83.17 |
| MAPU + SFT | 8 | 46.72 ± 1.69 | 68.86 ± 2.64 | 63.01 ± 0.96 | 61.19 ± 0.37 | 59.20 ± 3.02 | 70.19 ± 2.33 | 75.39 ± 3.45 | 66.25 ± 0.70 | 75.51 ± 0.37 | 74.61 ± 2.31 | 66.09 ± 0.19 | 0.80 | 1.38 |
|  | 4 | 47.06 ± 0.32 | 70.34 ± 3.06 | 63.33 ± 1.38 | 64.90 ± 1.81 | 55.99 ± 2.05 | 72.63 ± 1.37 | 78.31 ± 1.21 | 68.17 ± 1.71 | 75.47 ± 1.01 | 77.83 ± 2.81 | 67.40 ± 0.59 | 1.99 | 5.32 |
|  | 2 | 50.99 ± 0.53 | 71.86 ± 0.73 | 64.21 ± 0.54 | 64.99 ± 5.66 | 54.38 ± 1.89 | 73.65 ± 1.03 | 77.55 ± 0.84 | 69.98 ± 1.08 | 75.27 ± 0.61 | 75.62 ± 1.23 | 67.85 ± 0.62 | 5.54 | 20.88 |

Table 6: F1 scores for each source-target pair, with domain IDs as established by Ragab et al. (2023a), on the MFD dataset across different SFDA methods.

| Method | RF | 0 → 1 | 0 → 2 | 0 → 3 | 1 → 0 | 1 → 2 | 1 → 3 | 2 → 0 | 2 → 1 | 2 → 3 | 3 → 0 | 3 → 1 | 3 → 2 | Average (%) ↑ | MACs (M) ↓ | # Params. (K) ↓ |
|---|---|---|---|---|---|---|---|---|---|---|---|---|---|---|---|---|
| SHOT | - | 98.26 ± 1.89 | 86.02 ± 2.72 | 97.77 ± 2.40 | 87.56 ± 3.47 | 87.88 ± 2.60 | 99.97 ± 0.05 | 72.39 ± 17.01 | 98.81 ± 1.25 | 99.18 ± 0.99 | 88.72 ± 1.55 | 100.00 ± 0.00 | 86.56 ± 3.26 | 91.93 ± 1.37 | 58.18 | 199.3 |
| SHOT + SFT | 8 | 99.19 ± 0.35 | 87.42 ± 2.85 | 99.02 ± 0.49 | 86.57 ± 3.77 | 88.48 ± 1.70 | 99.97 ± 0.05 | 82.21 ± 2.93 | 98.59 ± 1.10 | 99.21 ± 0.73 | 88.57 ± 2.30 | 100.00 ± 0.00 | 88.31 ± 3.31 | 93.13 ± 0.54 | 3.52 | 3.33 |
|  | 4 | 99.59 ± 0.08 | 86.31 ± 1.81 | 99.73 ± 0.19 | 84.91 ± 3.69 | 90.67 ± 0.52 | 99.97 ± 0.05 | 83.63 ± 2.20 | 97.15 ± 2.21 | 99.05 ± 0.47 | 87.60 ± 2.47 | 100.00 ± 0.00 | 99.88 ± 1.16 | 93.21 ± 0.64 | 8.80 | 12.80 |
|  | 2 | 99.87 ± 0.12 | 87.05 ± 2.89 | 99.81 ± 0.13 | 85.72 ± 0.59 | 91.06 ± 1.00 | 99.97 ± 0.05 | 87.49 ± 1.10 | 99.19 ± 0.49 | 99.54 ± 0.31 | 88.88 ± 0.69 | 100.00 ± 0.00 | 88.48 ± 0.66 | 93.92 ± 0.33 | 24.66 | 50.18 |
| NRC | - | 98.10 ± 2.44 | 85.00 ± 10.08 | 98.31 ± 1.92 | 88.53 ± 1.73 | 88.50 ± 3.31 | 100.00 ± 0.00 | 74.52 ± 20.08 | 98.64 ± 1.19 | 99.02 ± 0.80 | 88.69 ± 1.68 | 100.00 ± 0.00 | 87.08 ± 4.27 | 92.30 ± 1.33 | 58.18 | 199.3 |
| NRC + SFT | 8 | 99.27 ± 0.42 | 87.48 ± 2.94 | 99.16 ± 0.37 | 86.58 ± 3.62 | 88.43 ± 1.62 | 99.97 ± 0.05 | 81.96 ± 2.38 | 98.64 ± 1.06 | 99.32 ± 0.69 | 88.44 ± 2.20 | 100.00 ± 0.00 | 88.43 ± 3.39 | 93.14 ± 0.70 | 3.52 | 3.33 |
|  | 4 | 99.56 ± 0.12 | 86.40 ± 1.96 | 99.76 ± 0.22 | 84.96 ± 3.91 | 90.53 ± 0.74 | 99.97 ± 0.05 | 80.91 ± 5.74 | 97.15 ± 2.12 | 99.02 ± 0.51 | 87.62 ± 2.65 | 100.00 ± 0.00 | 89.77 ± 1.10 | 92.97 ± 0.45 | 8.80 | 12.80 |
|  | 2 | 99.92 ± 0.08 | 86.77 ± 3.66 | 99.87 ± 0.09 | 85.70 ± 0.60 | 91.03 ± 1.15 | 99.97 ± 0.05 | 87.02 ± 0.89 | 99.16 ± 0.40 | 99.51 ± 0.28 | 88.88 ± 0.69 | 100.00 ± 0.00 | 88.48 ± 0.66 | 93.86 ± 0.36 | 24.66 | 50.18 |
| AAD | - | 98.20 ± 2.32 | 85.04 ± 10.09 | 99.40 ± 0.49 | 81.12 ± 8.98 | 86.88 ± 2.31 | 100.00 ± 0.00 | 62.98 ± 18.46 | 98.75 ± 1.26 | 99.16 ± 1.04 | 88.98 ± 1.78 | 100.00 ± 0.00 | 87.33 ± 4.41 | 90.65 ± 2.48 | 58.18 | 199.3 |
| AAD + SFT | 8 | 99.87 ± 0.09 | 88.60 ± 3.10 | 99.87 ± 0.12 | 84.73 ± 3.75 | 89.45 ± 1.35 | 100.00 ± 0.00 | 80.73 ± 3.37 | 98.97 ± 0.69 | 99.62 ± 0.45 | 87.99 ± 1.58 | 100.00 ± 0.00 | 89.55 ± 2.86 | 93.28 ± 0.55 | 3.52 | 3.33 |
|  | 4 | 99.97 ± 0.05 | 90.53 ± 1.96 | 99.95 ± 0.05 | 84.00 ± 4.74 | 91.22 ± 1.55 | 100.00 ± 0.00 | 80.73 ± 3.35 | 98.62 ± 0.33 | 99.49 ± 0.38 | 88.24 ± 1.67 | 100.00 ± 0.00 | 90.92 ± 1.42 | 93.67 ± 0.41 | 8.80 | 12.80 |
|  | 2 | 99.97 ± 0.05 | 90.92 ± 1.75 | 100.00 ± 0.00 | 84.11 ± 2.67 | 92.06 ± 1.52 | 100.00 ± 0.00 | 88.62 ± 0.52 | 98.89 ± 0.44 | 99.84 ± 0.08 | 88.80 ± 0.49 | 100.00 ± 0.00 | 89.08 ± 0.58 | 94.41 ± 0.19 | 24.66 | 50.18 |
| MAPU | - | 94.76 ± 1.08 | 87.53 ± 1.05 | 94.09 ± 1.72 | 86.86 ± 0.39 | 87.51 ± 1.72 | 99.08 ± 0.91 | 78.61 ± 3.75 | 99.16 ± 0.38 | 98.70 ± 1.34 | 88.48 ± 0.99 | 100.00 ± 0.00 | 85.80 ± 2.82 | 91.71 ± 0.47 | 58.18 | 199.3 |
| MAPU + SFT | 8 | 94.85 ± 2.99 | 86.16 ± 3.50 | 94.84 ± 4.56 | 84.31 ± 2.02 | 86.63 ± 2.43 | 99.81 ± 0.33 | 86.10 ± 3.25 | 99.32 ± 0.62 | 99.21 ± 0.73 | 88.21 ± 3.24 | 100.00 ± 0.00 | 86.32 ± 0.00 | 92.11 ± 1.56 | 3.52 | 3.33 |
|  | 4 | 89.45 ± 5.03 | 85.45 ± 3.94 | 92.15 ± 3.78 | 86.33 ± 4.70 | 86.31 ± 0.67 | 99.67 ± 0.57 | 84.70 ± 1.40 | 98.23 ± 2.02 | 98.42 ± 1.97 | 88.21 ± 3.24 | 99.84 ± 0.28 | 85.38 ± 1.02 | 91.18 ± 1.64 | 8.80 | 12.80 |
|  | 2 | 92.39 ± 7.56 | 86.29 ± 4.12 | 93.62 ± 5.38 | 85.61 ± 3.81 | 86.01 ± 1.08 | 100.00 ± 0.00 | 86.27 ± 1.36 | 99.37 ± 0.38 | 99.56 ± 0.12 | 88.18 ± 1.87 | 100.00 ± 0.00 | 84.83 ± 0.15 | 91.84 ± 1.85 | 24.66 | 50.18 |

Table 7: F1 scores for each source-target pair, with domain IDs as established by Ragab et al. (2023a), on the HHAR dataset across different SFDA methods.

| Method | RF | 0 → 2 | 0 → 6 | 1 → 6 | 2 → 7 | 3 → 8 | 4 → 5 | 5 → 0 | 6 → 1 | 7 → 4 | 8 → 3 | Average (%) ↑ | MACs (M) ↓ | # Params. (K) ↓ |
|---|---|---|---|---|---|---|---|---|---|---|---|---|---|---|
| SHOT | - | 78.72 ± 1.36 | 62.94 ± 0.41 | 92.71 ± 0.72 | 64.15 ± 0.82 | 81.65 ± 0.18 | 97.41 ± 0.19 | 32.42 ± 0.02 | 97.58 ± 0.44 | 96.44 ± 1.74 | 97.14 ± 0.12 | 80.12 ± 0.29 | 9.04 | 198.21 |
| SHOT + SFT | 8 | 76.05 ± 9.93 | 66.31 ± 6.43 | 92.97 ± 0.35 | 64.59 ± 0.49 | 92.50 ± 9.28 | 97.36 ± 0.11 | 34.85 ± 4.88 | 97.80 ± 0.67 | 97.01 ± 0.21 |  | 81.73 ± 1.47 | 0.53 | 3.19 |
|  | 4 | 78.07 ± 6.48 | 64.06 ± 3.29 | 93.04 ± 0.09 | 64.75 ± 0.19 | 92.59 ± 9.41 | 97.60 ± 0.19 | 35.08 ± 4.62 | 97.88 ± 0.11 | 97.68 ± 0.24 | 89.76 ± 12.54 | 81.05 ± 0.45 | 1.34 | 12.53 |
|  | 2 | 81.15 ± 1.21 | 62.98 ± 0.23 | 92.80 ± 1.25 | 65.11 ± 0.17 | 98.91 ± 0.16 | 97.48 ± 0.10 | 37.59 ± 4.63 | 98.10 ± 0.32 | 98.04 ± 0.43 | 97.00 ± 0.00 | 82.92 ± 0.27 | 3.79 | 49.63 |
| NRC | - | 73.05 ± 0.68 | 72.40 ± 1.29 | 92.96 ± 0.29 | 61.61 ± 0.31 | 80.63 ± 0.22 | 97.04 ± 0.30 | 33.77 ± 2.08 | 96.03 ± 0.28 | 94.71 ± 0.36 | 75.81 ± 0.31 | 77.80 ± 0.16 | 9.04 | 198.21 |
| NRC + SFT | 8 | 72.77 ± 0.25 | 68.05 ± 5.07 | 93.20 ± 0.27 | 58.41 ± 9.00 | 97.03 ± 1.78 | 97.41 ± 0.19 | 42.83 ± 0.02 | 97.70 ± 0.25 | 97.81 ± 0.12 | 75.71 ± 0.38 | 80.09 ± 0.56 | 0.53 | 3.19 |
|  | 4 | 74.37 ± 0.32 | 70.01 ± 3.77 | 93.38 ± 0.39 | 63.27 ± 0.26 | 96.47 ± 0.27 | 97.54 ± 0.11 | 39.73 ± 2.58 | 97.14 ± 0.09 | 95.19 ± 0.22 | 75.57 ± 0.33 | 80.27 ± 0.55 | 1.34 | 12.53 |
|  | 2 | 74.79 ± 0.12 | 68.67 ± 4.77 | 93.52 ± 0.28 | 58.29 ± 9.25 | 97.83 ± 1.37 | 97.66 ± 0.22 | 34.50 ± 5.01 | 97.30 ± 0.38 | 96.54 ± 0.98 | 75.35 ± 0.13 | 79.45 ± 1.51 | 3.79 | 49.63 |
| AAD | - | 77.82 ± 9.33 | 64.86 ± 0.83 | 93.16 ± 0.46 | 66.14 ± 0.07 | 97.55 ± 1.71 | 97.89 ± 0.21 | 33.99 ± 2.39 | 97.58 ± 0.94 | 98.45 ± 0.13 | 97.20 ± 0.00 | 82.25 ± 1.14 | 9.04 | 198.21 |
| AAD + SFT | 8 | 78.16 ± 2.25 | 76.63 ± 16.99 | 92.84 ± 0.56 | 64.30 ± 0.17 | 97.73 ± 1.66 | 98.12 ± 0.50 | 27.98 ± 5.05 | 97.74 ± 0.53 | 98.65 ± 0.13 | 97.09 ± 0.12 | 82.92 ± 1.66 | 0.53 | 3.19 |
|  | 4 | 77.85 ± 16.03 | 74.10 ± 19.41 | 92.68 ± 0.59 | 65.66 ± 0.36 | 97.27 ± 0.21 | 98.28 ± 0.67 | 30.79 ± 3.48 | 97.79 ± 0.34 | 98.52 ± 0.22 | 97.07 ± 0.11 | 83.15 ± 0.88 | 1.34 | 12.53 |
|  | 2 | 82.72 ± 4.32 | 64.89 ± 4.07 | 92.81 ± 0.21 | 65.46 ± 0.70 | 99.23 ± 0.20 | 97.87 ± 0.09 | 35.84 ± 5.02 | 98.19 ± 0.09 | 98.51 ± 0.30 | 97.00 ± 0.00 | 83.25 ± 0.34 | 3.79 | 49.63 |
| MAPU | - | 75.26 ± 4.86 | 62.89 ± 0.65 | 95.49 ± 1.02 | 65.20 ± 0.51 | 99.27 ± 0.32 | 97.31 ± 0.21 | 31.36 ± 4.53 | 95.65 ± 3.32 | 98.10 ± 0.12 | 82.22 ± 13.01 | 80.32 ± 1.16 | 9.04 | 198.21 |
| MAPU + SFT | 8 | 68.40 ± 5.72 | 74.03 ± 18.84 | 93.75 ± 0.31 | 64.78 ± 0.26 | 92.23 ± 11.57 | 97.48 ± 0.13 | 29.24 ± 4.70 | 97.90 ± 0.69 | 86.18 ± 18.72 |  | 80.16 ± 2.38 | 0.53 | 3.19 |
|  | 4 | 74.85 ± 6.47 | 62.81 ± 0.53 | 93.28 ± 0.22 | 64.60 ± 0.25 | 93.33 ± 10.44 | 97.48 ± 0.10 | 30.50 ± 2.99 | 97.65 ± 0.50 | 98.03 ± 0.12 | 89.89 ± 12.29 | 80.24 ± 1.22 | 1.34 | 12.53 |
|  | 2 | 75.62 ± 7.66 | 63.15 ± 0.50 | 93.43 ± 0.11 | 64.55 ± 0.38 | 99.21 ± 0.22 | 97.54 ± 0.12 | 30.75 ± 2.78 | 98.32 ± 0.37 | 98.24 ± 0.12 | 86.12 ± 18.65 | 80.69 ± 2.45 | 3.79 | 49.63 |

Table 8: F1 scores for each source-target pair, with domain IDs as established by Ragab et al. (2023a), on the WISDM dataset across different SFDA methods.

| Method | RF | 17 → 23 | 20 → 30 | 23 → 32 | 28 → 4 | 2 → 11 | 33 → 12 | 35 → 31 | 5 → 26 | 6 → 19 | 7 → 18 | Average (%) ↑ | MACs (M) ↓ | # Params. (K) ↓ |
|---|---|---|---|---|---|---|---|---|---|---|---|---|---|---|
| SHOT | - | 58.58 ± 2.34 | 70.21 ± 1.29 | 72.16 ± 9.68 | 61.68 ± 4.00 | 78.72 ± 1.75 | 50.70 ± 2.30 | 65.89 ± 0.51 | 31.20 ± 0.96 | 74.85 ± 4.97 | 40.89 ± 9.92 | 60.49 ± 1.53 | 9.04 | 198.21 |
| SHOT + SFT | 8 | 54.53 ± 18.59 | 72.03 ± 0.40 | 81.69 ± 1.95 | 53.97 ± 0.00 | 72.69 ± 3.82 | 59.70 ± 12.21 | 64.56 ± 9.23 | 37.24 ± 8.27 | 76.58 ± 0.81 | 47.82 ± 4.29 | 62.08 ± 2.00 | 0.53 | 3.19 |
|  | 4 | 53.46 ± 6.77 | 75.43 ± 1.58 | 76.05 ± 7.35 | 62.35 ± 4.17 | 76.05 ± 3.74 | 50.38 ± 5.95 | 66.42 ± 1.46 | 30.25 ± 0.50 | 77.42 ± 2.27 | 43.92 ± 5.09 | 61.17 ± 0.32 | 1.34 | 12.53 |
|  | 2 | 58.18 ± 14.69 | 70.93 ± 20.90 | 80.10 ± 1.56 | 60.46 ± 8.39 | 75.42 ± 0.35 | 61.82 ± 11.81 | 66.26 ± 2.29 | 32.85 ± 1.51 | 80.85 ± 8.22 | 33.49 ± 4.54 | 62.04 ± 2.75 | 3.79 | 49.63 |
| NRC | - | 47.23 ± 11.21 | 64.66 ± 1.45 | 81.43 ± 2.65 | 61.68 ± 11.67 | 76.62 ± 4.43 | 34.44 ± 2.75 | 78.61 ± 4.57 | 32.22 ± 0.28 |  |  | 58.54 ± 1.39 | 9.04 | 198.21 |
| NRC + SFT | 8 | 39.61 ± 1.75 | 69.40 ± 7.84 | 81.39 ± 1.58 | 60.90 ± 13.30 | 75.01 ± 0.00 | 60.79 ± 10.03 | 67.58 ± 2.29 | 40.11 ± 7.27 | 77.85 ± 3.01 | 33.09 ± 1.06 | 60.57 ± 1.63 | 0.53 | 3.19 |
|  | 4 | 47.73 ± 7.38 | 75.93 ± 1.37 | 76.58 ± 8.77 | 64.32 ± 6.75 | 76.71 ± 4.83 | 48.80 ± 4.91 | 76.09 ± 2.02 | 30.60 ± 0.80 | 76.27 ± 0.27 | 41.71 ± 6.55 | 60.06 ± 0.92 | 1.34 | 12.53 |
|  | 2 | 39.01 ± 1.09 | 70.63 ± 5.44 | 81.63 ± 1.34 | 63.71 ± 10.13 | 75.32 ± 0.00 | 49.26 ± 0.70 | 67.16 ± 1.17 | 31.44 ± 1.42 | 76.03 ± 0.13 | 37.94 ± 4.02 | 59.21 ± 1.18 | 3.79 | 49.63 |
| AAD | - | 56.63 ± 2.31 | 65.89 ± 3.00 | 66.63 ± 12.44 | 58.24 ± 7.55 | 77.97 ± 4.18 | 51.83 ± 3.47 | 47.96 ± 14.12 | 29.22 ± 0.66 | 67.11 ± 5.68 | 41.79 ± 6.70 | 56.33 ± 1.40 | 9.04 | 198.21 |
| AAD + SFT | 8 | 47.82 ± 6.26 | 71.76 ± 3.24 | 66.46 ± 7.21 | 65.12 ± 10.69 | 75.10 ± 5.69 | 47.25 ± 4.73 | 67.99 ± 2.02 | 29.40 ± 0.26 | 61.80 ± 1.55 | 44.21 ± 1.22 | 57.69 ± 2.06 | 0.53 | 3.19 |
|  | 4 | 52.32 ± 7.72 | 74.97 ± 0.46 | 66.65 ± 5.75 | 68.28 ± 5.36 | 68.57 ± 13.95 | 47.25 ± 5.33 | 65.85 ± 1.36 | 29.31 ± 0.13 | 58.21 ± 4.45 | 41.58 ± 1.64 | 58.57 ± 2.39 | 1.34 | 12.53 |
|  | 2 | 52.36 ± 9.50 | 70.70 ± 3.96 | 72.38 ± 4.30 | 67.40 ± 2.31 | 69.02 ± 13.95 | 44.21 ± 0.15 | 69.00 ± 1.63 | 30.42 ± 1.45 | 63.62 ± 2.42 | 50.11 ± 1.06 | 58.07 ± 2.75 | 3.79 | 49.63 |
| MAPU | - | 65.36 ± 9.33 | 66.00 ± 0.61 | 73.10 ± 12.28 | 66.81 ± 10.92 | 68.50 ± 6.65 | 58.73 ± 12.04 | 68.35 ± 5.11 | 31.48 ± 0.16 | 79.44 ± 9.08 | 31.47 ± 1.14 | 60.92 ± 1.70 | 9.04 | 198.21 |
| MAPU + SFT | 8 | 68.09 ± 10.85 | 70.04 ± 4.02 | 79.14 ± 6.07 | 60.25 ± 11.34 | 72.88 ± 3.48 | 59.47 ± 6.28 | 83.03 ± 7.22 | 31.35 ± 0.04 | 73.98 ± 3.69 | 47.31 ± 3.28 | 62.55 ± 2.17 | 0.53 | 3.19 |
|  | 4 | 73.47 ± 3.85 | 69.68 ± 3.93 | 80.85 ± 0.72 | 67.60 ± 19.71 | 73.96 ± 1.82 | 61.60 ± 9.32 | 64.94 ± 0.00 | 33.86 ± 1.86 | 80.85 ± 8.22 | 53.31 ± 4.72 | 66.01 ± 3.60 | 1.34 | 12.53 |
|  | 2 | 60.04 ± 4.66 | 72.04 ± 4.64 | 65.85 ± 12.32 | 61.97 ± 9.95 | 75.13 ± 0.44 | 60.42 ± 10.41 | 64.02 ± 2.62 | 33.03 ± 2.50 | 81.90 ± 8.84 | 50.11 ± 1.06 | 62.45 ± 2.66 | 3.79 | 49.63 |

Table 9: F1 scores for each source-target pair, with domain IDs as established by Ragab et al. (2023a), on the UCIHAR dataset across different SFDA methods.

| Method | RF | 12 → 16 | 18 → 27 | 20 → 5 | 24 → 8 | 28 → 27 | 2 → 11 | 30 → 20 | 6 → 23 | 7 → 13 | 9 → 18 | Average (%) ↑ | MACs (M) ↓ | # Params. (K) ↓ |
|---|---|---|---|---|---|---|---|---|---|---|---|---|---|---|
| SHOT | - | 53.11 ± 4.28 | 100.00 ± 0.00 | 90.83 ± 4.60 | 100.00 ± 0.00 | 100.00 ± 0.00 | 100.00 ± 0.00 | 84.68 ± 5.09 | 97.82 ± 1.89 | 100.00 ± 0.00 | 88.48 ± 5.22 | 91.49 ± 0.60 | 9.28 | 200.13 |
| SHOT + SFT | 8 | 54.63 ± 17.43 | 100.00 ± 0.00 | 95.83 ± 4.53 | 100.00 ± 0.00 | 100.00 ± 0.00 | 100.00 ± 0.00 | 84.91 ± 2.87 | 98.91 ± 1.89 | 99.94 ± 1.06 | 93.92 ± 6.96 | 92.60 ± 1.99 | 0.57 | 3.43 |
|  | 4 | 64.21 ± 11.40 | 100.00 ± 0.00 | 92.48 ± 9.43 | 100.00 ± 0.00 | 100.00 ± 0.00 | 100.00 ± 0.00 | 85.40 ± 1.80 | 98.91 ± 1.89 | 100.00 ± 0.00 | 87.55 ± 11.89 | 92.86 ± 1.02 | 1.41 | 13.01 |
|  | 2 | 70.97 ± 2.38 | 100.00 ± 0.00 | 82.84 ± 1.63 | 98.89 ± 1.93 | 100.00 ± 0.00 | 99.63 ± 0.64 | 83.58 ± 1.23 | 97.82 ± 1.89 | 100.00 ± 0.00 | 82.74 ± 6.20 | 91.65 ± 0.73 | 3.92 | 50.59 |
| NRC | - | 56.47 ± 10.19 | 100.00 ± 0.00 | 96.10 ± 0.64 | 96.91 ± 5.35 | 100.00 ± 0.00 | 100.00 ± 0.00 | 83.48 ± 3.33 | 97.82 ± 1.89 | 100.00 ± 0.00 | 81.65 ± 5.26 | 91.24 ± 1.08 | 9.28 | 200.13 |
| NRC + SFT | 8 | 66.04 ± 6.76 | 100.00 ± 0.00 | 87.09 ± 1.05 | 98.89 ± 1.93 | 100.00 ± 0.00 | 100.00 ± 0.00 | 87.92 ± 1.28 | 96.73 ± 0.00 | 96.72 ± 2.98 | 85.21 ± 0.96 | 91.86 ± 0.85 | 0.57 | 3.43 |
|  | 4 | 67.98 ± 2.76 | 100.00 ± 0.00 | 95.27 ± 0.09 | 100.00 ± 0.00 | 100.00 ± 0.00 | 100.00 ± 0.00 | 88.79 ± 5.64 | 96.73 ± 0.00 | 97.78 ± 3.85 | 93.92 ± 6.96 | 94.05 ± 1.01 | 1.41 | 13.01 |
|  | 2 | 71.22 ± 2.80 | 100.00 ± 0.00 | 85.88 ± 0.00 | 100.00 ± 0.00 | 100.00 ± 0.00 | 100.00 ± 0.00 | 84.60 ± 5.80 | 96.73 ± 0.00 | 97.78 ± 3.85 | 78.93 ± 12.81 | 91.51 ± 1.65 | 3.92 | 50.59 |
| AAD | - | 72.85 ± 2.04 | 100.00 ± 0.00 | 85.88 ± 0.89 | 96.66 ± 0.00 | 100.00 ± 0.00 | 100.00 ± 0.00 | 84.83 ± 3.14 | 96.73 ± 0.00 | 93.33 ± 0.00 | 85.23 ± 0.94 | 91.55 ± 0.42 | 9.28 | 200.13 |
| AAD + SFT | 8 | 74.30 ± 5.08 | 100.00 ± 0.00 | 87.69 ± 0.00 | 98.89 ± 1.93 | 100.00 ± 0.00 | 100.00 ± 0.00 | 81.61 ± 5.49 | 96.73 ± 0.00 | 97.78 ± 3.85 | 78.62 ± 5.26 | 91.56 ± 0.66 | 0.57 | 3.43 |
|  | 4 | 74.05 ± 0.40 | 100.00 ± 0.00 | 86.77 ± 0.94 | 97.77 ± 1.93 | 100.00 ± 0.00 | 100.00 ± 0.00 | 84.35 ± 5.65 | 96.73 ± 0.00 | 100.00 ± 0.00 | 82.74 ± 6.20 | 92.24 ± 0.40 | 1.41 | 13.01 |
|  | 2 | 73.30 ± 2.56 | 100.00 ± 0.00 | 82.78 ± 2.76 | 97.77 ± 1.93 | 100.00 ± 0.00 | 100.00 ± 0.00 | 79.19 ± 2.13 | 98.91 ± 1.89 | 97.78 ± 3.85 | 86.32 ± 0.00 | 91.08 ± 0.18 | 3.92 | 50.59 |
| MAPU | - | 60.45 ± 8.59 | 100.00 ± 0.00 | 82.94 ± 2.06 | 100.00 ± 0.00 | 100.00 ± 0.00 | 100.00 ± 0.00 | 85.17 ± 0.44 | 100.00 ± 0.00 | 100.00 ± 0.00 | 80.05 ± 13.71 | 90.86 ± 1.62 | 9.28 | 200.13 |
| MAPU + SFT | 8 | 60.50 ± 9.50 | 100.00 ± 0.00 | 86.63 ± 8.60 | 100.00 ± 0.00 | 100.00 ± 0.00 | 100.00 ± 0.00 | 88.76 ± 1.46 | 97.82 ± 1.89 | 99.29 ± 1.22 | 86.25 ± 0.13 | 91.93 ± 0.77 | 0.57 | 3.43 |
|  | 4 | 68.07 ± 4.73 | 99.75 ± 0.43 | 88.29 ± 6.68 | 99.63 ± 0.65 | 100.00 ± 0.00 | 99.26 ± 0.94 | 89.21 ± 4.08 | 98.91 ± 1.89 | 97.43 ± 3.59 | 87.55 ± 11.89 | 91.93 ± 0.77 | 1.41 | 13.01 |
|  | 2 | 72.29 ± 2.93 | 100.00 ± 0.00 | 83.47 ± 3.06 | 100.00 ± 0.00 | 100.00 ± 0.00 | 100.00 ± 0.00 | 87.68 ± 0.42 | 97.82 ± 1.89 | 97.78 ± 3.85 | 75.58 ± 0.00 | 91.46 ± 0.49 | 3.92 | 50.59 |

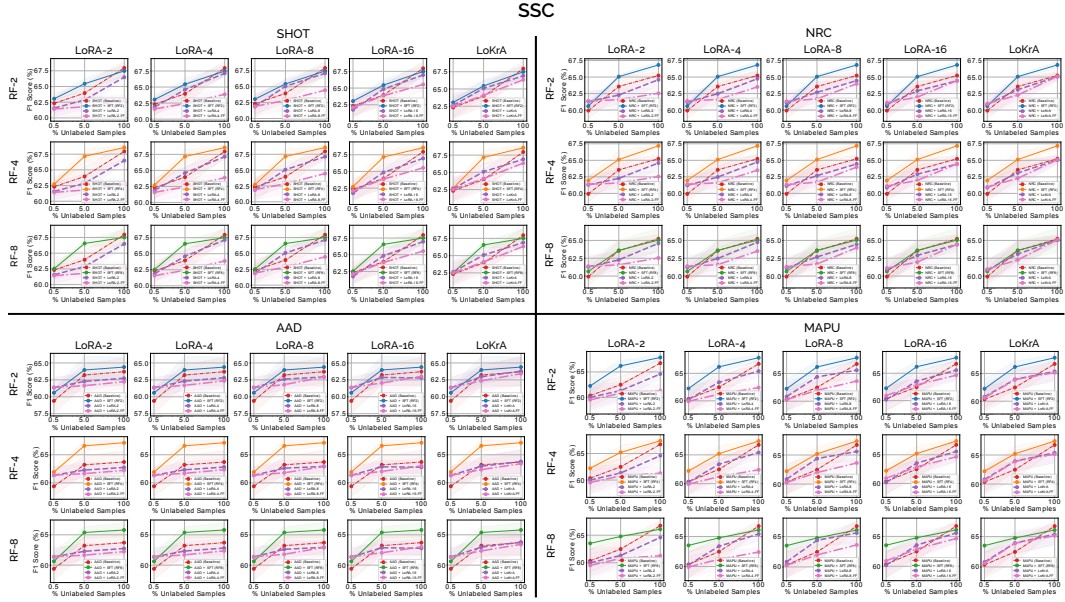

Figure 13: Comparison of the vanilla LoRA-style PEFT method, fine-tuning all components (in Purple) and fine-tuning while freezing the randomly initialized factors (in Magenta), against baseline methods (SHOT, NRC, AAD, MAPU) and our proposed approach (SFT) at different $RF$ values on the SSC dataset.

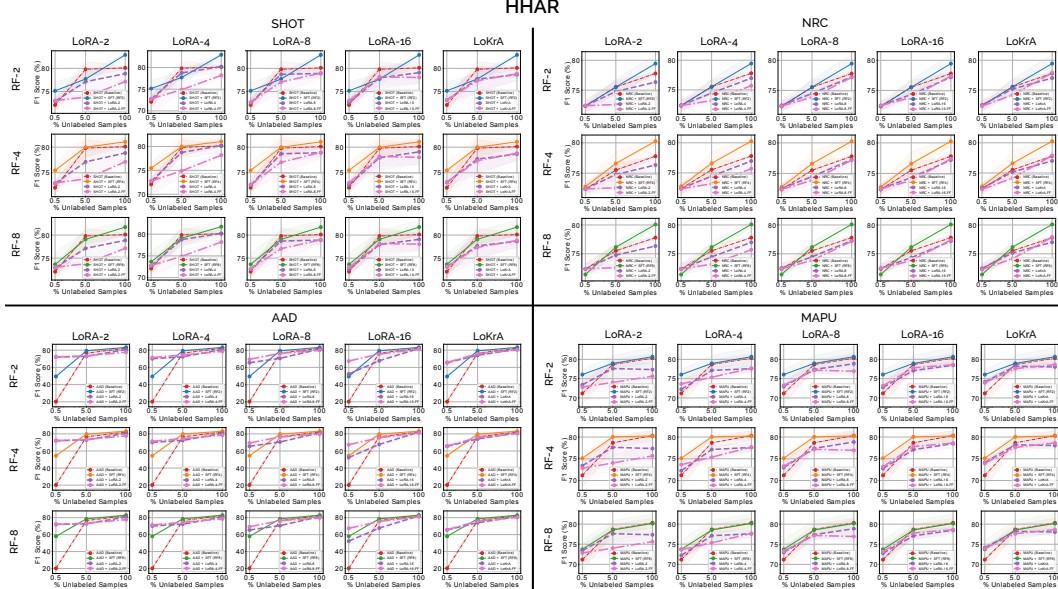

Figure 14: Comparison of the vanilla LoRA-style PEFT method, fine-tuning all components (in Purple) and fine-tuning while freezing the randomly initialized factors (in Magenta), against baseline methods (SHOT, NRC, AAD, MAPU) and our proposed approach (SFT) at different $RF$ values on the HHAR dataset.

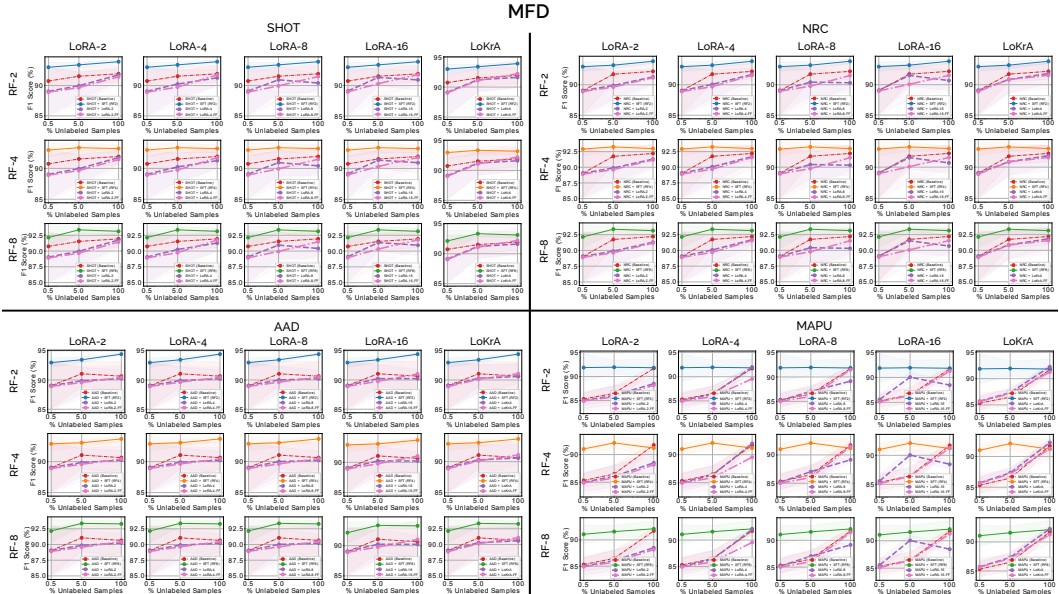

Figure 15: Comparison of the vanilla LoRA-style PEFT method, fine-tuning all components (in Purple) and fine-tuning while freezing the randomly initialized factors (in Magenta), against baseline methods (SHOT, NRC, AAD, MAPU) and our proposed approach (SFT) at different $RF$ values on the MFD dataset.

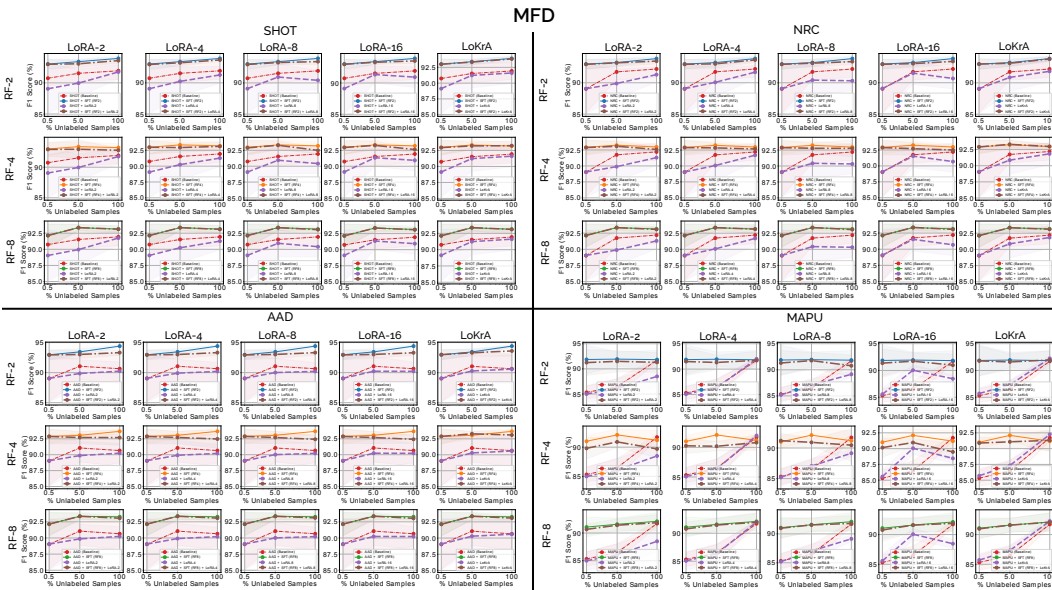

Figure 16: Performance comparison of SFT with LoRA-style PEFT methods on the MFD dataset at different sample ratio of target samples used for finetuning, on SHOT, NRC, AAD, and MAPU. The results highlight the trade-offs between parameter efficiency and predictive performance across different adaptation methods.

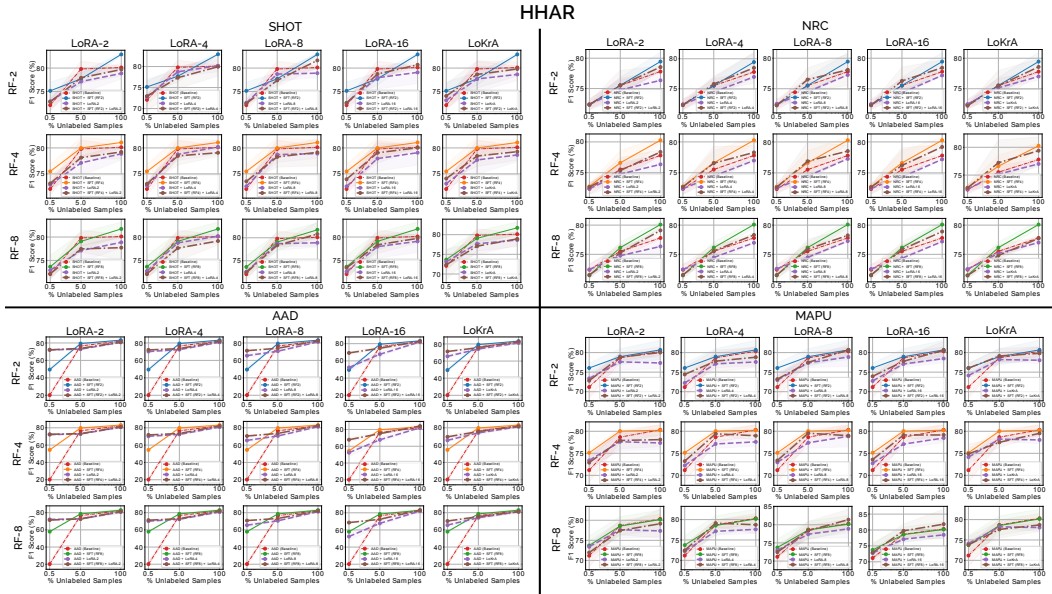

Figure 17: Performance comparison of SFT with LoRA-style PEFT methods on the HHAR dataset at different sample ratio of target samples used for finetuning, on SHOT, NRC, AAD, and MAPU. The results highlight the trade-offs between parameter efficiency and predictive performance across different adaptation methods.

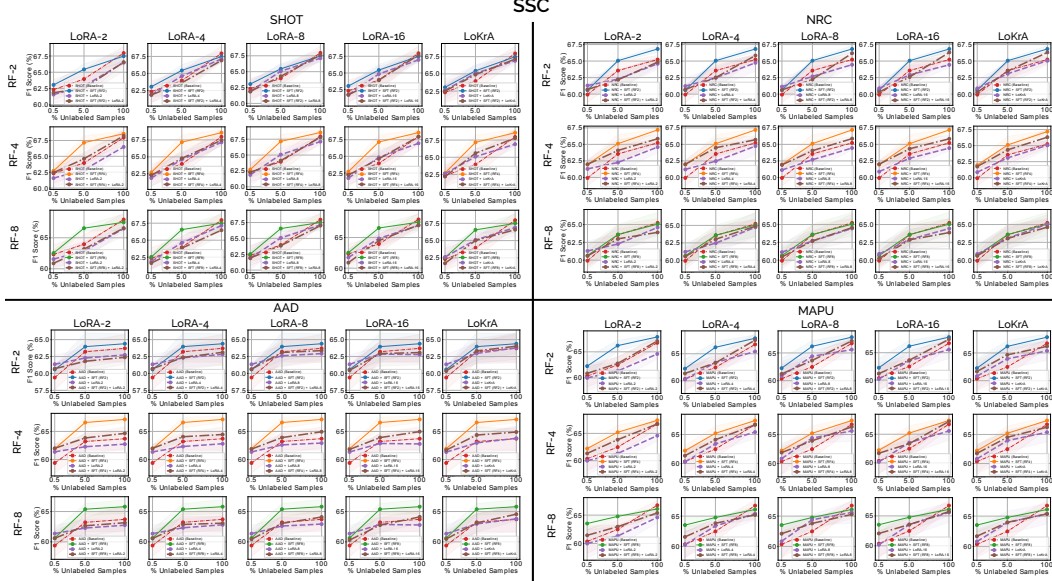

Figure 18: Performance comparison of SFT with LoRA-style PEFT methods on the SSC dataset at different sample ratio of target samples used for finetuning, on SHOT, NRC, AAD, and MAPU. The results highlight the trade-offs between parameter efficiency and predictive performance across different adaptation methods.

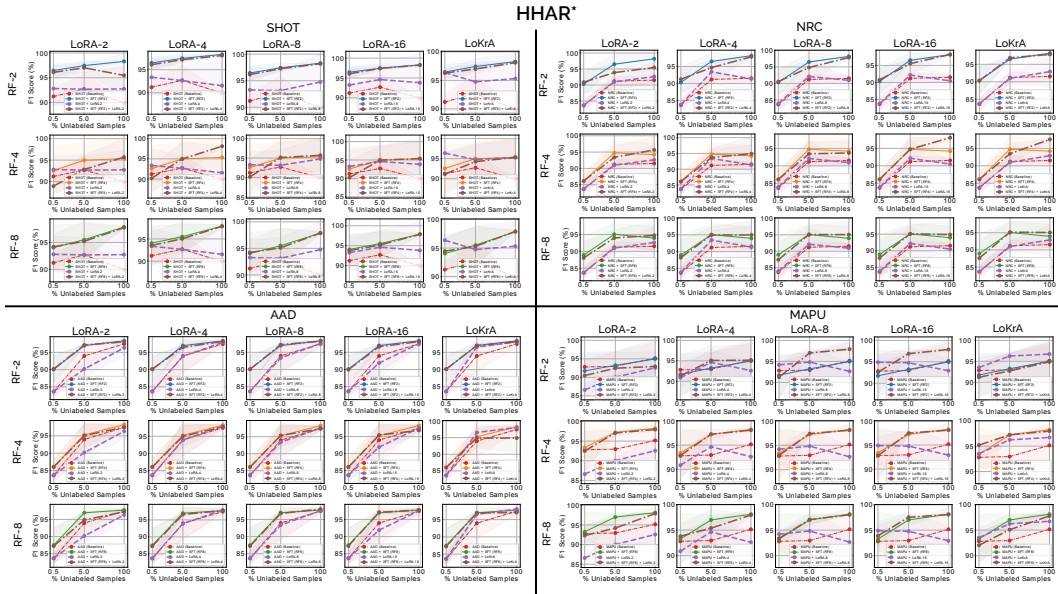

Figure 19: Performance comparison of SFT with LoRA-style PEFT methods on the HHAR dataset, for the Many-to-one setting (noted with an asterisk (*)), at different sample ratio of target samples used for finetuning, on SHOT, NRC, AAD, and MAPU. The results highlight the trade-offs between parameter efficiency and predictive performance across different adaptation methods.

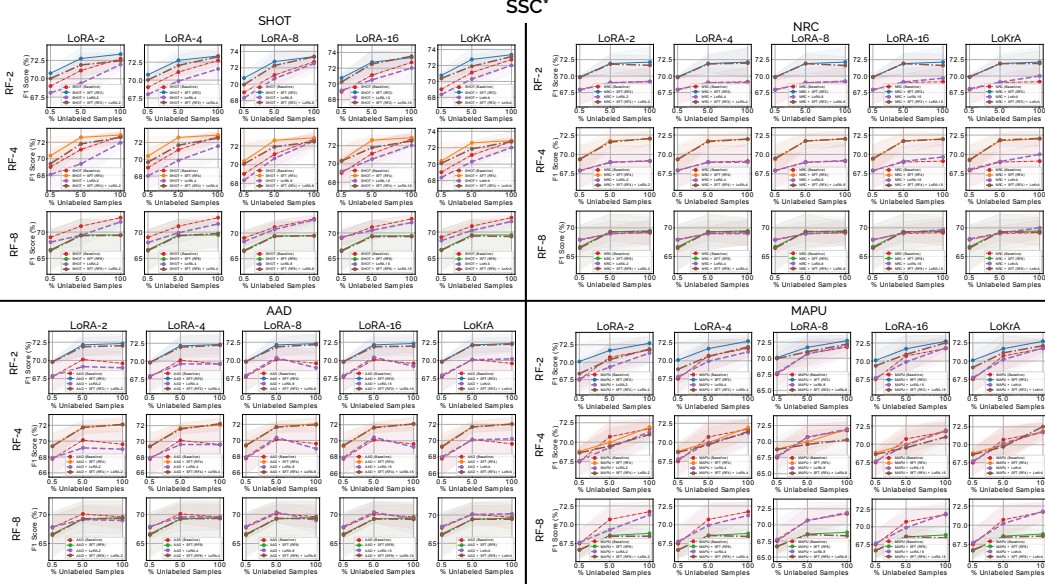

Figure 20: Performance comparison of SFT with LoRA-style PEFT methods on the SSC dataset, for the Many-to-one setting (noted with an asterisk (*)), at different sample ratio of target samples used for finetuning, on SHOT, NRC, AAD, and MAPU. The results highlight the trade-offs between parameter efficiency and predictive performance across different adaptation methods.

Table 10: Comparison of inference overhead (measured in MACs) and the number of parameters fine-tuned during adaptation (# Params.) for various combinations of SFT and LoRA-style PEFT methods. The table highlights the trade-offs between parameter efficiency and computational cost across different hybrid configurations.

| Method | MFD | | HHAR/HHAR* | | SSC/SSC* | |
|---|---|---|---|---|---|---|
| | MACs (M) $\downarrow$ | # Params. (K ) $\downarrow$ | MACs (M) $\downarrow$ | # Params. (K) $\downarrow$ | MACs (M) $\downarrow$ | # Params. (K) $\downarrow$ |
| Baseline | 58.18 | 199.3 | 9.04 | 198.21 | 12.92 | 83.17 |
| SFT (RF8) | 3.52 | 3.33 | 0.53 | 3.19 | 0.80 | 1.38 |
| SFT (RF8) + LoKrA | 3.52 | 0.41 | 0.53 | 0.34 | 0.80 | 0.26 |
| SFT (RF8) + LoRA-2 | 3.52 | 0.32 | 0.53 | 0.22 | 0.80 | 0.26 |
| SFT (RF8) + LoRA-4 | 3.52 | 1.03 | 0.53 | 0.60 | 0.80 | 0.82 |
| SFT (RF8) + LoRA-8 | 3.52 | 3.59 | 0.53 | 1.88 | 0.80 | 2.95 |
| SFT (RF8) + LoRA-16 | 3.52 | 13.33 | 0.53 | 6.45 | 0.80 | 11.15 |
| SFT (RF4) | 8.80 | 12.80 | 1.34 | 12.53 | 1.99 | 5.32 |
| SFT (RF4) + LoKrA | 8.80 | 0.93 | 1.34 | 0.86 | 1.99 | 0.51 |
| SFT (RF4) + LoRA-2 | 8.80 | 0.45 | 1.34 | 0.35 | 1.99 | 0.33 |
| SFT (RF4) + LoRA-4 | 8.80 | 1.28 | 1.34 | 0.86 | 1.99 | 0.98 |
| SFT (RF4) + LoRA-8 | 8.80 | 4.10 | 1.34 | 2.39 | 1.99 | 3.27 |
| SFT (RF2) + LoRA-16 | 8.80 | 14.35 | 1.34 | 7.47 | 1.99 | 11.79 |
| SFT (RF2) | 24.66 | 50.18 | 3.79 | 49.63 | 5.54 | 20.88 |
| SFT (RF2) + LoKrA | 24.66 | 1.38 | 3.79 | 1.24 | 5.54 | 0.92 |
| SFT (RF2) + LoRA-2 | 24.66 | 0.71 | 3.79 | 0.60 | 5.54 | 0.49 |
| SFT (RF2) + LoRA-4 | 24.66 | 1.80 | 3.79 | 1.37 | 5.54 | 1.30 |
| SFT (RF2) + LoRA-8 | 24.66 | 5.13 | 3.79 | 3.42 | 5.54 | 3.91 |
| SFT (RF2) + LoRA-16 | 24.66 | 16.40 | 3.79 | 9.52 | 5.54 | 13.07 |
| LoKrA | 58.18 | 3.46 | 9.04 | 3.33 | 12.92 | 3.46 |
| LoRA-2 | 58.18 | 1.22 | 9.04 | 1.11 | 12.92 | 1.22 |
| LoRA-4 | 58.18 | 2.82 | 9.04 | 2.40 | 12.92 | 2.82 |
| LoRA-8 | 58.18 | 7.18 | 9.04 | 5.46 | 12.92 | 7.18 |
| LoRA-16 | 58.18 | 20.5 | 9.04 | 13.62 | 12.92 | 20.50 |

Table 11: Ablation studies on finetuning different parameter subspaces during target-side adaptation for SHOT. The asterisk (*) on top of the dataset names denote the Many-to-one setting (*cf*. Appendix A.9 for reference).

| Method | RF | Parameters Finetuned | SSC | | | SSC* | | | HHAR | | | HHAR* | | |
|---|---|---|---|---|---|---|---|---|---|---|---|---|---|---|
| | | | F1 Score (%) $\uparrow$ | MACs (M) $\downarrow$ | # Params. (K) $\downarrow$ | F1 Score (%) $\uparrow$ | MACs (M) $\downarrow$ | # Params. (K) $\downarrow$ | F1 Score (%) $\uparrow$ | MACs (M) $\downarrow$ | # Params. (K) $\downarrow$ | F1 Score (%) $\uparrow$ | MACs (M) $\downarrow$ | # Params. (K) $\downarrow$ |
| No Adaptation | - | None | 56.87 ± 1.24 | 12.92 | 0 | 63.19 ± 0.93 | 12.92 | 0 | 67.41 ± 0.11 | 9.04 | 0 | 79.03 ± 1.22 | 9.04 | 0 |
| SHOT | - | Entire Backbone | 67.95 ± 1.04 | 12.92 | 83.17 | 72.74 ± 0.60 | 12.92 | 83.17 | 80.12 ± 0.29 | 9.04 | 198.21 | 89.94 ± 0.34 | 9.04 | 198.21 |
| SHOT | - | BN | 63.26 ± 0.97 | 12.92 | 0.45 | 68.98 ± 2.16 | 12.92 | 0.45 | 72.43 ± 0.83 | 9.04 | 0.64 | 84.85 ± 4.25 | 9.04 | 0.64 |
| | 8 | Factor Matrices ($\mathbf{V}^{(1)}$, $\mathbf{V}^{(2)}$) | 67.00 ± 0.06 | 0.80 | 3.33 | 70.25 ± 2.16 | 0.80 | 3.33 | 81.78 ± 0.92 | 0.53 | 7.18 | 95.55 ± 3.57 | 0.53 | 7.18 |
| | 8 | Core Tensor ($\mathcal{T}$) | 67.50 ± 1.33 | 0.80 | 1.38 | 69.45 ± 2.51 | 0.80 | 1.38 | 81.73 ± 1.47 | 0.53 | 3.19 | 97.88 ± 0.20 | 0.53 | 3.19 |
| | 8 | Factor Matrices ($\mathbf{V}^{(1)}$, $\mathbf{V}^{(2)}$) + Core Tensor ($\mathcal{T}$) | 66.97 ± 0.69 | 0.80 | 4.71 | 70.18 ± 2.15 | 0.80 | 4.71 | 81.5 ± 1.33 | 0.53 | 10.37 | 95.35 ± 4.88 | 0.53 | 10.37 |
| | 4 | Factor Matrices ($\mathbf{V}^{(1)}$, $\mathbf{V}^{(2)}$) | 68.17 ± 0.33 | 1.99 | 6.66 | 73.07 ± 0.53 | 1.99 | 6.66 | 81.58 ± 0.88 | 1.34 | 14.35 | 95.37 ± 4.88 | 1.34 | 14.35 |
| SHOT + SFT | 4 | Core Tensor ($\mathcal{T}$) | 68.56 ± 0.44 | 1.99 | 5.32 | 72.87 ± 0.33 | 1.99 | 5.32 | 81.05 ± 0.45 | 1.34 | 12.53 | 95.34 ± 4.82 | 1.34 | 12.53 |
| | 4 | Factor Matrices ($\mathbf{V}^{(1)}$, $\mathbf{V}^{(2)}$) + Core Tensor ($\mathcal{T}$) | 67.57 ± 0.74 | 1.99 | 11.98 | 73.09 ± 0.46 | 1.99 | 11.98 | 81.46 ± 0.91 | 1.34 | 26.88 | 93.68 ± 6.82 | 1.34 | 26.88 |
| | 2 | Factor Matrices ($\mathbf{V}^{(1)}$, $\mathbf{V}^{(2)}$) | 66.96 ± 0.33 | 5.54 | 13.31 | 74.19 ± 0.94 | 5.54 | 13.31 | 81.82 ± 1.01 | 3.79 | 28.68 | 98.30 ± 0.06 | 3.79 | 28.68 |
| | 2 | Core Tensor ($\mathcal{T}$) | 67.48 ± 0.89 | 5.54 | 20.88 | 73.36 ± 0.66 | 5.54 | 20.88 | 82.92 ± 0.27 | 3.79 | 49.63 | 98.30 ± 0.06 | 3.79 | 49.63 |
| | 2 | Factor Matrices ($\mathbf{V}^{(1)}$, $\mathbf{V}^{(2)}$) + Core Tensor ($\mathcal{T}$) | 66.29 ± 1.25 | 5.54 | 34.19 | 73.84 ± 0.77 | 5.54 | 34.19 | 82.63 ± 0.01 | 3.79 | 78.31 | 98.30 ± 0.11 | 3.79 | 78.31 |

Table 12: Ablation studies on finetuning different parameter subspaces during target-side adaptation for NRC. The asterisk (*) on top of the dataset names denote the Many-to-one setting (*cf*. Appendix A.9 for reference).

| Method | RF | Parameters Finetuned | SSC | | | SSC* | | | HHAR | | | HHAR* | | |
|---|---|---|---|---|---|---|---|---|---|---|---|---|---|---|
| | | | F1 Score (%) $\uparrow$ | MACs (M) $\downarrow$ | # Params. (K) $\downarrow$ | F1 Score (%) $\uparrow$ | MACs (M) $\downarrow$ | # Params. (K) $\downarrow$ | F1 Score (%) $\uparrow$ | MACs (M) $\downarrow$ | # Params. (K) $\downarrow$ | F1 Score (%) $\uparrow$ | MACs (M) $\downarrow$ | # Params. (K) $\downarrow$ |
| No Adaptation | - | None | 56.87 ± 1.24 | 12.92 | 0 | 63.19 ± 0.93 | 12.92 | 0 | 67.41 ± 0.11 | 9.04 | 0 | 79.03 ± 1.22 | 9.04 | 0 |
| NRC | - | Entire Backbone | 65.23 ± 0.59 | 12.92 | 83.17 | 69.14 ± 2.24 | 12.92 | 83.17 | 77.80 ± 0.16 | 9.04 | 198.21 | 91.53 ± 0.92 | 9.04 | 198.21 |
| NRC | - | BN | 63.22 ± 0.92 | 12.92 | 0.45 | 68.86 ± 2.37 | 12.92 | 0.45 | 72.51 ± 0.81 | 9.04 | 0.64 | 85.84 ± 4.09 | 9.04 | 0.64 |
| | 8 | Factor Matrices ($\mathbf{V}^{(1)}$, $\mathbf{V}^{(2)}$) | 65.15 ± 1.39 | 0.80 | 3.33 | 69.40 ± 3.27 | 0.80 | 3.33 | 79.32 ± 0.98 | 0.53 | 7.18 | 95.64 ± 4.41 | 0.53 | 7.18 |
| | 8 | Core Tensor ($\mathcal{T}$) | 65.05 ± 1.66 | 0.80 | 1.38 | 69.47 ± 2.38 | 0.80 | 1.38 | 80.09 ± 0.56 | 0.53 | 3.19 | 93.94 ± 3.49 | 0.53 | 3.19 |
| | 8 | Factor Matrices ($\mathbf{V}^{(1)}$, $\mathbf{V}^{(2)}$) + Core Tensor ($\mathcal{T}$) | 65.12 ± 1.44 | 0.80 | 4.71 | 69.45 ± 2.88 | 0.80 | 4.71 | 79.41 ± 1.02 | 0.53 | 10.37 | 94.02 ± 3.25 | 0.53 | 10.37 |
| | 4 | Factor Matrices ($\mathbf{V}^{(1)}$, $\mathbf{V}^{(2)}$) | 66.80 ± 0.60 | 1.99 | 6.66 | 72.10 ± 0.79 | 1.99 | 6.66 | 80.07 ± 1.08 | 1.34 | 14.35 | 94.17 ± 3.46 | 1.34 | 14.35 |
| NRC + SFT | 4 | Core Tensor ($\mathcal{T}$) | 67.19 ± 0.14 | 1.99 | 5.32 | 71.99 ± 0.88 | 1.99 | 5.32 | 80.27 ± 0.55 | 1.34 | 12.53 | 94.22 ± 3.39 | 1.34 | 12.53 |
| | 4 | Factor Matrices ($\mathbf{V}^{(1)}$, $\mathbf{V}^{(2)}$) + Core Tensor ($\mathcal{T}$) | 66.49 ± 0.43 | 1.99 | 11.98 | 72.05 ± 0.88 | 1.99 | 11.98 | 80.26 ± 0.71 | 1.34 | 26.88 | 94.04 ± 7.20 | 1.34 | 26.88 |
| | 2 | Factor Matrices ($\mathbf{V}^{(1)}$, $\mathbf{V}^{(2)}$) | 66.18 ± 0.42 | 5.54 | 13.31 | 71.89 ± 1.38 | 5.54 | 13.31 | 79.72 ± 0.64 | 3.79 | 28.68 | 98.17 ± 0.07 | 3.79 | 28.68 |
| | 2 | Core Tensor ($\mathcal{T}$) | 66.83 ± 0.51 | 5.54 | 20.88 | 72.12 ± 2.26 | 5.54 | 20.88 | 79.45 ± 1.51 | 3.79 | 49.63 | 98.19 ± 0.05 | 3.79 | 49.63 |
| | 2 | Factor Matrices ($\mathbf{V}^{(1)}$, $\mathbf{V}^{(2)}$) + Core Tensor ($\mathcal{T}$) | 66.84 ± 0.04 | 5.54 | 34.19 | 72.05 ± 0.88 | 5.54 | 34.19 | 79.64 ± 1.33 | 3.79 | 78.31 | 95.99 ± 3.53 | 3.79 | 78.31 |

Table 13: Ablation studies on finetuning different parameter subspaces during target-side adaptation for AAD. The asterisk (*) on top of the dataset names denote the Many-to-one setting (*cf*. Appendix A.9 for reference).

| Method | RF | Parameters Finetuned | SSC | | | SSC* | | | HHAR | | | HHAR* | | |
|---|---|---|---|---|---|---|---|---|---|---|---|---|---|---|
| | | | F1 Score (%) $\uparrow$ | MACs (M) $\downarrow$ | # Params. (K) $\downarrow$ | F1 Score (%) $\uparrow$ | MACs (M) $\downarrow$ | # Params. (K) $\downarrow$ | F1 Score (%) $\uparrow$ | MACs (M) $\downarrow$ | # Params. (K) $\downarrow$ | F1 Score (%) $\uparrow$ | MACs (M) $\downarrow$ | # Params. (K) $\downarrow$ |
| No Adaptation | - | None | 56.87 ± 1.24 | 12.92 | 0 | 63.19 ± 0.93 | 12.92 | 0 | 67.41 ± 0.11 | 9.04 | 0 | 79.03 ± 1.22 | 9.04 | 0 |
| AAD | - | Entire Backbone | 63.71 ± 2.06 | 12.92 | 83.17 | 69.60 ± 2.05 | 12.92 | 83.17 | 82.25 ± 1.14 | 9.04 | 198.21 | 97.45 ± 0.84 | 9.04 | 198.21 |
| AAD | - | BN | 62.83 ± 0.93 | 12.92 | 0.45 | 69.01 ± 2.13 | 12.92 | 0.45 | 72.35 ± 0.90 | 9.04 | 0.64 | 84.12 ± 3.44 | 9.04 | 0.64 |
| | 8 | Factor Matrices ($\mathbf{V}^{(1)}$, $\mathbf{V}^{(2)}$) | 65.57 ± 1.05 | 0.80 | 3.33 | 69.08 ± 2.37 | 0.80 | 3.33 | 83.46 ± 1.26 | 0.53 | 7.18 | 97.93 ± 0.55 | 0.53 | 7.18 |
| | 8 | Core Tensor ($\mathcal{T}$) | 65.80 ± 1.17 | 0.80 | 1.38 | 69.47 ± 2.88 | 0.80 | 1.38 | 82.92 ± 1.66 | 0.53 | 3.19 | 97.74 ± 0.52 | 0.53 | 3.19 |
| | 8 | Factor Matrices ($\mathbf{V}^{(1)}$, $\mathbf{V}^{(2)}$) + Core Tensor ($\mathcal{T}$) | 65.36 ± 1.16 | 0.80 | 4.71 | 68.45 ± 1.98 | 0.80 | 4.71 | 82.99 ± 1.94 | 0.53 | 10.37 | 97.92 ± 0.55 | 0.53 | 10.37 |
| | 4 | Factor Matrices ($\mathbf{V}^{(1)}$, $\mathbf{V}^{(2)}$) | 66.46 ± 0.60 | 1.99 | 6.66 | 71.96 ± 1.28 | 1.99 | 6.66 | 82.51 ± 0.88 | 1.34 | 14.35 | 97.56 ± 0.61 | 1.34 | 14.35 |
| AAD + SFT | 4 | Core Tensor ($\mathcal{T}$) | 67.14 ± 0.57 | 1.99 | 5.32 | 72.08 ± 0.79 | 1.99 | 5.32 | 83.15 ± 0.88 | 1.34 | 12.53 | 98.25 ± 0.29 | 1.34 | 12.53 |
| | 4 | Factor Matrices ($\mathbf{V}^{(1)}$, $\mathbf{V}^{(2)}$) + Core Tensor ($\mathcal{T}$) | 66.70 ± 0.45 | 1.99 | 11.98 | 71.69 ± 1.12 | 1.99 | 11.98 | 82.89 ± 1.45 | 1.34 | 26.88 | 97.57 ± 0.67 | 1.34 | 26.88 |
| | 2 | Factor Matrices ($\mathbf{V}^{(1)}$, $\mathbf{V}^{(2)}$) | 63.57 ± 2.12 | 5.54 | 13.31 | 72.12 ± 1.59 | 5.54 | 13.31 | 82.43 ± 0.59 | 3.79 | 28.68 | 98.28 ± 0.14 | 3.79 | 28.68 |
| | 2 | Core Tensor ($\mathcal{T}$) | 64.39 ± 1.45 | 5.54 | 20.88 | 72.32 ± 1.20 | 5.54 | 20.88 | 83.25 ± 0.34 | 3.79 | 49.63 | 98.36 ± 0.36 | 3.79 | 49.63 |
| | 2 | Factor Matrices ($\mathbf{V}^{(1)}$, $\mathbf{V}^{(2)}$) + Core Tensor ($\mathcal{T}$) | 64.16 ± 2.41 | 5.54 | 34.19 | 72.17 ± 1.72 | 5.54 | 34.19 | 83.15 ± 0.03 | 3.79 | 78.31 | 98.26 ± 0.05 | 3.79 | 78.31 |

Table 14: Ablation Studies on finetuning different parameter subspaces during target-side adaptation on MAPU. The asterisk (*) on top of the dataset names denote the Many-to-one setting (*cf*. Appendix A.9 for reference).

| Method | RF | Parameters Finetuned | SSC F1 Score (%)↑ | SSC MACs (M)↓ | SSC # Params. (K)↓ | SSC* F1 Score (%)↑ | SSC* MACs (M)↓ | SSC* # Params. (K)↓ | HHAR F1 Score (%)↑ | HHAR MACs (M)↓ | HHAR # Params. (K)↓ | HHAR* F1 Score (%)↑ | HHAR* MACs (M)↓ | HHAR* # Params. (K)↓ |
|---|---|---|---|---|---|---|---|---|---|---|---|---|---|---|
| No Adaptation | - | None | 54.96 ± 0.87 | 12.92 | 0 | 63.94 ± 0.84 | 12.92 | 0 | 66.67 ± 1.03 | 9.04 | 0 | 82.88 ± 1.30 | 9.04 | 0 |
| MAPU | - | Entire Backbone | 66.73 ± 0.85 | 12.92 | 83.17 | 71.74 ± 1.30 | 12.92 | 83.17 | 80.32 ± 1.16 | 9.04 | 198.21 | 95.16 ± 4.29 | 9.04 | 198.21 |
| MAPU | - | BN | 61.07 ± 1.58 | 12.92 | 0.45 | 69.17 ± 2.70 | 12.92 | 0.45 | 71.40 ± 1.63 | 9.04 | 0.64 | 86.15 ± 3.71 | 9.04 | 0.64 |
| MAPU + SFT | 8 | Factor Matrices ($\mathbf{V}^{(1)}$, $\mathbf{V}^{(2)}$) | 66.05 ± 0.75 | 0.80 | 3.33 | 68.73 ± 1.22 | 0.80 | 3.33 | 80.42 ± 1.51 | 0.53 | 7.18 | 98.00 ± 0.04 | 0.53 | 7.18 |
|  |  | Core Tensor ($\mathcal{T}$) | 66.09 ± 0.19 | 0.80 | 1.38 | 68.85 ± 1.30 | 0.80 | 1.38 | 80.16 ± 2.38 | 0.53 | 3.19 | 98.12 ± 0.04 | 0.53 | 3.19 |
|  |  | Factor Matrices ($\mathbf{V}^{(1)}$, $\mathbf{V}^{(2)}$) + Core Tensor ($\mathcal{T}$) | 66.17 ± 0.53 | 0.80 | 4.71 | 68.58 ± 0.96 | 0.80 | 4.71 | 80.29 ± 1.04 | 0.53 | 10.37 | 98.06 ± 0.05 | 0.53 | 10.37 |
|  | 4 | Factor Matrices ($\mathbf{V}^{(1)}$, $\mathbf{V}^{(2)}$) | 67.60 ± 0.07 | 1.99 | 6.66 | 72.08 ± 1.48 | 1.99 | 6.66 | 79.88 ± 1.20 | 1.34 | 14.35 | 98.19 ± 0.10 | 1.34 | 14.35 |
|  |  | Core Tensor ($\mathcal{T}$) | 67.40 ± 0.59 | 1.99 | 5.32 | 71.94 ± 1.10 | 1.99 | 5.32 | 80.24 ± 1.22 | 1.34 | 12.53 | 98.25 ± 0.03 | 1.34 | 12.53 |
|  |  | Factor Matrices ($\mathbf{V}^{(1)}$, $\mathbf{V}^{(2)}$) + Core Tensor ($\mathcal{T}$) | 67.82 ± 0.21 | 1.99 | 11.98 | 71.52 ± 0.94 | 1.99 | 11.98 | 79.63 ± 1.08 | 1.34 | 26.88 | 98.20 ± 0.08 | 1.34 | 26.88 |
|  | 2 | Factor Matrices ($\mathbf{V}^{(1)}$, $\mathbf{V}^{(2)}$) | 67.13 ± 0.69 | 5.54 | 13.31 | 72.84 ± 1.15 | 5.54 | 13.31 | 80.51 ± 2.41 | 3.79 | 28.68 | 95.06 ± 5.17 | 3.79 | 28.68 |
|  |  | Core Tensor ($\mathcal{T}$) | 67.85 ± 0.62 | 5.54 | 20.88 | 72.73 ± 0.83 | 5.54 | 20.88 | 80.69 ± 2.45 | 3.79 | 49.63 | 94.96 ± 5.18 | 3.79 | 49.63 |
|  |  | Factor Matrices ($\mathbf{V}^{(1)}$, $\mathbf{V}^{(2)}$) + Core Tensor ($\mathcal{T}$) | 67.37 ± 0.36 | 5.54 | 34.19 | 72.93 ± 0.67 | 5.54 | 34.19 | 80.69 ± 1.26 | 3.79 | 78.31 | 95.02 ± 5.11 | 3.79 | 78.31 |

