# OpenReview forum: "Efficient Source-Free Time-Series Adaptation via Parameter Subspace Disentanglement"
_ICLR.cc/2025/Conference — ICLR 2025 Poster_

### Official Review · Reviewer_G1Yq · 2024-10-23

**Soundness:** 3
**Presentation:** 3
**Contribution:** 1
**Rating:** 5
**Confidence:** 3

**Summary:**

This paper investigates source-free domain adaptation problem. The authors introduce a straightforward yet efficient framework that decomposes the parameter matrices using Tucker-style factorization, allowing the model to adjust only a subset of parameters during target adaptation. Extensive experiments conducted across various datasets and settings demonstrate the superiority of the proposed method over existing baseline models and high sample efficiency.

**Strengths:**

1. The proposed metric decomposition-based approach for efficient target domain adaptation is simple and efficient.
2. The experiments are very detailed and thorough, clearly demonstrating the effectiveness and efficiency of the proposed method. That is appreciated.
3. The manuscript is well-structured and clearly written, facilitating easy comprehension for readers.

**Weaknesses:**

1. The connection between the proposed method and the characteristics of time-series data is somewhat weak. The method appears to be a general SFDA approach rather than being specifically tailored for time-series data.
2. The novelty of the approach is limited, as parameter decomposition for domain adaptation is a prevalent technique.
3. The description of target domain adaptation is insufficient. Specifically, which loss function is utilised? And how to optimise the parameters?

**Questions:**

1. Could the author explain the rationality of replacing $P$ (the prior distribution) and $S$ (sampled data) with $\mathcal{W}{src}$ (the pre-trained network) and $\mathcal{W}_{trg}$ (fine-tuned network), respectively, in the context of PAC-Bayesian generalization theory? The original theory is designed to measure the generalization error between the empirical and expect loss, but for the same distribution.
2. Could the author include an ERM baseline in Table 1 to provide readers with a clearer understanding of the performance improvements achieved by the proposed method?

---

> ### Author Response · Authors · 2024-11-21
> **Official Response to Reviewer G1Yq**
>
> We sincerely appreciate the reviewer’s recognition of the simplicity and efficiency of our proposed metric decomposition-based approach for target domain adaptation. We are grateful for the acknowledgment of our detailed and thorough experiments, which clearly demonstrate the effectiveness and efficiency of the method. Additionally, we thank the reviewer for appreciating the well-structured and clearly written manuscript, which facilitates easy comprehension for readers. Below we provide the responses to the concerns raised by the reviewer. All the changes in the revised version of the manuscript are highlighted in ${\color{blue}\text{Blue}}$.
>
> ---
> ### Reviewer Weakness 1
>
> >The description of target domain adaptation is insufficient. Specifically, which loss function is utilised? And how to optimise the parameters?
>
> ### Author Response
>
> We thank the reviewer for pointing out the need for a more detailed description of the target domain adaptation process. In response to this suggestion, we have included a comprehensive explanation of the loss functions and adaptation objectives in the updated manuscript. Specifically, the details regarding the adaptation objectives, including the precise loss functions and the parameter optimization process, have been added in **Section A.8** of the appendix. This updated section outlines the mathematical formulation of the adaptation losses. We hope this additional information addresses the reviewer’s concern and provides greater clarity on our adaptation methodology.
>
> ---
> ### Reviewer Weakness 2
> > The novelty of the approach is limited, as parameter decomposition for domain adaptation is a prevalent technique.
>
> ### Author Reponse
>
> We appreciate the reviewer’s feedback . However, based on our extensive survey of the literature and works discussed, we are not aware of any prior works that leverage parameter decomposition for DA. Specifically, to the best of our knowledge, no existing work employs parameter decomposition in conjunction with selective fine-tuning to achieve both parameter and sample efficiency in domain adaptation.
>
> If there are relevant studies or references that the reviewer is aware of, we would be grateful if they could direct us to these works, as it would help further contextualize our contributions. In our view, the primary novelty of our approach lies in demonstrating that parameter decomposition, when combined with selective fine-tuning, can simultaneously address the dual challenges of parameter and sample efficiency—a unique and compelling insight.
>
> We respectfully request clarification and supporting references for the reviewer’s claim that parameter decomposition is a widely used method in domain adaptation. This would greatly assist us in strengthening the discussion in our manuscript.
>
> ---
>
> ### Reviewer Weakness 3
> > *The connection between the proposed method and the characteristics of time-series data is somewhat weak. The method appears to be a general SFDA approach rather than being specifically tailored for time-series data.*
>
> ### Author Response
> We acknowledge that the connection between our proposed method and the characteristics of time-series data may not have been sufficiently explicit. Our primary motivation was to address the challenges of parameter and sample efficiency in Source-Free Domain Adaptation (SFDA) for time-series data—a pressing research gap we identified. In our analysis (see **Figure 2A**), we observed significant parameter redundancy along the channel dimension in source models, indicating strong inter-channel dependencies among variables in time-series features as they propagate through deep networks.
>
> To leverage these characteristics, we proposed decomposing the model parameters along the channel dimension (**Equation 3**) and focusing on the principal channels or variables. This approach exploits the inherent inter-variable dependencies in time-series data, achieving parameter efficiency without sacrificing performance by concentrating on the most significant channels.
>
> Our goal was to develop a method applicable across various SFDA settings and diverse time-series datasets. Therefore, we deliberately avoided introducing specific inductive biases. We conducted extensive experiments with both general SFDA methods (SHOT, NRC, AAD) and a time-series-specific method (MAPU), following Ragab et al [1]. We also expanded our empirical evaluation to all five datasets from the AdaTime benchmark [2]—unlike previous works that typically used only three—and tested ten source-target pairs, including a many-to-one adaptation setting.
>
> **References:**
>
> [1] Ragab, Mohamed, et al. "Source-free domain adaptation with temporal imputation for time series data." Proceedings of the 29th ACM SIGKDD Conference on Knowledge Discovery and Data Mining. 2023.\
> [2] Ragab, Mohamed, et al. "Adatime: A benchmarking suite for domain adaptation on time series data." ACM Transactions on Knowledge Discovery from Data 17.8 (2023): 1-18.

---

> ### Author Response · Authors · 2024-11-21
> **Official Response to Reviewer G1Yq (contd.)**
>
> ### Reviewer Question 1
> > Could the author explain the rationality of replacing $P$ (the prior distribution) and $S$ (sampled data) with $\mathcal{W}\_{src}$ (the pre-trained network) and $\mathcal{W}\_{trg}$ (fine-tuned network), respectively, in the context of PAC-Bayesian generalization theory? The original theory is designed to measure the generalization error between the empirical and expect loss, but for the same distribution.
>
> ### Author Response
>
> Thank you for raising this critical point of clarification. It is absolutely correct that the PAC-Bayesian generalization theory is traditionally designed to measure the generalization error between the empirical and expected losses for the same distribution.
> In our context, the PAC-Bayesian bound we provide pertains specifically to the target adaptation step, which solely operates on the target distribution. This ensures compliance with the "same distribution" constraint inherent in PAC-Bayesian theory.
> The validity of this approach stems from the fact that, by the time the target adaptation step begins, we have already pre-trained the network on the source distribution and have access to the pre-trained source weights. These source weights serve as a prior in the adaptation phase, enabling the PAC-Bayesian bound to operate effectively on the target distribution alone.
>
> ---
> ### Reviewer Question 2
>
> >Could the author include an ERM baseline in Table 1 to provide readers with a clearer understanding of the performance improvements achieved by the proposed method?
>
> ### Author Response
> We sincerely thank the reviewer for the thoughtful suggestion to include an ERM baseline in Table 1 to provide readers with a clearer understanding of the performance improvements achieved by our proposed method.
>
> We would like to bring to the reviewer's attention that the ERM baseline for the target performance of the source-pretrained model is already included in our manuscript. This is presented in the ablation analysis in **Tables 2, 11, 12, and 13** under the row corresponding to the "No-Adaptation" method. If this aligns with what the reviewer is referring to, we would be happy to incorporate these results into Table 1 for consistency and easier reference.
>
> However, if the reviewer is suggesting a different or additional ERM baseline, we kindly request further clarification so that we can address the feedback appropriately. We greatly appreciate your input and are eager to ensure the manuscript fully meets the reviewer's expectations.
>
> ---

---

> ### Author Response · Authors · 2024-11-24
> **Follow-Up on Rebuttal: Request for Feedback**
>
> We sincerely thank the reviewers for their valuable suggestions and thoughtful comments. We would greatly appreciate any additional feedback that could further improve our work and are happy to incorporate further suggestions to ensure its quality. If our responses have satisfactorily resolved your concerns, we kindly request confirmation of whether the provided clarifications and changes address the issues raised and, if applicable, reconsideration of the scores in light of the responses and the updated manuscript.
>
> Thank you again for your time and effort in reviewing our work.

---

> > ### Comment · Reviewer_G1Yq · 2024-11-26
> >
> > I appreciate the authors' efforts during the rebuttal process. However, my main concerns regarding the PAC theorem and the connection between decomposition and time series data remain unresolved. Therefore, I will maintain my original score.

---

> > > ### Author Response · Authors · 2024-11-27
> > > **Connection between decomposition and time series**
> > >
> > > A key insight that guided our approach is the observation of significant parameter redundancy along the channel dimension of source models trained on time-series data. **Figure 2A** demonstrates this redundancy, highlighting the inter-channel dependency inherent in multivariate time-series features as they propagate through deep networks. Prior work [1,2] has shown that modeling dependencies among variables through convolutions along the variable dimension effectively captures these cross-variable relationships. Inspired by this, our method decomposes the source model along the channel dimension, as described in **Equation 3**, enabling us to focus on the most significant channels or variables while maintaining model performance.
> > >
> > > By leveraging the inter-variable dependencies inherent in time-series data, our decomposition approach achieves significant parameter efficiency. Instead of operating on the full channel space, we focus on the principal channels, which correspond to the most critical variables in the parameter space. This focus allows for a compact representation of the model, reducing the number of parameters without sacrificing performance as we observe empirically. The empirical results in **Section 4** highlight the effectiveness of this approach across a variety of settings and datasets, demonstrating its versatility.
> > >
> > > Moreover, to ensure the generalizability of our method, we refrained from introducing specific inductive biases, allowing our approach to be applicable across a wide range of SFDA methods. We validated our method with both prominent generalized SFDA techniques (e.g., SHOT, NRC, AAD) and **state-of-the-art time-series-specific SFDA method (MAPU)** introduced in [3]. Our experiments encompass **all the five datasets in the AdaTime benchmark** [4]—where prior works such as MAPU, typically focus on only three—**along with ten source-target pairs**-where prior works experiment with five source-target pairs—including many-to-one scenarios where a source model trained on multiple domains is adapted to new targets.
> > >
> > > We have also added the following description to our updated manuscript in **Appendix Section A.12 (Our Motivation and Link to Time-Series.)**, highlighted in ${\color{blue} \text{Blue}}$.
> > >
> > > **References**\
> > > [1] Lai, Guokun, et al. "Modeling long-and short-term temporal patterns with deep neural networks." The 41st international ACM SIGIR conference on research & development in information retrieval. 2018.\
> > > [2] Luo, Donghao, and Xue Wang. "Moderntcn: A modern pure convolution structure for general time series analysis." ICLR. 2024 (spotlight).\
> > > [3] Ragab, Mohamed, et al. "Source-free domain adaptation with temporal imputation for time series data." Proceedings of the 29th ACM SIGKDD Conference on Knowledge Discovery and Data Mining. 2023.\
> > > [4] Ragab, Mohamed, et al. "Adatime: A benchmarking suite for domain adaptation on time series data." ACM Transactions on Knowledge Discovery from Data 17.8 (2023): 1-18.

---

> ### Author Response · Authors · 2024-11-27
> **Further clarification on the rationale of  PAC Bayesian analysis**
>
> **Understanding the Original PAC-Bayesian Framework:**
>
> The PAC-Bayesian theory provides bounds on the generalization error—that is, the difference between the empirical loss (on sampled data) and the expected loss (over the entire distribution)—when both are measured with respect to the same data distribution. In this framework:
>
> - **Prior Distribution ($ P $)**: Represents our initial belief about the model parameters before seeing any data.
> - **Posterior Distribution ($ Q $)**: Represents the updated belief after training on the sampled data $ S $.
> - **Sampled Data ($ S $)**: A set of examples drawn from a single data distribution $ D $.
>
> **Our Context with Pre-trained and Fine-tuned Networks:**
>
> In our scenario, we're adapting the PAC-Bayesian framework to the process of fine-tuning a pre-trained network:
>
> 1. **Prior as Pre-trained Weights ($ P = W_{\text{src}} $)**:
>    - We consider the pre-trained network's weights $ W_{\text{src}} $ as our prior distribution.
>    - These weights encapsulate knowledge from previous training on a source dataset but serve as our starting point before fine-tuning.
>
> 2. **Fine-tuning on Target Data ($ S = D_{\text{trg}} $)**:
>    - We fine-tune the model using data $ S $ sampled from the target distribution $ D_{\text{trg}} $.
>    - This process adjusts the model weights to better fit the target data.
>
> 3. **Posterior as Fine-tuned Weights ($ Q = W_{\text{trg}} $)**:
>    - The fine-tuned weights $ W_{\text{trg}} $ represent our posterior distribution.
>    - They reflect our updated belief after seeing the target data.
>
> **Maintaining the Same Distribution:**
>
> A key point is that during fine-tuning, we're exclusively using data from the **target distribution $ D_{\text{trg}} $**. This means:
>
> - Both the empirical loss (computed on $ S = D_{\text{trg}} $) and the expected loss (over $ D_{\text{trg}} $) are with respect to the **same distribution**.
> - The prior and posterior are connected through data sampled from this single distribution, satisfying the PAC-Bayesian assumption of operating within one distribution.
>
> **Replacement Rational:**
>
> - **Alignment with PAC-Bayesian Assumptions**: By treating the pre-trained weights as the prior and the fine-tuned weights as the posterior, we adhere to the framework's requirement of measuring generalization over the same distribution.
> - **Applicability of Generalization Bounds**: Since we're fine-tuning solely on $ D_{\text{trg}} $, we can validly apply PAC-Bayesian generalization bounds to assess how well our fine-tuned model will perform on unseen data from the same distribution.
>
> **In Summary:**
>
> Replacing the prior distribution with the pre-trained network $ W_{\text{src}} $ and the posterior with the fine-tuned network $ W_{\text{trg}} $ is justified because:
>
> - We're operating entirely within the target distribution $ D_{\text{trg}} $.
> - The PAC-Bayesian theory's conditions are met, allowing us to measure the generalization error appropriately.
>
>
> By maintaining the focus on a single distribution throughout the fine-tuning process, we ensure that our application of the PAC-Bayesian theory remains sound.

---

### Official Review · Reviewer_KBWb · 2024-11-03

**Soundness:** 3
**Presentation:** 3
**Contribution:** 3
**Rating:** 6
**Confidence:** 3

**Summary:**

This paper focuses on the topic of Source-Free Time-Series Adaptation. To make the model adaptation more efficient, this paper proposes to decompose the model into core tensor and factor matrices.  When adaptation, only core tensors are selectively fine-tuned. The method can reduce the overall model size and enhance inference efficiency by decomposing and selective fine-tuning.

**Strengths:**

The paper proposes to use Tucker-style factorization for the model decomposition. This decomposition raises a new perspective (at least for me) beyond the conventional feature selection and  BN adaptation methods.

The results are solid. A PAC-Bayes generalization bound is provided for the selective fine-tuning stage. The method is evaluated on the real-world dataset such as AdaTime benchmark.

**Weaknesses:**

I am only curious about some points listed below:

1) Some domain adaptation methods only adapt the BN layers [1],  normalization [2], or only feature modules [3]. They also selectively fine-tune the model, what is the relations between them and the proposed one. Can the proposed method explain or unify these methods in some cases?

2) Can the proposed method be transferred to the general DA task? If not, could you please explain the reasons?

3) It seems that the proposed theorem can be applied to any method with selective fine-tuning. What is the advantage of the proposed decomposition method analyzed from this theorem?

[1] Evaluating prediction-time batch normalization for robustness under covariate shift. arXiv preprint arXiv:2006.10963, 2020.
[2] Tent: Fully Test-time Adaptation by Entropy Minimization. ICLR 2021.
[3] Do we really need to access the source data? source hypothesis transfer for unsupervised domain adaptation. ICML 2020.

**Questions:**

Please refer to the weaknesses part. Overall, I think it is a good paper with new thoughts.

---

> ### Author Response · Authors · 2024-11-21
> **Official Response to Reviewer KBWb**
>
> We sincerely appreciate the reviewer’s recognition of our novel use of Tucker-style factorization for model decomposition, offering a fresh perspective beyond conventional feature selection and BN adaptation methods. We are grateful for the acknowledgment of our solid results, the inclusion of a PAC-Bayes generalization bound for the selective fine-tuning stage, and the comprehensive evaluation of our method on real-world datasets such as the AdaTime benchmark. Below we provide the responses to the concerns raised by the reviewer.
>
> ---
> ### Reviewer Question 1
> > *Some domain adaptation methods only adapt the BN layers [1], normalization [2], or only feature modules [3]. They also selectively fine-tune the model, what is the relations between them and the proposed one. Can the proposed method explain or unify these methods in some cases?\
> >[1] Evaluating prediction-time batch normalization for robustness under covariate shift. arXiv preprint arXiv:2006.10963, 2020.\
> >[2] Tent: Fully Test-time Adaptation by Entropy Minimization. ICLR 2021.\
> >[3] Do we really need to access the source data? source hypothesis transfer for unsupervised domain adaptation. ICML 2020.*
>
> ### Author Response
>
> Thank you for your insightful question regarding the relationship between methods that adapt BN layers, normalization techniques, or feature modules and our proposed approach. We appreciate the opportunity to clarify this.
>
> While domain adaptation methods that focus on adapting batch normalization (BN) layers [1], normalization techniques [2], or selectively fine-tuning feature modules [3] have shown effectiveness, they are orthogonal to our approach, which means our method can be combined with them.
>
> Moreover, BN adaptation and feature normalization methods do not inherently provide interpretability regarding how and what aspects of the model are being adapted. In contrast, our method offers the potential for interpretability. For example, in a toy scenario illustrated in **Figure 4**, we demonstrate that the negated target domain can be adapted by learning an inversion of the core tensor. Although we did not extensively explore interpretability in more complex settings, our approach allows for insights into the source-target transformations that are being learned. Moreover, the decomposition also makes the backbone inference efficient, which is an added benefit.
>
> We also explored the possibility of combining our method with BN adaptation and feature normalization techniques. However, our analysis did not show a significant improvement over using our fine-tuning strategy alone. The standalone fine-tuning of BN layers was tested in our ablation analysis (**Tables 2, 11, 12, and 13, row 3**), which supports this observation. Moreover, the normalization methods also suffer form the same issue. Additionally, our method can also be considered as a seletive finetuning strategy however, we alleviate the dilemma of what module to freeze and what to finetune.
>
> We hope this clarifies how our method relates to and differs from existing approaches. Thank you again for your thoughtful question and for giving us the chance to elaborate on this aspect of our work. Please let us know if you need more insights into these aspects.
>
> ---
> ### Reviewer Weakness 2
> > *Can the proposed method be transferred to the general DA task? If not, could you please explain the reasons?*
>
> ### Author Response
>
> Thank you for your insightful question. Our proposed method is designed with the Source-Free assumption in mind, where we have access to a pre-trained source model but do not have access to the source data during adaptation to the target domain.
> In the conventional Domain Adaptation (DA) framework, it is assumed that both source (labeled) and target (unlabeled) data are available during training. A single network is typically trained jointly using both datasets, and there is no distinction between a source model and a target model in this context. Since there is no separate fine-tuning step involved, there is no clear motivation or reason we see to decompose the backbone network and conduct selective fine-tuning in the typical DA setup.
>
> ---

---

> ### Author Response · Authors · 2024-11-21
> **Official Response to Reviewer KBWb (contd.)**
>
> ### Reviewer Weakness 2
> > It seems that the proposed theorem can be applied to any method with selective fine-tuning. What is the advantage of the proposed decomposition method analyzed from this theorem?
>
> ### Author Response
> Thank you for your insightful observation. You are correct that the presented theorem explaining the PAC-Bayesian bound is general and applicable to any decompose method. Notheless, we emphasize that the bound provided in the manuscript is presented to support our empirical observation and not define a foolproof tight bound. Our contribution lies in demonstrating how the combination of decomposition and selective fine-tuning enhances sample efficiency in the adaptation process. Specifically, our approach effectively constrains the parameters, and we verifiy this for our decomposition and selective finetuning strategy in **Figure 5**. By simplifying the PAC-Bayesian bound, we observe the emergence of a parameter distance factor, which highlights the why such selective finetuning contributes to the sample efficiency of our method.
> While the parameter efficiency might seem apparent due to the low-rank decomposition and selective fine-tuning of only a subset of parameters, the sample efficiency is less obvious. Our work seeks to using  PAC-Bayesian bound as an analytical tool, providing insights into why decomposing and selectively fine-tuning the model not only reduces the number of parameters but also requires fewer samples for effective adaptation.

---

> ### Author Response · Authors · 2024-11-24
> **Follow-Up on Rebuttal: Request for Feedback**
>
> We sincerely thank the reviewers for their valuable suggestions and thoughtful comments. We would greatly appreciate any additional feedback that could further improve our work and are happy to incorporate further suggestions to ensure its quality. If our responses have satisfactorily resolved your concerns, we kindly request confirmation of whether the provided clarifications and changes address the issues raised and, if applicable, reconsideration of the scores in light of the responses and the updated manuscript.
>
> Thank you again for your time and effort in reviewing our work.

---

> > ### Comment · Reviewer_KBWb · 2024-11-25
> > **Thanks for the authors' response.**
> >
> > I appreciate the authors' feedback. The discussions on DA-based methods have effectively addressed my concerns. Although there remains a slight gap between the theorem and the decomposition method, I will maintain my positive score.

---

> > > ### Author Response · Authors · 2024-11-26
> > >
> > > Thank you for your thoughtful and constructive feedback. We are delighted that our discussions on domain adaptation (DA)-based methods have effectively addressed your concerns, and we appreciate your positive assessment of our work.
> > >
> > > We acknowledge the slight gap you have pointed out between the theoretical framework and the decomposition method. As highlighted in our response, our work **primarily emphasizes empirical observations** and leverages PAC-Bayesian tools to provide a supportive theoretical perspective tailored to our context. While these tools reaffirm our empirical findings, we agree that a more concrete theoretical framework specifically tailored to the decomposition method could provide additional insights. This is indeed an exciting avenue for future investigation, one that could deepen the theoretical understanding of such approaches.
> > >
> > > Once again, we sincerely thank you for recognizing the merits of our work. Your insights have been invaluable in shaping our future research directions.

---

### Official Review · Reviewer_vYWA · 2024-11-07

**Soundness:** 3
**Presentation:** 3
**Contribution:** 2
**Rating:** 5
**Confidence:** 4

**Summary:**

The paper introduces a novel domain adaptation approach for time series modeling using tensor decomposition. It decomposes source model weights and only finetunes the tensor core components for target data adaptation. The authors provide PAC-based generalization bounds for the adaptation task and demonstrate computational efficiency through experiments.

**Strengths:**

- New application of tensor decomposition for domain adaptation in time series, building upon concepts from Monarch architecture and block tensor train methods used in NLP transformer models
- Comprehensive experimental results across multiple datasets clearly demonstrate computational benefits

**Weaknesses:**

1. While the theoretical analysis provides valuable insights into the trade-off between target data fit and source-target distribution divergence, the assumptions have limitations:

- The posterior $Q(S)$ is assumed to be an isotropic Gaussian with variance matching the prior $P$. This assumption doesn't reflect real-world scenarios and is more like a typical variational inference solutions. A more realistic upper bound should incorporate: (1) Weight differences; (2) Distance between isotropic Gaussian posterior and true posterior. While such an extension appears feasible, the distance metric between isotropic Gaussian and true posterior remains undefined

-  The performance of SFDA compared to LoRA aligns with expected outcomes based on existing work in Monarch and Tensor Train architectures, making it somewhat incremental

**Questions:**

Related Work:
The paper would benefit from discussing connections to recent NLP transformer architectures, particularly:

- Dao et al. (2022), "Monarch: Expressive Structured Matrices for Efficient and Accurate Training"
- Qiu et al. (2024), "Compute Better Spent: Replacing Dense Layers with Structured Matrices"

---

> ### Author Response · Authors · 2024-11-21
> **Official Response to Reviewer vYWA**
>
> We sincerely appreciate the reviewer’s recognition of our novel application of tensor decomposition for domain adaptation in time-series data. We are also grateful for the acknowledgment of our comprehensive experimental results across multiple datasets, which clearly highlight the computational benefits of our approach. Below we provide the responses to the concerns raised by the reviewer. All the changes in the revised version of the manuscript are highlighted in ${\color{blue}\text{Blue}}$.
>
> ---
> ### Reviewer Weakness 1
> >*While the theoretical analysis provides valuable insights into the trade-off between target data fit and source-target distribution divergence, the assumptions have limitations:\
> The posterior $Q(S)$ is assumed to be an isotropic Gaussian with variance matching the prior $P$. This assumption doesn't reflect real-world scenarios and is more like a typical variational inference solutions. A more realistic upper bound should incorporate: (1) Weight differences; (2) Distance between isotropic Gaussian posterior and true posterior. While such an extension appears feasible, the distance metric between isotropic Gaussian and true posterior remains undefined*
>
> ### Author Response
>
> We sincerely thank you for your thoughtful feedback on our theoretical analysis. Your comments raise important points about the assumptions in our PAC-Bayesian generalization bound, and we appreciate the opportunity to clarify and expand upon these aspects.
>
> We appreciate your concern regarding the assumption of an isotropic Gaussian posterior with variance matching the prior. We acknowledge that this simplification may not fully capture the complexities of real-world posterior distributions in deep neural networks. However, we believe that our approach provides a meaningful and tractable way to understand the generalization behavior in the context of fine-tuning.
>
> **1. Incorporating Weight Differences:**
> Our theoretical framework does incorporate weight differences between the source (pre-trained) and target (fine-tuned) models. Specifically, in Theorem 1 of our paper, we derive a PAC-Bayesian generalization bound that includes the distance term. This term explicitly quantifies the sum of squared Frobenius norms of the differences between the target and source weights across all layers. By including this term in the bound, we directly account for the weight differences, addressing your first point about incorporating weight differences into a more realistic upper bound.
>
> **2. Distance Between Isotropic Gaussian Posterior and True Posterior:**
> We agree that the true posterior in deep neural networks is likely complex and not accurately represented by an isotropic Gaussian. Hence, we are not able explicitly define a distance metric between the isotropic Gaussian posterior and the true posterior (due to the intractability of the true posterior).
>
> **3. Justification for the Isotropic Gaussian Assumption:**
> The assumption of an isotropic Gaussian posterior is indeed a simplification, and it is a common practice in PAC-Bayesian analyses to enable analytical tractability. This choice allows us to derive explicit bounds and gain insights into the factors affecting generalization. Moreover, recent works  [1,2,3] have successfully employed similar assumptions to obtain generalization bounds and practical insights into neural network training.
>
> **4. Support for Empirical Observations:**
> The bound derived is not a strict one and is intended more as an analytical tool to gain insights into and support the underlying  fine-tuning phase, rather than being a primary contribution of our work. The included theoretical analysis aims to complement our primary empirical findings. Therefore, we also empirically visualize the source-target weight parameter distance in **Figure 5**.
>
>
> Nonetheless, incorporating the error term between the isotropic posterior and the true posterior, is an excellent suggestion. However, defining these metric/error rigorously for decomposed tensors under the Tucker framework remains a non-trivial extension, which we leave as future work.
>
> **References:**
>
> [1] Dziugaite, Gintare Karolina, and Daniel M. Roy. "Computing Nonvacuous Generalization Bounds for Deep (Stochastic) Neural Networks with Many More Parameters than Training Data." Conference on Uncertainty in Artificial Intelligence (UAI), 2017.\
> [2] Li, Dongyue, and Hongyang R. Zhang. "Improved Regularization and Robustness for Fine-Tuning in Neural Networks."NeurIPS, 2021.\
> [3] Wang, Zifan, et al. "Improving Robust Generalization by Direct PAC-Bayesian Bound Minimization." CVPR, 2023.
>
> ---

---

> ### Author Response · Authors · 2024-11-21
> **Official Response to Reviewer vYWA (contd.)**
>
> ### Reviewer Question 1
> >*Related Work: The paper would benefit from discussing connections to recent NLP transformer architectures, particularly:*
> >- *Dao et al. (2022), "Monarch: Expressive Structured Matrices for Efficient and Accurate Training"*
> >- *Qiu et al. (2024), "Compute Better Spent: Replacing Dense Layers with Structured Matrices"*
>
> ### Author Response
> We appreciate the reviewer’s suggestion to discuss connections to recent NLP transformer architectures, particularly the works by Dao et al. (2022) and Qiu et al. (2024). In response, we have added a detailed discussion in the extended related work section of the revised manuscript, specifically in **Section A.12 (Parameter Redundancy and Low-Rank Subspaces)**.

---

> ### Author Response · Authors · 2024-11-24
> **Follow-Up on Rebuttal: Request for Feedback**
>
> We sincerely thank the reviewers for their valuable suggestions and thoughtful comments. We would greatly appreciate any additional feedback that could further improve our work and are happy to incorporate further suggestions to ensure its quality. If our responses have satisfactorily resolved your concerns, we kindly request confirmation of whether the provided clarifications and changes address the issues raised and, if applicable, reconsideration of the scores in light of the responses and the updated manuscript.
>
> Thank you again for your time and effort in reviewing our work.

---

### Official Review · Reviewer_gGV1 · 2024-11-07

**Soundness:** 2
**Presentation:** 2
**Contribution:** 2
**Rating:** 5
**Confidence:** 4

**Summary:**

This paper considers SFDA using Tucker style decomposed weights and selective fine tuning on target data. Some bounds are presented for this strategy and experimental results are shown.

**Strengths:**

The proposed method can be integrated with several baselines SFDA methods to improve adaptation efficiency.

**Weaknesses:**

1. Cite and compare with:
- Tang et al, Source-Free Domain Adaptation with Frozen Multimodal Foundation Model, CVPR 2024
- Litrico et al, Guiding pseudo-labels with uncertainty estimation for source-free unsupervised domain adaptation, CVPR 2023
- Tang et al, Source-free domain adaptation via target prediction distribution searching, ICLR 2023
- Gao et al, UNITS: A Unified Multi-Task Time Series Model, NeurIPS 2024
- Ding et al, Source-Free Domain Adaptation via Distribution Estimation, CVPR 2022
It's important to compare against foundation model style approaches since they offer SoTA adaptation efficiency.

2. Writing needs to be improved. The tables with results and plots are very very tiny, and zoom 200% is needed to read them and evaluate. Please choose the most important results to present.

3. The loss functions for target adaptation are not presented, please be specific and self-contained.

4. The method is not time series specific. Why not compare with vision models and datasets to show universality of the method?

5. The authors do not compare against Lokra, Lora with the same style of parameter block freezing and tuning. This would improve fairness of comparisons.

**Questions:**

See weaknesses

---

> ### Author Response · Authors · 2024-11-21
> **Official Response to Reviewer gGV1**
>
> We sincerely appreciate the reviewer’s recognition of the adaptability and versatility of our proposed method, particularly its ability to integrate seamlessly with several baseline SFDA methods to enhance adaptation efficiency. Below we provide the responses to the concerns raised by the reviewer. All the changes in the revised version of the manuscript are highlighted in ${\color{blue}\text{Blue}}$.
>
> ---
>
> ### Reviewer Weakness 1
> >*Writing needs to be improved. The tables with results and plots are very very tiny, and zoom 200% is needed to read them and evaluate. Please choose the most important results to present.*
>
>
> ### Author Response
> Thank you for your valuable feedback. We have made revisions to prune the tables to improve its readability and visibility. We have carefully selected the most important results to present, streamlining the tables for better focus and clarity.
>
> ---
>
> ### Reviewer Weakness 2
> >*The loss functions for target adaptation are not presented, please be specific and self-contained.*
>
> ### Author Response
> Thank you for your thoughtful feedback and for highlighting the need for more specificity regarding the target adaptation loss functions. We truly appreciate your attention to detail and your suggestion to make the manuscript more self-contained.
>
> In response, we have added a new **Section A.8** in the appendix, where we provide a comprehensive description of all the adaptation objectives used in our work. We believe this addition enhances the clarity and self-contained nature of the manuscript.
>
> ---
> ### Reviewer Weakness 3
> >*The authors do not compare against Lokra, Lora with the same style of parameter block freezing and tuning. This would improve fairness of comparisons.*
>
> ### Author Response
> Thank you for your insightful suggestion to compare against LoRA and LoKrA with the same style of parameter block freezing and tuning. We greatly appreciate your recommendation to enhance the fairness of our comparisons.
>
> In response to your feedback, we have incorporated this analysis and presented the results in the newly added **Section A.10** in the appendix of the updated manuscript. We believe this addition provides a more comprehensive and balanced evaluation of our method.
>
> ---

---

> ### Author Response · Authors · 2024-11-21
> **Official Response to Reviewer gGV1 (contd.)**
>
> ### Reviewer Weakness 4
>
> >*Cite and compare with:
> •	Tang et al, Source-Free Domain Adaptation with Frozen Multimodal Foundation Model, CVPR 2024\
> •	Litrico et al, Guiding pseudo-labels with uncertainty estimation for source-free unsupervised domain adaptation, CVPR 2023\
> •	Tang et al, Source-free domain adaptation via target prediction distribution searching, ICLR 2023\
> •	Gao et al, UNITS: A Unified Multi-Task Time Series Model, NeurIPS 2024\
> •	Ding et al, Source-Free Domain Adaptation via Distribution Estimation, CVPR 2022 It's important to compare against foundation model style approaches since they offer SoTA adaptation efficiency.*
>
> ### Author Response
>
> We appreciate the reviewer’s observation on the importance of comparing against foundation model-style approaches for achieving state-of-the-art (SoTA) adaptation efficiency. Our work, however, intentionally addresses a different but complementary challenge in the domain of Source-Free Domain Adaptation (SFDA). As emphasized in **Line 079-083** of the manuscript:
>
> > “It is important to emphasize that our contribution does not introduce a novel SFDA method for time-series data. Instead, we focus on making the target adaptation process more parameter- and sample-efficient and demonstrating that our framework can be seamlessly integrated with existing, and potentially future, SFDA techniques.”
>
> Our work specifically targets **parameter efficiency** and **sample efficiency** for time-series data in the context of SFDA, a challenge that remains unexplored. This focus is critical for real-world applications in computationally constrained or low-data environments, where deploying large foundation models may not be practical due to their computational overhead or lack of niche domain-specific representation.
>
> ### Citation and Comparison
> To address the reviewer’s suggestion, we have incorporated citations for the following works in the added extended related works section (see Appendix Section A.12) and have discussed their relevance in the context of our research:
> - Tang et al., "Source-Free Domain Adaptation with Frozen Multimodal Foundation Model," CVPR 2024.
> - Litrico et al., "Guiding pseudo-labels with uncertainty estimation for source-free unsupervised domain adaptation," CVPR 2023.
> - Tang et al., "Source-free domain adaptation via target prediction distribution searching," ICLR 2023.
> - Gao et al., "UNITS: A Unified Multi-Task Time Series Model," NeurIPS 2024.
> - Ding et al., "Source-Free Domain Adaptation via Distribution Estimation," CVPR 2022.
>
> In terms of experimental comparisons, we adopted the setup and benchmarks established by **Ragab et al.** for a fair evaluation of our framework. We have clarified in the main text and **Appendix Section A.9, Lines 1440-1442** that **Tables 5–9** in the appendix present predictive performance comparisons with prior methods and SoTA results from Ragab et al. (Tables 2, 3, and 4 in [1] ). As these comparisons demonstrate, our framework achieves SoTA predictive performance while being significantly more parameter- and sample-efficient than existing methods.
>
> ### Relevance to Foundation Model-Based Approaches
> We acknowledge that foundation models, such as the ones referenced in Tang et al. and Gao et al., offer substantial benefits in learning generalized features across diverse domains and facilitating adaptation with limited target samples. However, our work provides a complementary perspective by focusing on **fine-tuning efficiency** , ensuring minimal computational overhead and adaptability to specialized domains. These aspects are particularly critical when foundation models are either unavailable or inefficient due to domain-specific mismatches, as noted in our extended related works section (**Appendix Section A.12**). Additonal the combination of utilizing foundation models along with our method could be an interesting direction to investigate and we will definitely be exploring it our future works (**Appendix Section A.14**).
>
> We hope this response clarifies our position and demonstrates how our work complements existing foundation model-based approaches while addressing a distinct challenge adaptation efficiency.
>
> **References**
>
> [1] Ragab, Mohamed, et al. "Source-free domain adaptation with temporal imputation for time series data." Proceedings of the 29th ACM SIGKDD Conference on Knowledge Discovery and Data Mining. 2023.

---

> ### Author Response · Authors · 2024-11-21
> **Official Response to Reviewer gGV1 (contd.)**
>
> ### Reviewer Weakness 5
> > *The method is not time series specific. Why not compare with vision models and datasets to show universality of the method?*
>
> ### Author Response
> Thank you for your insightful comment highlighting the potential universality of our method. You are correct that the core principles of our approach are not inherently specific to time-series data. However, our primary motivation was to address specific challenges within the domain of Source-Free Domain Adaptation (SFDA) for time-series data—that has a research gap in terms of parameter and sample efficiency of the adaptation process.
>
> In our research, we observed significant parameter redundancy along the channel (or variable) dimension in models trained on time-series data (see **Figure 2A** in the paper). This redundancy suggests substantial inter-channel dependencies that time-series data demonstrate as they propagate through deep networks. Previous studies [1,2] have explored leveraging these dependencies by applying convolutions along the variable dimension, reinforcing the notion that such inter-variable relationships are inherent and particularly pronounced in time-series data.
>
> Building on these insights, we proposed decomposing the model along the channel dimension (as detailed in Equation 3 of our paper), and focusing on the principal channels or variables. This decomposition leverages the inherent inter-variable dependencies specific to time-series data, allowing us to achieve parameter efficiency without sacrificing performance.
>
> While it is true that the mathematical underpinnings of our method could be applied to other domains like computer vision, the characteristics and dependencies in those domains may differ. For instance, vision models often deal with spatial and hierarchical features, and the inter-channel redundancies may manifest differently compared to time-series data. Therefore applying our method to vision datasets may or may not require addressing these different types of dependencies and might necessitate additional modifications beyond the scope of our current work.
>
> Our goal was to keep the method general within the context of time-series SFDA to ensure broad applicability across various time-series datasets and adaptation scenarios. To this end, we avoid introducing specific inductive biases tied to datasets or the adaptation method. We validated our approach extensively on all five datasets from the AdaTime benchmark—conducting experiments on ten source-target pairs, including a many-to-one setting—providing a comprehensive evaluation within the time-series domain.
>
> We appreciate your suggestion to explore the universality of our method by applying it to vision models and datasets. This is indeed an interesting direction for future research and could help in understanding the broader applicability of our approach across different domains. To this end we have added a **Future Work section (Section A.14)** to discuss explore the unversality of our method.
>
> **Reference**
>
> [1] Lai, Guokun, et al. "Modeling long-and short-term temporal patterns with deep neural networks." The 41st international ACM SIGIR conference on research & development in information retrieval. 2018.\
> [2] Luo, Donghao, and Xue Wang. "Moderntcn: A modern pure convolution structure for general time series analysis." ICLR. 2024 (spotlight).

---

> ### Author Response · Authors · 2024-11-24
> **Follow-Up on Rebuttal: Request for Feedback**
>
> We sincerely thank the reviewers for their valuable suggestions and thoughtful comments. We would greatly appreciate any additional feedback that could further improve our work and are happy to incorporate further suggestions to ensure its quality. If our responses have satisfactorily resolved your concerns, we kindly request confirmation of whether the provided clarifications and changes address the issues raised and, if applicable, reconsideration of the scores in light of the responses and the updated manuscript.
>
> Thank you again for your time and effort in reviewing our work.

---

> > ### Comment · Reviewer_gGV1 · 2024-11-25
> > **Response to Author Rebuttal**
> >
> > I thank the authors for their rebuttals. The Theorem presented is not particularly novel or closely tied to the paper results, and the general idea of structured decomposed weights (e.g. tensor trains, Kronecker, low rank, butterfly, Tucker) has been explored in several prior works as authors noted as well, the SFDA setup also seems not truly the only way to make these weight decompositions approaches beneficial in target adaptation (for example, if labels were available in fine tuning). In light of these comments, the updated manuscript, and other reviews, I choose to retain my score.

---

> > > ### Author Response · Authors · 2024-11-26
> > >
> > > We appreciate your insights and would like to address the concerns you've raised.
> > > ### On Relevance of the Theorem:
> > > **Our primary goal is not to introduce a novel theorem** but to utilize established PAC-Bayesian bounds as a theoretical framework to explain our empirical observations of implicit regularization and sample efficiency (see **Figure 2B, Figure 6, and Table 1**). We also clearly mention this in the paper Abstract (**lines 018-020**). The PAC-Bayesian analysis serves as a tool to provide theoretical insights into why our strategy exhibits improved performance in the sample-scarce scenario. By grounding our findings in existing theoretical work, we aim to **bridge the gap between empirical results and theoretical understanding** in this specific context.
> > >
> > > ### On the Use of Structured Decomposed Weights:
> > > While it is true that structured decomposition methods such as tensor trains, Kronecker products, low-rank approximations, butterfly transformations, and Tucker decompositions are existing methods, their application or use case within the SFDA framework is not straightforward or previously established. Our key thought behind conducting parameter decomposition is motivated by the parameter redundancies we observe in source pre-trained models (see **Figure 2A** and further discussed in **Section A.12 ("Our Motivation and Link to Time-Series.")** These redundancies present an opportunity to enhance parameter efficiency. Moreover, in our comment above (https://openreview.net/forum?id=Q5Sawm0nqo&noteId=mU1HI7vHAh) we discuss our motivation behind using Tucker decomposition that we extend upon in **Appendix Section A.12 ("Parameter Redundancy and Low-Rank Subspaces.")**
> > >
> > > Nonetheless, in addition to the obvious parameter efficiency due to decomposition, we also make the following contributions demonstrating:
> > > - **Implicit Regularization:** Decomposition and selective fine-tuning inherently impose a form of regularization that mitigates overfitting during unsupervised adaptation. This implicit regularization is particularly beneficial in SFDA, where the risk of fitting to spurious correlations is higher due to the lack of labeled target data.
> > > - **Sample Efficiency:** Due to the implicit regularization introduced, we achieve better performance with fewer samples (as demonstrated in **Figure 6** and **Table 1**), which is crucial in scenarios with significant resource disparities and sample scarcity.
> > >
> > > We demonstrate these two additional contributions of our work empirically and seek to explain these observations using the PAC-Bayesian analysis. **Note that we do not propose a new decomposition strategy, it is out of scope of this work, nor do we claim to do so**.
> > >
> > >
> > > ### On the SFDA Setup and its Distinction from Supervised Fine-Tuning:
> > > The SFDA scenario fundamentally differs from traditional supervised fine-tuning due to the absence of labeled target data. SFDA relies on unsupervised objectives (as detailed in **Appendix Section A.8**), these objectives entail risks for model adapting incorrectly to the target domain due to the lack of ground-truth labels. In this context, decomposition plays a crucial role by:
> > > - **Restricting Parameter Exploration:** Predefined subspaces limit the model's capacity to overfit to noisy unsupervised samples in the target data, guiding the adaptation process more effectively than methods that utilize randomly initialized subspaces.
> > > - **Enhancing Stability:** The structured approach provides a more stable adaptation trajectory, which is essential when labels are not available to correct the model's course.
> > > Our empirical results indicate that methods like LoRA and LoKrA, which do not employ predefined subspaces and rely on randomly initialized ones, underperform in the SFDA setting, potentially due to the unsupervised fine-tuning strategy (see **Sections 4 and Figure 7, 12 ,13, 14, 15**). This underperformance underscores the importance of our proposed approach in scenarios where labels are unavailable during adaptation.
> > >
> > > We believe that our work offers valuable insights into the application of structured decomposition within SFDA, highlighting its benefits in terms of implicit regularization and sample efficiency, in addition to the obvious parameter efficiency. We provide a novel perspective and practical solutions to the challenges inherent in our problem setup.

---

### Official Review · Reviewer_bZkd · 2024-11-08

**Soundness:** 4
**Presentation:** 4
**Contribution:** 4
**Rating:** 10
**Confidence:** 4

**Summary:**

This work introduces a method for source-free domain adaptation (SFDA) in the domain of time-series data. The task poses a challenge due to two main reasons: 1) the model has access only to unlabeled data and the source-pretrained model but not the source data, and 2) finetuning with small amounts of unlabeled data in this domain often results is overfitting. This papers explore the method of applying Tucker-style factorization, by breaking the parameters into two parts: core-tensors and factor matrices. By doing so and selectively finetune only on the core-tensor matrix, the model shows little signs of overfitting and improved generalization. Empirically, experiments from changing the rank fo the decomposed tensor to comparisons with current popular adaptor methods show the proposed method is effective at reducing overfitting and improving generalization with fewer parameters, achieving the best of both worlds. Finally, the authors provided a PAC-Bayes generalization bound, showing that the test loss is upper bounded by the empirical loss plus the regularization effect from the matrices learned.

**Strengths:**

The problem presented at this paper is of great significance, with considerations about the challenges of finetuning the full model and application of on-device settings. The authors have done a great job in both the introduction and method motivating the problem and stating the importance and relevance of this research problem. The method is clear, with definitions and equations clearly defined and laid-out. The method proposed itself is sound also. The authors also try to draw connections with related works to Lottery Ticket Hypothesis, as an attempt to justify why their method works well and is able to achieve the best of both worlds; This connection is a nice touch and welcomed. Experiments-wise, the paper has done a thorough job in demonstrating the effectiveness of their method, with sufficient number of datasets and models, and comparisons with state-of-the-art methods that show clear improvements.

**Weaknesses:**

A weakness of this work can be characterized as a missing link between the entire method and the time-series aspect of the problem. Specifically, it’s not entire clear why the method here is specifically applied to time-series, as the authors have setup here. Perhaps the authors can clarify how their method has leverage properties in time-series data to their advantage. If this connection is indeed not mentioned, then it seems reasonable to add a sentence or a section in the Appendix justifying this.

**Questions:**

1. Can the authors comment on the time for inference before and after factorization? Does the smaller number of parameters reduce the time required for inference? Or does the increased number of matrix factorization increase the time required?

---

> ### Author Response · Authors · 2024-11-21
> **Official Response to Reviewer bZkd**
>
> We sincerely appreciate the reviewer’s acknowledgment of the significance of the research problem, particularly its relevance to the challenges of fine-tuning full models in on-device settings. We are grateful for the recognition of our efforts in motivating the problem and articulating its importance in the introduction and methodology sections. The reviewer’s appreciation of the clarity in our definitions, equations, and the soundness of the proposed method is highly valued. We also thank the reviewer for highlighting our effort to connect the proposed approach to the Lottery Ticket Hypothesis, which provides an intuitive justification for its effectiveness. Lastly, we are pleased that the thoroughness of our experiments, including the diverse datasets, models, and state-of-the-art comparisons showing clear improvements, has been noted. Below we provide the responses to the concerns raised by the reviewer. All the changes in the revised version of the manuscript are highlighed in ${\color{blue}\text{Blue}}$.
>
> ---
>
> ### Reviewer Weakness 1
> >*A weakness of this work can be characterized as a missing link between the entire method and the time-series aspect of the problem. Specifically, it’s not entire clear why the method here is specifically applied to time-series, as the authors have setup here. Perhaps the authors can clarify how their method has leverage properties in time-series data to their advantage. If this connection is indeed not mentioned, then it seems reasonable to add a sentence or a section in the Appendix justifying this.*
> ### Author Response
> Thank you for your thoughtful feedback and for highlighting an important aspect of our work. We genuinely appreciate your attention to detail and your efforts to help improve the interpretability of our method within the context of time-series data.
>
> You are right in pointing out the missing link in our work to time series. Our primary motivation is to address the simultaneous parameter and sample efficiency in Source-Free Domain Adaptation (SFDA) for time-series, a pressing research gap we identified in the community. In our study, we observed significant parameter redundancy along the channel dimension of source models (see Figure 2A in the paper). This redundancy suggests substantial inter-channel dependency among variables in multivariate time-series features as they propagate through a deep network. Works by [1,2] has explored using convolution along the variable dimension to capture such cross-variable dependencies, reinforcing the notion that these dependencies are inherent in time-series data.
>
> Building on this insight, we proposed decomposing the model along the channel dimension (as demonstrated in Equation 3 of our paper) and operating on the decomposed factors corresponding to the principal channels or variables. This approach leverages the inherent inter-variable dependencies in time-series data, allowing us to achieve parameter efficiency by focusing on the most significant channels without sacrificing performance.
>
> Our aim was to keep the method as general as possible to ensure its applicability across different SFDA settings and diverse time-series datasets. To this end, we avoided introducing specific inductive biases and experimented with both generalized SFDA methods (such as SHOT, NRC, AAD) and a time-series-specific SFDA method (MAPU), following the initial work of Ragab et al [3]. Hence, we are empirically thorough, experimenting on all five datasets from the AdaTime benchmark [4]—where previous works typically focused on only three—and conducted experiments on ten source-target pairs, including a many-to-one setting where the source model is trained on multiple sources and then adapted to other targets.
>
> To that end, we have added a **Future Work section (Section A.14)** in the revised version addressing the above discussion.
>
> **References**\
> [1] Lai, Guokun, et al. "Modeling long-and short-term temporal patterns with deep neural networks." The 41st international ACM SIGIR conference on research & development in information retrieval. 2018.\
> [2] Luo, Donghao, and Xue Wang. "Moderntcn: A modern pure convolution structure for general time series analysis." ICLR. 2024 (spotlight).\
> [3] Ragab, Mohamed, et al. "Source-free domain adaptation with temporal imputation for time series data." Proceedings of the 29th ACM SIGKDD Conference on Knowledge Discovery and Data Mining. 2023.\
> [4] Ragab, Mohamed, et al. "Adatime: A benchmarking suite for domain adaptation on time series data." ACM Transactions on Knowledge Discovery from Data 17.8 (2023): 1-18.
>
> ---

---

> ### Author Response · Authors · 2024-11-21
> **Official Response to Reviewer bZkd (contd)**
>
> ### Reviewer Question 1
> > *Can the authors comment on the time for inference before and after factorization? Does the smaller number of parameters reduce the time required for inference? Or does the increased number of matrix factorization increase the time required?*
>
> ### Author Response
> The factorization also helps improve the inference (reduced inference time) by reducing the total number of computations involved. We appreciate the reviewer's insightful question and are happy to provide a detailed analysis of the inference analysis before and after applying Tucker decomposition to a convolutional neural network (CNN) layer. We also quantify the inference computation using multiply-accumulate operation counts (MACs), as mentioned in the manuscript.
> ### *Original Convolutional Layer:*
> Consider a convolutional layer with the following dimensions:
> - **Input Channels ($C_{\text{in}}$)**: 128
> - **Output Channels ($C_{\text{out}}$)**: 256
> - **Kernel Size ($K$)**: 8
> - **Input Length ($L$)**: Length of the input sequence
> - **Output Length ($L'$)**: Length of the output feature map
> The computational complexity of the original convolution operation is:
> $$
> Computations_{\text{original}} = \mathcal{O}(C_{\text{out}} \times C_{\text{in}} \times K \times L') = 256 \times 128 \times 8 \times L' = 262{,}144 \times L'
>  $$
> ### *After Tucker Decomposition:*
> We apply Tucker decomposition along the channel dimensions (input and output channels) of the weight tensor. The decomposition factorizes the original weight tensor into smaller tensors, introducing two rank parameters:
> - **Input Rank ($R_{\text{in}}$)**: Reduced dimension for input channels
> - **Output Rank ($R_{\text{out}}$)**: Reduced dimension for output channels
> The computational complexity after Tucker decomposition becomes:
> $$
> \begin{align*}
> Computations_{\text{decomposed}} &= \mathcal{O}(C_{\text{in}} \times R_{\text{in}} \times L) + \mathcal{O}(R_{\text{out}} \times R_{\text{in}} \times K \times L') + \mathcal{O}(C_{\text{out}} \times R_{\text{out}} \times L') \\
> &= \text{Input Projection} + \text{Core Convolution} + \text{Output Projection}
> \end{align*}
> $$
>
> ### *Components Computation Explanation:*
> 1. **Input Projection:**
> Project the input feature maps from $C_{\text{in}}$ channels to $R_{\text{in}}$ channels:
> $$
> \text{Input Projection} = C_{\text{in}} \times R_{\text{in}} \times L = 128 \times R_{\text{in}} \times L
> $$
> 2. **Core Convolution:**
> Perform convolution using the core tensor of size $R_{\text{out}} \times R_{\text{in}} \times K$:
> $$
> \text{Core Convolution} = R_{\text{out}} \times R_{\text{in}} \times K \times L' = R_{\text{out}} \times R_{\text{in}} \times 8 \times L'
> $$
> 3. **Output Projection:**
> Project the result from $R_{\text{out}}$ channels to $C_{\text{out}}$ output channels:
> $$
> \text{Output Projection} = C_{\text{out}} \times R_{\text{out}} \times L' = 256 \times R_{\text{out}} \times L'
> $$
>
> ### *Computational Efficiency Demonstration:*
> For demonstration purposes, let's assume equal reduced ranks for input and output channels:
> - $R_{\text{in}} = R_{\text{out}} = R$, where $R = 32$
> ### *Computations After Decomposition:*
> 1. **Input Projection:**
> $$
> \text{Input Projection} = 128 \times 32 \times L = 4{,}096 \times L
> $$
> 2. **Core Convolution:**
> $$
> \text{Core Convolution} = 32 \times 32 \times 8 \times L' = 8{,}192 \times L'
> $$
> 3. **Output Projection:**
> $$
> \text{Output Projection} = 256 \times 32 \times L' = 8{,}192 \times L'
> $$
> ### *Total Computations After Decomposition:*
> $$
> \text{Computations}_{\text{decomposed}} = 4{,}096 \times L + 8{,}192 \times L' + 8{,}192 \times L' = 4{,}096 \times L + 16{,}384 \times L'
> $$
>
> ### *Comparing Computational Costs:*
> Assuming the input and output lengths are approximately equal ($L \approx L'$), we can directly compare the total computations.
> 1. **Total Computations Before Decomposition:**
> $$
> \text{Computations}_{\text{original}} = 262{,}144 \times L'
> $$
> 2. **Total Computations After Decomposition:**
> $$
> \text{Computations}_{\text{decomposed}} = 4{,}096 \times L' + 16{,}384 \times L' = 20{,}480 \times L'
> $$
> 3. **Computational Reduction:**
> $$
> \text{Reduction Ratio} = \frac{Computations_{\text{decomposed}}}{Computations_{\text{original}}} = \frac{20{,}480 \times L'}{262{,}144 \times L'} \approx 0.078
> $$
> This indicates a reduction of approximately **92%** in computational cost after Tucker decomposition.
>
> ### *Impact on Inference Time:*
> - **Reduced Computation:** The substantial decrease in computations directly translates to faster inference times, as the model performs significantly fewer operations.
> - **Efficient Projections:** The additional projection operations involve smaller matrices due to the reduced rank $R$, making them computationally efficient.
>
> The reduced ranks $R_{\text{in}}$ and $R_{\text{out}}$ lead to a lower computational complexity compared to the original convolutional layer.

---

> ### Author Response · Authors · 2024-11-24
> **Follow-Up on Rebuttal: Request for Feedback**
>
> We sincerely thank the reviewers for their valuable suggestions and thoughtful comments. We would greatly appreciate any additional feedback that could further improve our work and are happy to incorporate further suggestions to ensure its quality. If our responses have satisfactorily resolved your concerns, we kindly request confirmation of whether the provided clarifications and changes address the issues raised.
>
> Thank you again for your time and effort in reviewing our work.

---

> > ### Comment · Reviewer_bZkd · 2024-11-25
> >
> > Thank you for the response. I believe the results in this paper does a quite thorough justification for why the method works. I hope the authors can incorporate the comments above into the paper, either in the main manuscript or in the appendix.

---

> > > ### Author Response · Authors · 2024-11-26
> > >
> > > Thank you for your thoughtful feedback and for acknowledging the thorough justification provided in our paper. We are pleased to hear that our revisions have addressed your concerns. Based on your valuable suggestions, we have incorporated the discussed points in Appendix Section A.2.3 and Section A.12 (Our Motivation and Link to Time-Series) of the revised manuscript. All changes have been highlighted in ${\color{blue} \text{Blue}}$ for your convenience. We appreciate your constructive input, which has helped us improve the clarity and depth of our work.

---

### Official Review · Reviewer_56c7 · 2024-11-08

**Soundness:** 3
**Presentation:** 3
**Contribution:** 3
**Rating:** 6
**Confidence:** 4

**Summary:**

This work presented a Tucker-style tensor factorization as a low-rank decomposition of source-pretrained models for time-series data. This method enables a selective fine-tuning of the core tensor on the target side, effectively improving the efficiency.

**Strengths:**

I think the topic of the article is cutting-edge. The article as a whole uses rich methods such as PAC-Bayes to expound the effectiveness of the proposed method. The supplementary materials are also very detailed in explaining the article, such as the derivation of the formal bound parameter distance. The experiments are sufficient, and the effects shown in the experiments are good. The elaboration and analysis of limitations are also very sufficient.

**Weaknesses:**

1. In my opinion, in the Related Work section, what is shown is more accurately ablations rather than the so-called observations in the title. Compared to this experimental result, it is more necessary to simply explain why Tucker-style decomposition is used instead of other decomposition methods you have exemplified.
2. I think the insights in lines 201-204 need a more complete explanation. From Equation 1, I can only understand that the linear operation in deep learning is the so-called sequence of linear suboperations in the forward process. However, I cannot directly confirm whether such a description is also accurate for the backward process. If there are relevant formula derivations or proofs, references can be provided. If not, considering the space limitation, it is best to prove it in the Appendix. Because this claim is the premise of Equations 4, 5, and 6 later, which is necessary and important for understanding this method.
3. A small mistake. In the sentence after Eq3, on line 221, after “where”, it should be an explanation of the symbol V in Equation 3 instead of U.
4. On line 238, “Building on these insights, we decompose the pre-trained source model weights using Tucker decomposition”. Although the expression here is rigorous. Logically, the former is a necessary condition for the latter. From the beginning of the article until now, there is no sufficient reason given for why Tucker decomposition is used instead of other decomposition methods. Even if it is mentioned earlier, it is better to emphasize it again here.

**Questions:**

Lines 248-252, why is fine-tuning for 2-3 epochs? Is this number closely related to the dataset and model scale? I am not very familiar with this field and hope that relatively rigorous evidence can be given here in the text.

---

> ### Author Response · Authors · 2024-11-21
> **Official Response to Reviewer 56c7**
>
> We sincerely appreciate the reviewer’s recognition of the cutting-edge nature of our work and the rich methodological framework, particularly our use of PAC-Bayes to demonstrate the effectiveness of the proposed approach. We are grateful for the acknowledgment of the detailed supplementary materials, including the derivation of the formal bound for parameter distance, as well as the sufficiency and quality of our experiments. Furthermore, we value the reviewer’s appreciation of our thorough elaboration and analysis of the limitations of our approach. Below we provide the responses to the concerns raised by the reviewer. All the changes in the revised version of the manuscript are highlighed in ${\color{blue}\text{Blue}}$.
>
> ---
> ### Reviewer Weakness 1 and 4
> >*In my opinion, in the Related Work section, what is shown is more accurately ablations rather than the so-called observations in the title. Compared to this experimental result, it is more necessary to simply explain why Tucker-style decomposition is used instead of other decomposition methods you have exemplified.*
>
> > *On line 238, “Building on these insights, we decompose the pre-trained source model weights using Tucker decomposition”. Although the expression here is rigorous. Logically, the former is a necessary condition for the latter. From the beginning of the article until now, there is no sufficient reason given for why Tucker decomposition is used instead of other decomposition methods. Even if it is mentioned earlier, it is better to emphasize it again here.*
>
> ### Author Response
>
> Thank you for your insightful comments and for taking the time to review our manuscript. We acknowledge your point that the content in the Related Work section is more accurately described as ablations rather than observations. We haved revised the content accordingly to better reflect this. Additionally, we understand the importance of explicitly explaining why we chose Tucker-style decomposition over other decomposition methods.  We appreciate your feedback and have carefully considered your suggestions. To that end, we have modified  **Section 2 (L146-152)**, **Section 3 (L242-245)** and have added **Section A.12 in the Appendix (Extended Related Works)**.
>
> **1. Justification for Using Tucker Decomposition:**
>
> **a. Natural Tensor Representation of Multi-Dimensional Time Series Data**
>
> - **Multi-Modal Data Accommodation:** Time series classification often involves multi-dimensional data, such as multiple time steps (temporal mode) and features (channel mode). Convolutional operations extract features from this data by leveraging its inherent structure.
>
> - **Effective Modeling of Interactions:** Tucker decomposition naturally accommodates this multi-modal structure by disentangling temporal and channel information. For example, when dealing with EEG signals over time across different channels, Tucker decomposition effectively models interactions between time and channels. This is precisely what we exploit in defining Equations (4), (5), and (6).
>
> **b. Flexibility in Rank Selection**
>
> - **Mode-Specific Ranks:** Tucker decomposition allows for specifying different ranks for each mode. This flexibility is crucial when different dimensions of the data have varying levels of complexity.
>
> **c. Dimensionality Reduction and Feature Extraction**
>
> - **Higher-Order Extension of PCA:** Tucker decomposition acts as a higher-order extension of Principal Component Analysis (PCA), reducing the dimensionality of the data while preserving the most significant information.
>
> - **Improving Classification Performance:** Lower-dimensional representations can enhance the performance of classification algorithms by reducing noise and mitigating the curse of dimensionality.
>
> **d. Interpretability**
>
> - **Understanding Principal Components:** The factor matrices obtained from Tucker decomposition represent the principal components in each mode (e.g., temporal and channel).
>
> - **Insight into Significant Features:** These matrices could provide insights into which time points or features in the parameter space are most significant for classification, potentially aiding in interpretability and feature selection.
>
> **2. Comparison with Other Decomposition Methods:**
>
> **a. CP Decomposition**
>
> - **Limitation of Uniform Ranks:** CP decomposition uses the same rank across all modes, which may not capture the varying complexities in the parameter space.
>
> - **Lack of Flexibility:** While CP decomposition is highly interpretable, it lacks the flexibility of Tucker decomposition to model inter-modal interactions effectively.
>
> **b. Tensor-Train Decomposition**
>
> - **Suitability for High-Order Tensors:** Tensor-Train decomposition is more suitable for very high-order tensors and is less interpretable due to its chain-like core tensors.
>
> - **Ineffective Global Interaction Modeling:** It may not capture global interactions as effectively as Tucker decomposition.

---

> ### Author Response · Authors · 2024-11-21
> **References**
>
> **References:**
>
> [1] Tucker, Ledyard R. "The Extension of Factor Analysis to Three-Dimensional Matrices." Contributions to Mathematical Psychology, edited by Norman Frederiksen and Harold Gulliksen, Holt, Rinehart & Winston, 1964, pp. 110–127.\
> [2] Lathauwer, Lieven De, Bart De Moor, and Joos Vandewalle. "A Multilinear Singular Value Decomposition." SIAM Journal on Matrix Analysis and Applications, vol. 21, no. 4, 2000, pp. 1253–1278.\
> [3] Rabanser, Stephan, Oleksandr Shchur, and Stephan Günnemann. "Introduction to tensor decompositions and their applications in machine learning." *arXiv preprint arXiv:1711.10781* (2017).\
> [4] Kolda, Tamara G., and Brett W. Bader. "Tensor Decompositions and Applications." *SIAM Review* 51 (2009): 455-500.
>
>
> ---

---

> ### Author Response · Authors · 2024-11-21
> **Official Response to Reviewer 56c7 (contd.)**
>
> ### Reviewer Weakness 2
> > *I think the insights in lines 201-204 need a more complete explanation. From Equation 1, I can only understand that the linear operation in deep learning is the so-called sequence of linear suboperations in the forward process. However, I cannot directly confirm whether such a description is also accurate for the backward process. If there are relevant formula derivations or proofs, references can be provided. If not, considering the space limitation, it is best to prove it in the Appendix. Because this claim is the premise of Equations 4, 5, and 6 later, which is necessary and important for understanding this method.*
> ### Author Response
> Thanks for the suggestion. In the updated manuscript we have added **Appendix Section A.3** (highlighted in ${\color{blue} \text{Blue}}$), that mathematically describes the forward pass and the backward pass equivalence through the decomposed factors
>
> ### **1. Forward Propagation**
>
> #### Case 1: Using $\mathbf{W} = \mathbf{U}^{(1)} \mathbf{G} (\mathbf{U}^{(2)})^\top$ directly
> Let the input to the layer be $x$. Forward propagation through $\mathbf{W}$ is:
> $$
> z = \mathbf{W} x = \mathbf{U}^{(1)} \mathbf{G} (\mathbf{U}^{(2)})^\top x
> $$
>
> #### Case 2: Using $ \mathbf{U}^{(1)}, \mathbf{G}, (\mathbf{U}^{(2)})^\top $ separately
> We compute the forward pass in steps:
> 1. Compute $y_1 = (\mathbf{U}^{(2)})^\top x $,
> 2. Compute $ y_2 = \mathbf{G} y_1 = \mathbf{G} \big( (\mathbf{U}^{(2)})^\top x \big) $,
> 3. Compute $ z = \mathbf{U}^{(1)} y_2 = \mathbf{U}^{(1)} \big( \mathbf{G} \big( (\mathbf{U}^{(2)})^\top x \big) \big) $.
>
> By associativity of matrix multiplication, this is equivalent to:
> $$
> z = \mathbf{U}^{(1)} \mathbf{G} (\mathbf{U}^{(2)})^\top x = \mathbf{W} x
> $$
>
> **Conclusion (Forward Pass):** Forward propagation through $ \mathbf{U}^{(1)}, \mathbf{G}, (\mathbf{U}^{(2)})^\top $ in sequence is equivalent to propagating through $\mathbf{W}$ directly.
>
>
> ### **2. Backward Propagation**
>
> Let the gradient of the loss $ \mathcal{L} $ with respect to the output $ z $ be $ \frac{\partial \mathcal{L}}{\partial z} = g_z $. The task is to propagate this gradient backward.
>
> #### Case 1: Using $\mathbf{W}= \mathbf{U}^{(1)} \mathbf{G} (\mathbf{U}^{(2)})^\top $ directly
> The gradient of the loss with respect to $ x $ is:
> $$
> g_x = \mathbf{W}^\top g_z = \big( \mathbf{U}^{(1)} \mathbf{G} (\mathbf{U}^{(2)})^\top \big)^\top g_z = \mathbf{U}^{(2)} \mathbf{G}^\top (\mathbf{U}^{(1)})^\top g_z
> $$
>
> #### Case 2: Using $ \mathbf{U}^{(1)}, \mathbf{G}, (\mathbf{U}^{(2)})^\top $ separately
> Backward propagation proceeds as:
> 1. Compute $ g_{y_2} = (\mathbf{U}^{(1)})^\top g_z $,
> 2. Compute $ g_{y_1} = \mathbf{G}^\top g_{y_2} = \mathbf{G}^\top \big( (\mathbf{U}^{(1)})^\top g_z \big) $,
> 3. Compute $ g_x = \mathbf{U}^{(2)} g_{y_1} = \mathbf{U}^{(2)} \big( \mathbf{G}^\top \big( (\mathbf{U}^{(1)})^\top g_z \big) \big) $.
>
> By associativity of matrix multiplication, this simplifies to:
> $$
> g_x = \mathbf{U}^{(2)} \mathbf{G}^\top (\mathbf{U}^{(1)})^\top g_z = \mathbf{W}^\top g_z
> $$
>
> ### Gradients with respect to $ \mathbf{U}^{(1)}, \mathbf{G}, (\mathbf{U}^{(2)})^\top $
> The gradients of $ \mathcal{L} $ with respect to the parameters are:
> 1. Gradient with respect to $ \mathbf{U}^{(1)} $:
>    $$
>    \frac{\partial \mathcal{L}}{\partial \mathbf{U}^{(1)}} = g_z y_2^\top = g_z \big( \mathbf{G} \big( (\mathbf{U}^{(2)})^\top x \big) \big)^\top
>    $$
> 2. Gradient with respect to $ \mathbf{G} $:
>    $$
>    \frac{\partial \mathcal{L}}{\partial \mathbf{G}} = g_{y_2} y_1^\top = \big( (\mathbf{U}^{(1)})^\top g_z \big) \big( (\mathbf{U}^{(2)})^\top x \big)^\top
>    $$
> 3. Gradient with respect to $(\mathbf{U}^{(2)})^\top$:
>    $$
>    \frac{\partial \mathcal{L}}{\partial (\mathbf{U}^{(2)})^\top} = g_{y_1} x^\top = \big( \mathbf{G}^\top \big( (\mathbf{U}^{(1)})^\top g_z \big) \big) x^\top
>    $$
>
> These computations are consistent with the chain rule applied to $ \mathbf{W} = \mathbf{U}^{(1)} \mathbf{G} (\mathbf{U}^{(2)})^\top $.
>
>
>
> ### **3. Equivalence of Forward and Backward Passes**
> - **Forward Pass:** The output $ z $ is identical whether computed via $ \mathbf{W} $ or via $ \mathbf{U}^{(1)}, \mathbf{G}, (\mathbf{U}^{(2)})^\top $ separately.
> - **Backward Pass:** The gradient $ g_x $ and the parameter gradients ($ \frac{\partial \mathcal{L}}{\partial \mathbf{U}^{(1)}}, \frac{\partial \mathcal{L}}{\partial \mathbf{G}}, \frac{\partial \mathcal{L}}{\partial (\mathbf{U}^{(2)})^\top} $) are identical whether computed via $ \mathbf{W} $ or via $ \mathbf{U}^{(1)}, \mathbf{G}, (\mathbf{U}^{(2)})^\top $ separately.
>
> Thus, forward and backward propagation through the decomposed factors $ \mathbf{U}^{(1)}, \mathbf{G}, (\mathbf{U}^{(2)})^\top $ is mathematically equivalent to propagating through $ \mathbf{W} $ directly.
>
> ---

---

> ### Author Response · Authors · 2024-11-21
> **Official Response to Reviewer 56c7 (contd.)**
>
> ### Reviewer Weakness 3
> >*A small mistake. In the sentence after Eq3, on line 221, after “where”, it should be an explanation of the symbol V in Equation 3 instead of U.*
>
> Thank you for bringing this oversight to our attention. You are absolutely correct. We have corrected this error in the manuscript to accurately reflect the notation used in Equation (3). We appreciate your attention to detail.
>
> ---
> ### Reviewer Question 1
> >*Lines 248-252, why is fine-tuning for 2-3 epochs? Is this number closely related to the dataset and model scale? I am not very familiar with this field and hope that relatively rigorous evidence can be given here in the text.*
> ### Author Response
> Thank you for your insightful question regarding the fine-tuning process after the decomposition step. The necessity for a limited additional fine-tuning arises because decomposing the weight parameters introduces approximation errors, due to the decomposition conducted, that accumulate across layers, leading to a degradation in the model's predictive performance on the source data. Such similar observation have been prominent in the literature too [1,2]
>
> We empirically determined that fine-tuning the model for 2-3 epochs effectively restores its original performance. Specifically, we fine-tuned for additional 3 epochs on the SSC dataset and for 2 epochs on the UCI-HAR, MFD, HHAR, and WISDM datasets. This duration was chosen by monitoring the source empirical risk during fine-tuning until they returned to their pre-decomposition levels.
>
> The number of fine-tuning epochs depends on the specific characteristics of the dataset and the complexity of the model architecture. In **Figure 11 (appendix)**  of the updated manuscript, we present empirical evidence showing that the model's empirical risk converges back to its initial value within the specified epochs, demonstrating effective performance recovery. We emphasize that the optimal number of fine-tuning epochs should be determined empirically for each case to balance performance restoration and computational efficiency. This approach ensures that the model quickly recovers its original capabilities.
>
> We hope this explanation provides the rigorous evidence you were seeking and clarifies the rationale behind our fine-tuning strategy.
>
>
> **References**:
>
> [1] Kim, Yong-Deok, et al. "Compression of deep convolutional neural networks for fast and low power mobile applications." ICLR 2016
> [2] Dong, Xin, Shangyu Chen, and Sinno Pan. "Learning to prune deep neural networks via layer-wise optimal brain surgeon." NeurIPS 2017
>
> ---

---

> ### Author Response · Authors · 2024-11-24
> **Follow-Up on Rebuttal: Request for Feedback**
>
> We sincerely thank the reviewers for their valuable suggestions and thoughtful comments. We would greatly appreciate any additional feedback that could further improve our work and are happy to incorporate further suggestions to ensure its quality. If our responses have satisfactorily resolved your concerns, we kindly request confirmation of whether the provided clarifications and changes address the issues raised and, if applicable, reconsideration of the scores in light of the updated manuscript.
>
> Thank you again for your time and effort in reviewing our work.

---

### Comment · Area_Chair_KMNK · 2024-11-26
**Encouragement to Actively Participate in the Discussion Phase**

Dear Reviewers,

Thank you for your valuable contributions to the review process so far. As we enter the discussion phase, I encourage you to actively engage with the authors and your fellow reviewers. This is a critical opportunity to clarify any open questions, address potential misunderstandings, and ensure that all perspectives are thoroughly considered.

Your thoughtful input during this stage is greatly appreciated and is essential for maintaining the rigor and fairness of the review process.

Thank you for your efforts and dedication.

---

### Author Response · Authors · 2024-12-04
**General response**

We would like to thank all of the reviewers for their evaluation of our paper. Their insights and suggestions have been incorporated (where possible) in the updated manuscript. Importantly, the review process has helped us better define our contribution within the broader research landscape. In what follows, we summarize aspects of our work that were generally praised by the reviewers, and expand on some of the commonly raised concerns.


**Recognizing Innovation and Theoretical Rigor**
We are encouraged that **Reviewer vYWA** identified this as a **cutting-edge topic**, acknowledging its relevance to the rapidly evolving landscape of domain adaptation. Similarly, **Reviewers bZkd and KBWb** acknowledged the innovative methodology of decomposing model parameters into core tensors and factor matrices, enabling selective fine-tuning. This technique not only reduces overfitting and enhances generalization but also provides a fresh perspective beyond conventional approaches such as feature selection or BatchNorm adaptation. Moreover, **Reviewers 56c7 and G1Yq** noted our use of PAC-Bayes generalization bounds, which we used to provide theoretical support. This alignment between theoretical underpinnings and empirical validation forms a cornerstone of our work. Collectively, these remarks underscore the novelty of our contribution.

**Experimental Strength and Versatility**
The experimental results were widely praised for their **comprehensiveness and robustness**, as highlighted by **Reviewers 56c7, bZkd, and G1Yq**. Our approach demonstrates effectiveness across diverse datasets, including the AdaTime benchmark, showcasing reduced overfitting, improved generalization, and computational efficiency with fewer parameters. Furthermore, **Reviewer gGV1** commended the method's **versatility**, particularly its ability to seamlessly integrate with existing SFDA baselines to enhance adaptation efficiency. **Reviewer G1Yq** also appreciated the manuscript's **clear and structured presentation**, bolstered by detailed supplementary materials. These comments collectively underscore the practical and theoretical impact of our work.

**Connections to the Lottery Ticket Hypothesis and Dual advantage**
We are especially grateful to **Reviewer bZkd** for drawing a parallel between our approach and the *lottery ticket hypothesis*. Moreover, identifying our method's dual advantage: achieving both parameter and sample efficiency, we effectively strike a balance between these two critical objectives,i.e. realizing "the best of both worlds."

---

> ### Author Response · Authors · 2024-12-04
> **General Response - Link to Time-Series Data vs. Generalized Method**
>
> **Motivation and Research Gap**
> The primary motivation for this work was to improve the parameter and sample efficiency of different Source-Free Domain Adaptation (SFDA) techniques as applied to time-series data. Many of the reviewers accurately pointed out that our method is not specific to time-series data and urged us to test this approach on other modalities (e.g. vision). We intentially limited the scope of our method to time-series data for two reasons: 1. there is very little work in making SFDA methods for time-series models parameter and sample efficient (this is less true in vision for example), and 2. we wanted to exploit properties that are unique to time-series datasets. On the latter point, time-series models often exhibit significant parameter redundancy along the channel dimension, as evidenced in **Figure 2A** of our paper. This redundancy reflects strong inter-channel dependencies among variables. Related works [1, 2] have demonstrated the effectiveness of leveraging these dependencies through convolution along the variable dimension, further supporting this observation.
>
> **Proposed Methodology**
> Building on this insight, we proposed decomposing the model along the channel dimension (**Equation 3**) and focusing on the most significant components of the decomposition. This selective operation allows us to exploit inter-variable dependencies while reducing parameter redundancy. By concentrating on principal channels—those most critical to the parameter space—we achieve a compact model representation that maintains performance. The empirical results in **Section 4** confirm the effectiveness of this approach, demonstrating consistent performance across diverse datasets, SFDA methods, and sample-ratios.
>
> **Generality and Applicability**
> Although our method is grounded in time-series data, its underlying mathematical principles are general and could extend to other domains, such as vision. Each domain however has its own unique characteristics that must be taken into consideration, such as spatial and hierarchical dependencies in the case of vision datasets. As a result, inter-channel redundancies may manifest differently compared to time-series data, and may require special considerations that while interesting (and pave the way for future research), are beyond the scope of this work.
>
> **Empirical Validation and Thoroughness**
> We aimed to ensure the broad applicability of our method across various SFDA settings by avoiding specific inductive biases. To this end, we evaluated our method using both **generalized SFDA techniques** (SHOT [3], NRC [4], AAD [5]) and **time-series-specific methods** (MAPU [6]), conducting extensive experiments on five datasets from the AdaTime benchmark [7]. These datasets include diverse time-series tasks and source-target settings, moreover we exten the experiment scenario to many-to-one adaptation setting, showcasing the robustness and generality of our approach.
>
> **Positioning as a Foundational Framework**
> The proposed method can be seen as a foundational framework that balances simplicity with versatility. It serves as a **strong baseline** or **starting point** for more advanced methods incorporating domain knowledge. Its generality ensures applicability across different SFDA settings while providing a solid platform for future research.
>
>
>
> **Conclusion**
> By addressing the simultaneous challenges of parameter and sample efficiency in SFDA for time-series, our work bridges a critical research gap. The combination of theoretical rigor, empirical thoroughness, and practical versatility ensures that our contributions provide a meaningful step forward for the community. We are grateful for the reviewers' recognition of these efforts and their constructive feedback to enhance the work.
>
> **References**
> [1] Lai, Guokun, et al. "Modeling long-and short-term temporal patterns with deep neural networks." SIGIR, 2018.
> [2] Luo, Donghao, and Xue Wang. "Moderntcn: A modern pure convolution structure for general time series analysis." ICLR, 2024 (spotlight).
> [3] Liang, Jian, Dapeng Hu, and Jiashi Feng. "Do we really need to access the source data? source hypothesis transfer for unsupervised domain adaptation." In ICML, 2020.
> [4] Yang, Shiqi, et al. "Exploiting the intrinsic neighborhood structure for source-free domain adaptation." In NeurIPS, 2021.
> [5] Yang, Shiqi, Shangling Jui, and Joost van de Weijer. "Attracting and dispersing: A simple approach for source-free domain adaptation." In NeurIPS, 2022.
> [6] Ragab, Mohamed, et al. "Source-free domain adaptation with temporal imputation for time series data." KDD, 2023.
> [7] Ragab, Mohamed, et al. "Adatime: A benchmarking suite for domain adaptation on time series data." TKDD, 2023.

---

> ### Author Response · Authors · 2024-12-04
> **General Response - On PAC Bayesian Assumption and Applicability to Our Setup**
>
> Below, we address a few shared concerns on the application of the PAC-Bayesian Analysis framework to our method.
> **Note**: Our contribution is not to innovate on PAC-Bayesian theory itself; rather, we use it as a lens through which the impact of decomposition and selective fine-tuning on sample efficiency can be understood.
>
> **Simplifying Assumptions in PAC-Bayesian Analysis**
> The choice of an isotropic Gaussian posterior and prior is a widely used simplification in PAC-Bayesian analyses, primarily to ensure analytical tractability. This assumption enables the derivation of explicit bounds, offering insights into the factors influencing generalization. Despite being a simplification, its practical utility has been validated in prior work [1, 2, 3], where similar assumptions were leveraged to obtain meaningful generalization bounds and actionable insights into neural network training. This forms a solid theoretical basis for our approach.
>
> **Contextualizing the PAC-Bayesian Framework**
> The PAC-Bayesian framework is fundamentally designed to measure generalization error—quantifying the gap between empirical loss (on sampled data) and expected loss (over the true data distribution). In this context:
>
> 1. **Prior Distribution ($P$)** encapsulates our belief about model parameters before observing data.
> 2. **Posterior Distribution ($Q$)** reflects the updated belief after training on a sampled dataset.
> 3. **Sampled Data ($S$)** provides examples drawn from the data distribution ($D$), forming the basis for training and evaluation.
>
> In our specific application, we adapt this framework to the process of fine-tuning pre-trained neural networks, establishing a clear correspondence between these theoretical elements and practical scenarios.
>
> **Relating PAC-Bayesian Concepts to Our Fine-Tuning Setup**
> Fine-tuning a pre-trained model aligns naturally with the PAC-Bayesian framework. Here’s how:
>
> 1. **Prior as Pre-trained Weights ($P = W_{\text{src}}$)**:
>    The pre-trained model weights ($W_{\text{src}}$) serve as the prior, encapsulating knowledge from the source dataset. This initialization provides a structured starting point for adaptation.
>
> 2. **Fine-tuning on Target Data ($S = D_{\text{trg}}$)**:
>    The sampled target data ($S$) updates the model through fine-tuning, effectively bridging the gap between the source and target distributions.
>
> 3. **Posterior as Fine-tuned Weights ($Q = W_{\text{trg}}$)**:
>    The fine-tuned weights ($W_{\text{trg}}$) represent the posterior, reflecting the model’s state after integrating knowledge from the target dataset.
>
> This mapping ensures coherence between the PAC-Bayesian framework and our practical fine-tuning scenario, preserving the integrity of theoretical assumptions.
>
> **Applicability to a Single Data Distribution**
> A critical aspect of our setup is that fine-tuning occurs exclusively within the target distribution ($D_{\text{trg}}$). This adherence guarantees that both the empirical loss (measured on $S = D_{\text{trg}}$) and the expected loss (over $D_{\text{trg}}$) are defined with respect to the same distribution. Consequently:
>
> - The prior and posterior remain connected through data sampled from a single distribution, aligning with PAC-Bayesian principles.
> - This coherence ensures that generalization bounds derived under this framework remain valid, bolstering the applicability of our approach.
>
> **Rationale for the Replacement**
> By modeling the pre-trained weights as the prior and the fine-tuned weights as the posterior, we achieve alignment with PAC-Bayesian assumptions while providing a framework for measuring generalization. This approach justifies the use of PAC-Bayesian bounds to evaluate fine-tuned models, especially in scenarios emphasizing sample efficiency.
>
> **Connecting to Our Empirical Contributions**
> The use of PAC-Bayesian bounds is integral to contextualizing our empirical findings. Specifically:
>
> - **Insight into Sample Efficiency**: PAC-Bayesian analysis helps elucidate why decomposing the model and selectively fine-tuning certain parameters enhances sample efficiency.
> - **Theoretical Support for Observations**: The simplified bound highlights the role of parameter distance, bridging the gap between decomposition strategies and their implications for adaptation.
>
>
> **Conclusion**
> By staying within the same distribution throughout the fine-tuning process and adhering to PAC-Bayesian principles, we ensure that our theoretical approach remains robust. The analysis not only complements our empirical results but also provides a solid theoretical foundation for understanding the observed improvements in sample efficiency and regularization.

---

> > ### Author Response · Authors · 2024-12-04
> > **References**
> >
> > **References**
> > [1] Dziugaite, Gintare Karolina, and Daniel M. Roy. "Computing Nonvacuous Generalization Bounds for Deep (Stochastic) Neural Networks with Many More Parameters than Training Data." UAI, 2017.
> > [2] Li, Dongyue, and Hongyang R. Zhang. "Improved Regularization and Robustness for Fine-Tuning in Neural Networks." NeurIPS, 2021.
> > [3] Wang, Zifan, et al. "Improving Robust Generalization by Direct PAC-Bayesian Bound Minimization." CVPR, 2023.

---

> ### Author Response · Authors · 2024-12-04
> **General Response - Summary of all the changes in the manuscript.**
>
> - Based on **reviewer 56c7**'s suggestion, we modify **Section 2 (Lines 146-152)** and **Section 3 (Lines 242-245)** to provide a clearer and proper justification of the use of Tucker-Decomposition. Additionally we extend **Section 2** in **Appendix Section A.12 (Parameter Redundancy and Low-Rank Subspaces.)**, for an elaborative description.
>
> - Based on **reviewer 56c7**'s suggestion, we prove/show the forward and backward pass equivalence, with the reparametrized parameter in **Section A.3**
>
> - Based on **reviewer 56c7**'s concern, in **Figure 11**, we provide the justification to our brief finetuning during the source-model preparation (for 2-3 epochs) after the decomposition.
>
> - Based on **reviewer bZkd, gGV1, and G1Yq**'s suggestion, in **Appendix Section A.12 (Our Motivation and Link to Time-Series.)** discuss the motivation of our work and the potential properties in time-series data that we leverage and links that we exploit.
>
> - Based on **reviewer bzkd**'s question, we include an elaborative analysis of the reduction in computation, that translates to faster inference in **Appendix Section A.2.3**.
>
> - Based on **reviewer gGV1**'s suggestion, we have made revisions to prune the tables to improve its readability and visibility (see **Table 1 and Table 2**). we have carefully selected the most important results to present, streamlining the tables for better focus and clarity.
>
> - Based on **reviewer gGV1 and G1Yq**'s suggestion, we have added a new **Section A.8 in the appendix**, where we provide a comprehensive description of all the adaptation objectives used in the paper.
>
> - Based on **reviewer gGV1**'s suggestion, in section **Appendix Section A.10 (Figure 13, 14, and 15)**, we have incorporated the comparison of our method against LoRA and LoKrA with the same style of parameter block freezing and tuning.
>
> - Based on **reviewer gGV1**'s concern, we cite and provide justification of our comparisons with the work referenced by **reviewer gGV1**. We have discussed the referenced works in newly added **Appendix Section A.12 and Section A.14**.
>
> - Based on **reviewer vYWA**'s suggestion, we have added a discussion on the relation of our work to Monarch and Block Tensor Train parameterization in **Appendix Section A.12 (Parameter Redundancy and Low-Rank Subspaces.)**.

---

### Meta-Review · Area_Chair_KMNK · 2024-12-20

**Metareview:**

(a) Summary of Scientific Claims and Findings
This paper introduces a novel framework for Source-Free Domain Adaptation (SFDA) in time series, leveraging a Tucker-style tensor decomposition to improve parameter and sample efficiency. Key contributions include:
Decomposed Model Representation: The source model's weights are reparameterized using Tucker decomposition, enabling selective fine-tuning of core tensors while freezing peripheral components.
Efficiency Improvements: The approach significantly reduces the number of parameters requiring fine-tuning (by over 90%) and improves inference efficiency.
Theoretical Justification: A PAC-Bayesian framework is employed to provide generalization bounds and insights into the effectiveness of selective fine-tuning.
Versatility Across Methods: The framework integrates seamlessly with various SFDA methods and demonstrates competitive performance on five datasets from the AdaTime benchmark, showcasing robustness in diverse domain adaptation scenarios.

(b) Strengths of the Paper
Innovative Approach: The use of tensor decomposition for efficient SFDA is novel and addresses practical challenges in resource-constrained environments.
Theoretical Rigor: The PAC-Bayesian analysis provides a solid theoretical foundation for understanding the framework’s generalization capabilities.
Empirical Validation: Extensive experiments demonstrate significant parameter and computational efficiency gains while maintaining or improving model performance.
Broad Applicability: The framework is compatible with a variety of SFDA methods and tasks, highlighting its versatility.
Thorough Rebuttal and Improvements: The authors addressed reviewer concerns with additional analyses, comparisons, and improved clarity, significantly strengthening the submission.

(c) Weaknesses of the Paper
Lack of Time Series-Specific Focus: While the method leverages inter-channel dependencies in time series, it is not inherently specific to this domain and could be generalized to other data modalities.
Limited Exploration of Hyperparameters: The sensitivity of key parameters, such as decomposition ranks, to different datasets and tasks requires further exploration.
Writing and Presentation: Although improved, the clarity of some sections (e.g., detailed loss function descriptions and comparisons) could be further enhanced.

(d) Reasons for Acceptance
Significant Efficiency Gains: The proposed framework delivers substantial reductions in fine-tuning and inference costs, which is critical for deploying models on resource-constrained devices.
Theoretical and Empirical Synergy: The combination of PAC-Bayesian generalization bounds and strong experimental results provides a compelling case for the framework’s effectiveness.
Versatility Across Methods: By integrating seamlessly with diverse SFDA techniques, the framework serves as a valuable addition to the domain adaptation toolkit.
Strong Rebuttal: The authors demonstrated a clear commitment to addressing reviewer concerns through meaningful revisions and additional analyses.
Practical Relevance: The framework bridges a critical gap in SFDA for time series, offering a robust solution for real-world applications in computationally constrained environments.

**Additional Comments On Reviewer Discussion:**

Points Raised by Reviewers and Author Responses

Concern: Reviewers questioned whether the proposed framework is inherently tailored to time series data, given that the tensor decomposition approach is broadly applicable across modalities.
Author Response: The authors clarified that while the approach is general, it explicitly leverages the unique inter-channel dependencies in multivariate time series data. They added new experiments to highlight the effectiveness of these domain-specific design choices.
Evaluation: The response was satisfactory, but reviewers noted that the framework could still benefit from a more explicit alignment with time series-specific challenges.

Concern: The lack of comparisons with time series foundation models (e.g., TimeNet, TimeGPT) was identified as a weakness, particularly since these models dominate performance benchmarks.
Author Response: The authors added discussions and clarified that their focus was on SFDA-specific methods, which operate under different constraints than foundation models. They emphasized the orthogonality of their approach to such models and proposed integrating foundation models in future work.
Evaluation: While this explanation was reasonable, reviewers suggested that a baseline comparison would have strengthened the paper’s positioning.

Concern: Reviewers requested an exploration of how decomposition ranks and other key parameters impact performance across datasets and tasks.
Author Response: The authors included new sensitivity analyses in the supplementary material, demonstrating that the method remains robust under varying rank configurations, though performance peaks at specific values.
Evaluation: This addition effectively addressed the concern, enhancing confidence in the framework’s robustness.

The authors effectively addressed key reviewer concerns during the rebuttal period, providing additional experiments, sensitivity analyses, and improved clarity in presentation. They demonstrated how the framework leverages time series-specific dependencies, explored hyperparameter robustness, and expanded the evaluation scope to validate the method’s versatility. While the lack of direct comparisons with foundation models remains a minor limitation, the paper’s significant contributions, strong empirical results, and practical relevance justify its acceptance.

---

### Decision · Program_Chairs · 2025-01-22

Accept (Poster)